



# Net ozone production and its relationship to NO$_x$ and VOCs in the marine boundary layer around the Arabian Peninsula

Ivan Tadic[1], John N. Crowley[1], Dirk Dienhart[1], Philipp Eger[1], Hartwig Harder[1], Bettina Hottmann[1], Monica Martinez[1], Uwe Parchatka[1], Jean-Daniel Paris[2], Andrea Pozzer[1,4], Roland Rohloff[1], Jan Schuladen[1], Justin Shenolikar[1], Sebastian Tauer[1], Jos Lelieveld[1,3], and Horst Fischer[1]

[1]Atmospheric Chemistry Department, Max Planck Institute for Chemistry, Mainz, Germany
[2]Laboratoire des Sciences du Climat et de l'Environnement, LSCE/IPSL, CEA-CNRS-UVSQ, Université Paris-Saclay, Gif-sur-Yvette, France
[3]Energy, Environment and Water Research Center, The Cyprus Institute, Nicosia, Cyprus
[4]International Centre for Theoretical Physics, Trieste, Italy

*Correspondence to*: Ivan Tadic (i.tadic@mpic.de)

**Abstract.** Strongly enhanced tropospheric ozone mixing ratios have been reported in the Arabian Basin, a region with intense solar radiation and high concentrations of ozone precursors such as nitrogen oxides and volatile organic compounds. To analyze photochemical ozone production in the marine boundary layer (MBL) around the Arabian Peninsula, we use ship-borne observations of NO, NO$_2$, O$_3$, OH, HO$_2$, HCHO, actinic flux, water vapor, pressure and temperature obtained during the summer 2017 Air Quality and Climate in the Arabian Basin (AQABA) campaign, compare them to simulation results of the ECHAM-MESSy atmospheric chemistry (EMAC) general circulation model. Net ozone production rates (NOPR) were greatest over the Gulf of Oman, the Northern Red Sea and the Arabian Gulf with median values of 14 ppb$_v$ day$^{-1}$, 16 ppb$_v$ day$^{-1}$ and 28 ppb$_v$ day$^{-1}$, respectively. NOPR over the Mediterranean, the Southern Red Sea and the Arabian Sea did not significantly deviate from zero; however, results for the Arabian Sea indicate weak net ozone production of 5 ppb$_v$ day$^{-1}$, and net ozone destruction over the Mediterranean and the Southern Red Sea with -2 ppb$_v$ day$^{-1}$ and -4 ppb$_v$ day$^{-1}$, respectively. Constrained by measured HCHO/NO$_2$-ratios, our photochemistry calculations show that net ozone production in the MBL around the Arabian Peninsula occurs mostly in a transition regime between NO$_x$- and VOC-limitation with a tendency towards NO$_x$-limitation except over the Northern Red Sea and the Oman Gulf.



## 1 Introduction

Revenues from exploitation of the great oil reserves in the states of and around the Arabian Peninsula have propelled remarkable economic development associated with industrialization and urbanization. Strong population growth and anthropogenic emissions of gases and particulates in the last few decades have resulted in the Middle East becoming a hotspot for air pollution and associated health effects, while it is also one of the regions worldwide where climate change is particularly rapid (Lelieveld et al., 2016a). Unique meteorological conditions such as intense solar radiation, high temperatures and aridity, as well as strong anthropogenic emissions of volatile organic compounds (VOCs) and $NO_x$ (= NO + $NO_2$) by on- and off-shore petrochemical industries, dense ship traffic, fossil energy production for air conditioning and desalination, and urban development are expected to further intensify in the future and contribute to photochemical ozone production (Lelieveld et al, 2009; Krotkov et al., 2016; Pfannerstill et al., 2019). Understanding the sources and sinks of $NO_x$ and other ozone precursors on and around the Arabian Peninsula is therefore of major importance for atmospheric chemistry studies, including the investigation of net ozone production rates (NOPR) (Monks et al., 2015; Reed et al., 2016; Bozem et al., 2017).

$NO_x$ plays a central role in atmospheric photochemistry (Nakamura et al., 2003; Tuzson et al., 2013; Reed et al., 2016). It is the primary precursor for tropospheric ozone ($O_3$), secondary organic aerosols and photochemical smog in urban areas (Hollaway et al., 2012; Javed et al., 2019). Main ground-based sources of NO and $NO_2$ are fossil fuel combustion and to a lesser extent bacterial processes in soils, and both lightning and aircraft emissions in the upper troposphere (Nakamura et al., 2003; Miyazaki et al., 2017; Javed et al., 2019). Transport of $NO_x$ in the atmosphere is relatively limited due to its short lifetime of a few hours (Reed et al., 2016). It is removed from the troposphere mainly by conversion to $HNO_3$ (via reaction with OH) during the day, or the formation of $N_2O_5$ (in the reaction of $NO_2$ with $NO_3$ at night-time), which also leads to formation of nitric acid by heterogeneous hydrolysis on aerosol surfaces (Crutzen, 1973; Liu et al., 2016; Reed et al., 2016). Ultimately, the deposition of $HNO_3$ constitutes the major loss process of $NO_x$ from the atmosphere. Ozone is a secondary pollutant that is photochemically formed in the troposphere from its precursors $NO_x$ and VOCs (Bozem et al., 2017; Jaffe et al., 2018). It is an important greenhouse gas, an atmospheric oxidant and the most important primary precursor for OH (Lelieveld et al., 2004; Monks et al., 2015; Bozem et al., 2017). $O_3$ in the planetary boundary layer causes health damage, notably respiratory diseases, and reduces crop yields (Monks et al., 2015; Jaffe et al., 2018).

$NO_x$ and $O_3$ mixing ratios in the troposphere vary from less than 20 $ppt_v$ and 10 $ppb_v$, respectively, for pristine conditions such as the remote marine boundary layer (MBL) up to mixing ratios of several hundreds of $ppb_v$ in regions with heavy automobile traffic and in international shipping lanes (for $NO_x$) and downwind of urbanized areas (for $O_3$) (Reed et al., 2016; Jaffe et al., 2018). Low $NO_x$ environments such as the clean MBL and the lower free troposphere are considered net ozone destruction regimes whereas the upper troposphere and areas with anthropogenic emissions of ozone precursors are regions



of net ozone production (Klonecki and Levy, 1997; Bozem et al., 2017). Measurements performed in the Houston Ship Channel revealed that NOPR can be of the order of several tens of ppb h$^{-1}$ even in marine environments (Zhou et al., 2014).

In the last decade much effort has been successfully devoted to the mitigation of $NO_x$ emissions over Europe and America, and levels of reactive nitrogen trace gases have decreased (Miyazaki et al., 2017). But in Asia, India and the Middle East, $NO_x$ emissions have substantially increased during the last decade so that the global $NO_x$ burden has essentially remained constant (Miyazaki et al., 2017). $NO_x$ emissions by ocean-going vessels have attracted considerable attention as they are reported to account for 15 % of the global $NO_x$ emission burden (Celik et al., 2019). Model calculations suggest that the

Arabian Gulf, with an estimated annual $NO_x$ emission density of about one ton km$^{-2}$ from ship traffic, is among the regions with highest $NO_x$ emission densities worldwide (Johansson et al., 2017). Although $NO_x$ emissions in the Red Sea and Arabian Sea areas were reported to be three and five times smaller than for the Arabian Gulf, respectively, these values are still 50-100 times larger than the emission density reported for the South Pacific Ocean, for example (Johansson et al., 2017).

In the present study, we characterize photochemical NOPR in the MBL around the Arabian Peninsula. In Sect. 2, the

campaign, instrument description, data processing and a description of the methods used in this study is presented. In Sect. 3, mixing ratios of nitrogen oxides and ozone around the Arabian Peninsula are reported. Based on concurrent measurements of $HO_x$, actinic flux, temperature and pressure, noontime $RO_2$ mixing ratios are estimated and used to calculate NOPR in the different regions around the Arabian Peninsula. Observation-based analysis of $HCHO/NO_2$-ratios will be used to distinguish between $NO_x$- or VOC-limited chemistry in the particular regions. A comparison of the results with data retrieved from the

3D global circulation model EMAC is also included.

## 2 Experimental

### 2.1 AQABA campaign

The AQABA ship campaign (**A**ir **Q**uality and Climate in the **A**rabian **Ba**sin) investigated the chemical composition of the MBL around the Arabian Peninsula. From late June to early September 2017, the *Kommandor Iona* Research and Survey

Vessel sailed from Toulon (France) to Kuwait and back in order to perform gas-phase and particle measurements in the region. The gas-phase and aerosol measurement instrumentation was housed in five laboratory containers on the front deck. Trace gases were sampled via a 6 m high, 20 cm diameter cylindrical stainless steel inlet (total mass flow of 10,000 SLM) installed on the front deck of the vessel. Air was drawn at a flow rate of 28.5 SLM into the laboratory container containing the HCHO and $NO_x$ measurements via a ½`` PFA tubing (residence times of ~ 4 s). For the other trace gas measurements

(excluding $H_2O$ vapor which was measured on the top of the ship mast in the front) sampling was achieved through bypass systems. The OH and $HO_2$ detection units were placed on the prow to allow for inlets with residence times less than 10 ms.



The *Kommandor Iona* left Malta in late June 2017 traversing the Mediterranean Basin, the Suez Canal and the Northern Red Sea. A 3 day stop-over at KAUST University (Saudi Arabia) was made from July 11[th] to July 13[th] before passing the Southern Red Sea area. On July 17[th], we briefly stopped at Djibouti port before passing the Gulf of Aden, the Arabian Sea

and the Gulf of Oman. Kuwait at the northern end of the Arabian Gulf marked the turning point of the ship cruise where, during a second 3-day stop-over, scientific staff was exchanged. The *Kommandor Iona* started the second leg on August 03[rd] 2017 arriving in Toulon (France) in early September 2017 without any further stops. Figure 1 shows the ship's route subdivided into six different regimes.

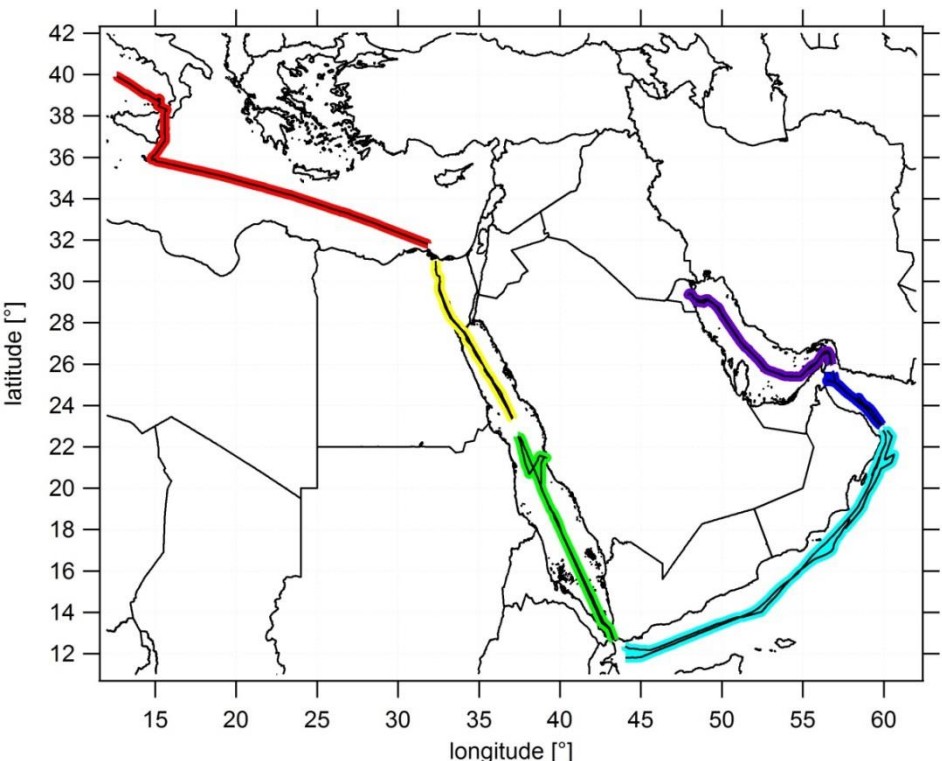

**Figure 1: Ship cruises during both legs and color-coded subdivision into six different regimes. The following abbreviations will be used: AG for Arabian Gulf (purple), OG for Oman Gulf (dark blue), AS for Arabian Sea (blue), SRS for Southern Red Sea (green), NRS for Northern Red Sea (yellow), M for Mediterranean (red).**

To enhance the statistical significance of our results and due to comparable signatures of the $NO_x$ and $O_3$ measurements in

the northern part of the Red Sea, the Suez Gulf and the Suez Canal, we have combined these regions which are represented by the 'Northern Red Sea' (NRS). For the same reasons we have merged the Gulf of Aden with the Arabian Sea (AS). Table 1 shows the range of latitudinal and longitudinal coordinates of the different regions. See supplementary table ST1 for a detailed day to day description of the route.





**Table 1: Range of latitudinal and longitudinal coordinates and dates during both legs of the different regions.**

| region (abbreviation) | latitudinal range | longitudinal range | Date (1[st] leg) | Date (2[nd] leg) |
|---|---|---|---|---|
| Mediterranean (M) | 31.810° N- 39.923° N | 12.620° E- 31.850° E | --- | 25.08.2017 – 31.08.2017 |
| Northern Red Sea (NRS) | 23.343° N- 30.986° N | 32.305° E- 37.085° E | 03.07.2017 – 08.07.2017 | 21.08.2017 – 24.08.2017 |
| Southern Red Sea (SRS) | 12.672° N- 22.494° N | 37.411° E- 43.327° E | 09.07.2017 – 16.07.2017 | 17.08.2017 – 20.08.2017 |
| Arabian Sea (AS) | 11.797° N- 22.782° N | 44.035° E- 60.636° E | 18.07.2017 – 24.07.2017 | 07.08.2017 – 16.08.2017 |
| Oman Gulf (OG) | 23.050° N- 25.622° N | 56.492° E- 59.913° E | 24.07.2017 – 27.07.2017 | 05.08.2017 – 07.08.2017 |
| Arabian Gulf (AG) | 25.396° N- 29.425° N | 47.920° E- 56.772° E | 27.07.2017 – 31.07.2017 | 03.08.2017 – 05.08.2017 |

## 2.2 Measurements of nitrogen oxides during AQABA

Chemiluminescent detection of NO and $NO_2$ is a widely applied method to quantify mixing ratios from the $ppm_v$ down to the low $ppt_v$ range (Nakamura et al., 2003; Pollack et al., 2011; Hosaynali Beygi et al., 2011; Reed et al., 2016). During AQABA

we deployed a compact, robust and commercially available two-channel chemiluminescence instrument CLD 790 SR (ECO Physics AG, Dürnten, Switzerland) that has been optimized for in situ field measurements during the last decade (Hosaynali Beygi et al., 2011). The measurement principle of the CLD is based on the addition of $O_3$ to NO to produce stoichiometric quantities of excited state $NO_2^*$ that will emit an infrared photon ($\lambda > 600$ nm) forming the chemiluminescent detection principle for NO (Drummond et al., 1985; Reed et al., 2016). Both channels feature an identical layout and were operated at

a mass flow of 1.5 SLM during AQABA. One channel of the CLD ($NO_c$-channel) has additionally been equipped with a LED solid state photolytic converter (Droplet Measurement Techniques, Boulder, Colorado) installed upstream of the $O_3$ addition to selectively photolyze $NO_2$ to NO, which is subsequently measured. In this section, we will concentrate on modifications made prior to the campaign and especially on operational conditions of the photolytic converter during the campaign. Further details on the measurement principle are described elsewhere (Pollack et al., 2011; Hosaynali Beygi et al.,

2011; Reed et al., 2016).

During AQABA, the cylindrical photolytic converter (length 14 cm, volume 0.079 l) was operated at a constant pressure of 95 hPa yielding a residence time of ~ 0.3 s. The photolytic $NO_2$ converter features a set of 50 UV LED units attached to each end of the converter. The emission profile of the UV LED units was characterized in laboratory measurements to peak at 398





nm with a Full Width at Half Maximum (FWHM) of 16 nm. UV-induced positive bias in the $NO_2$ measurements due to
photolysis of HONO, $BrONO_2$, $NO_3$ and $ClNO_2$ to produce NO was characterized ahead of the campaign to be 7.7 %, 7.2 %,
5.6 % and 1.5 % of the respective ambient concentration of HONO, $BrONO_2$, $NO_3$ and $ClNO_2$ respectively, and was
neglected for the observations in this study due to small daytime concentrations of these molecules in the MBL. To calculate
the UV-induced positive bias we used absorption cross sections from the MPI-Mainz UV/VIS Spectral Atlas of Gaseous
Molecules (Keller-Rudek et al., 2013). To limit wall loss of $NO_2$, the inner cavity surface is made of PTFE
(polytetrafluoroethylene), which may potentially provide a reservoir (via surface adsorption) for $NO_y$ that can thermally
dissociate to increase the background signal of the $NO_2$ measurement (Reed et al., 2016). The conversion efficiency $K_e$ of
the photolytic $NO_2$ conversion was estimated by gas phase titration (SYCOS K-GPT-DLR, ansyco, Karlsruhe, Germany)
several times before, during and after the campaign at $(29.4 \pm 0.9)$ % allowing the calculation of $NO_2$ concentrations by
$[NO_2] = \frac{[NO_c] - [NO]}{K_e}$. To avoid chemical interferences due to adding ozone in excess during a gas phase titration, a small but
not vanishing amount of NO has always been left unoxidized during gas phase titrations.

During AQABA, regular dry zero-air measurements as well as NO and $NO_2$ calibrations were performed autonomously over
a 10 minute period every 6 hours to accurately quantify the instrumental background and to correct for sensitivity drifts. An
autonomous cycle of '2 min zero air measurements – 2 min NO calibration – 2 min zero air measurement – 2 min $NO_2$
calibration – 2 min zero air measurement' was implemented. Continuous flows NO and $NO_2$ calibration gases were added to
the synthetic airflow or directed to a pump by switching solenoid valves. The NO calibration standard $(1.954 \pm 0.039)$
$ppm_v$ NO in $N_2$, Air Liquide, Germany) used during the campaign was compared to a primary standard $(5.004 \pm 0.025)$
$ppm_v$ (NPL, Teddington, UK) after the campaign yielding an effective NO mixing ratio of $(2.060 \pm 0.057)$ $ppm_v$ in the NO
calibration gas. Both zero air measurements and NO calibrations were performed with a total flow of 3.44 SLM achieving an
overflow of 0.44 SLM to guarantee ambient air free standard measurements by adding the calibration gas to the zero air
flow. During AQABA, NO calibrations at 2.5 $ppb_v$ were achieved. During the first leg of the campaign, zero air was sampled
from a bottle (Westfalen AG, Germany), whereas during the second leg zero air was generated from a zero air generator (Air
Purifier CAP 180, acuraLine). Zero air measurements generated with the zero air generator were statistically not
significantly different from those achieved by a bottle. To correctly account for the photomultiplier background and
chemical interferences due to reactions of ozone with ambient alkenes additional pre-chamber measurements were
performed every 5 minutes as well as at the beginning of zero air measurements and calibrations for 25 s each. This
correction is removing a large fraction of the interference signal from alkenes. However, in regions where alkene
concentrations are notably high, the CLD is prone to enhanced backgrounds. A schematic setup of the two-channel CLD
instrument is given in Figure 2.





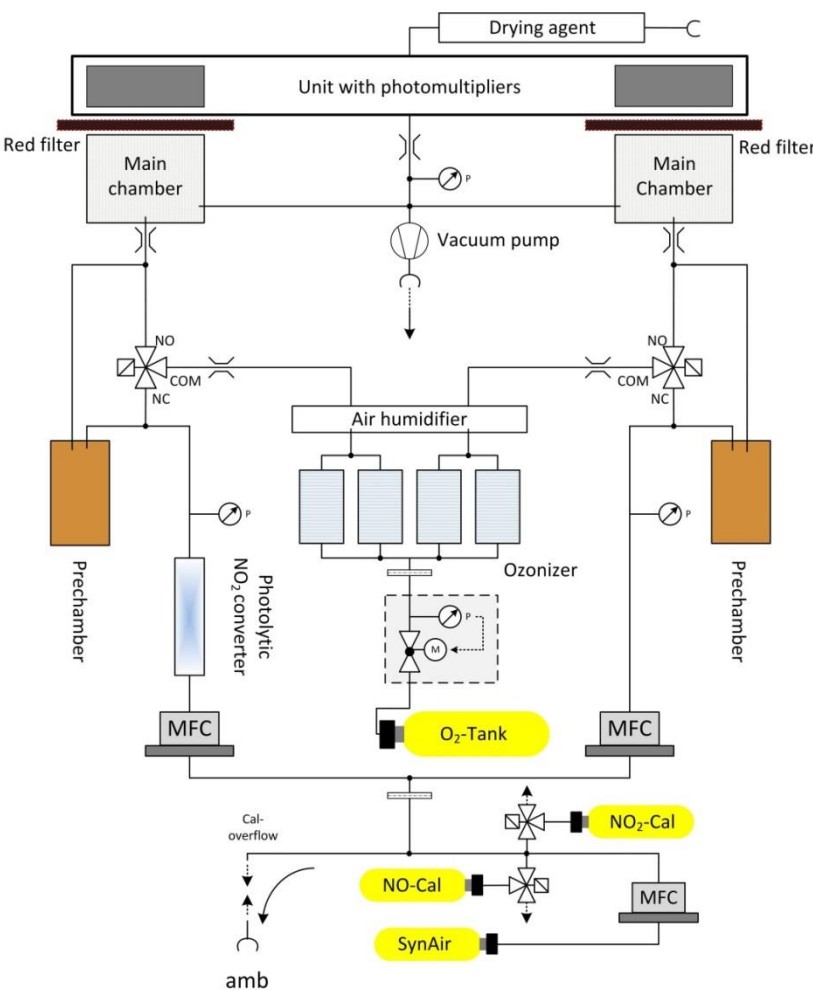


**Figure 2: Schematic setup of the two channel CLD instrument in the configuration used during AQABA. NO and NO$_2$ calibration gases were running continuously and were added to the zero airflow by switching the respective solenoid valves.**

The total measurement uncertainty (TMU) in the NO data is 5.5 % at a 5 min integration time and a confidence level of $1\sigma$.

The limit of detection in the NO channel was estimated as the full width at half maximum of the frequency distribution of all zero air measurements obtained during the campaign to be 9 ppt$_v$ at a 5 min integration time and a confidence level of $1\sigma$. The TMU in the NO$_2$ data is 7 % $\pm$ 112 ppt$_v$ at a confidence level of $1\sigma$ and an integration time of 5 min. As the zero air measurements in the NO$_2$ channel produced an increased background affected by memory effects after exposure to high NO$_x$ levels e.g. during measurements of stack emissions, the NO$_2$ raw data were initially processed without converter background

subtraction. As we therefore expect the CLD NO$_2$ data to be offset due to not being initially background corrected, the converter background was estimated at 112 ppt$_v$ from the centre of a Gaussian fit representing the difference of 1-minute averaged CLD NO$_2$ and concurrent cavity ring-down spectroscopy (CRDS) NO$_2$ measurements for data points below 10 ppb$_v$ as we expect the NO$_2$ CLD to be offset due to not being background corrected. Setting the threshold for calculating the





difference of the two concurrent data sets to 10 ppb$_v$ is somewhat arbitrary, however, changing this limit to 5 ppb$_v$ or 20 ppb$_v$
does not significantly vary the estimated offset of the CLD NO$_2$ data. The offset correction of 112 ppt$_v$ was taken as the
ultimate absolute measurement uncertainty of the CLD NO$_2$ measurement. Further corrections of to the final CLD data
include residence time corrections as well as corrections for NO and O$_3$ losses and the subsequent formation of NO$_2$ in the
sampling line (Ryerson et al., 2000).  Both NO and NO$_2$ CLD data have also been corrected for nonlinearities for
concentrations higher than 55 ppb$_v$, as experienced during probing of stack emissions.

**2.3 Further measurements used in this study**

An extensive set of concurrent measurements providing mixing ratios of O$_3$, NO$_2$, HCHO, OH, HO$_2$, absolute humidity and
actinic flux, temperature and pressure data obtained during AQABA was used in this study. Ozone was measured with an
absorption photometer (Model 202 Ozone Monitor, 2B Technologies, Boulder, Colorado) based on the well-established
absorption of the mercury line in the Hartley band at 254 nm (Viallon et al., 2015). Eliminating water and particle
interferences during sampling was achieved via sampling through a nafion tube and a Teflon filter. The ozone monitor was
zeroed ten times during the campaign. NO$_2$ was further measured by cavity ring-down spectroscopy (Sobanski et al., 2016)
and used for correcting the instrumental background of the CLD NO$_2$ data, as described above (the correction was taken as
the ultimate absolute measurement uncertainty in the CLD NO$_2$ data). Note that in this study we will use the NO$_2$ CLD data
rather than the NO$_2$ CRDS data as the temporal coverage of the CLD NO$_2$ data over the course of the campaign is about 60
% compared to about 35 % for the cavity ring-down measurement. Formaldehyde (HCHO) was measured with an Aerolaser
4021 (AERO-LASER GmbH, Garmisch-Partenkirchen, Germany), which is a fully automatized monitor based on the
Hantzsch technique (Kormann et al., 2003). H$_2$O measurements were obtained using a cavity ring-down spectroscopy
monitor (PICARRO G2401, Santa Clara, California) supervised by Laboratoire des Sciences du Climat et de
l'Environnement (LSCE) (Kwok et al., 2015). Measurements of OH and HO$_2$ were performed with the custom-built
190    **H**ydr**O**xyl **R**adical measurement **U**nit based on fluorescence **S**pectroscopy (HORUS) instrument based on laser-induced
fluorescence (LIF) spectroscopy of the OH molecule and NO titration of HO$_2$ to OH followed by LIF spectroscopy detection
of the OH molecule (Martinez et al., 2010; Regelin et al., 2013). HO$_2$ data used in this study is still preliminary due to not
yet quantified interference of organic peroxy radicals RO$_2$. This correction is expected to reduce the final HO$_2$ concentrations
by less than 20 % around noontime, which will not significantly change the results presented here. Wavelength resolved
195    down-welling actinic flux was measured with a spectral radiometer (model CCD Spectroradiometer 85237). The *j*-values for
NO$_2$ and O$_3$ were not corrected for upwelling UV radiation and were estimated to have a ~ 10 % measurement uncertainty
(Meusel et al., 2016). The radiometer was installed 10 m above sea level, respectively 5 m above the front deck surface.
Decreases in sensitivity due to sensor contamination with e.g. sea-spray were corrected with a linear interpolation between
two (daily) cleaning events. Temperature and pressure measurements were performed with the Shipborne **Eu**ropean
200    **C**ommon **A**utomatic **W**eather **S**tation (EUCAWS), a weather station specifically designed for ships. The weather station
incorporates sensors, processing units, satellite positioning and communication systems in one device and is implemented





and coordinated by the European National Meteorological Service EUMETNET. Table 2 lists the measurement methods and the TMU for each observation.

**Table 2: List of observations and gas phase measurements during AQABA. The TMU at a confidence level of 1σ and at the particular temporal resolution as well as a reference of the measurement operability are given.**

| Molecule | Method | TMU | References |
|---|---|---|---|
| NO | chemiluminescence | 6 % | Hosaynali Beygi et al., 2011 |
| $NO_2$ | photolysis-chemiluminescence | 7 % | Hosaynali Beygi et al., 2011 |
| $NO_2$ | cavity ring-down spectroscopy | 7 % | Sobanski et al., 2016 |
| $O_3$ | UV absorbance | 2 % | Viallon et al., 2015 |
| OH | LIF | 20 % | Martinez et al., 2010 |
| $HO_2$ | NO titration / LIF | 20 % | Martinez et al., 2010 |
| HCHO | Hantzsch technique | 13 % | Kormann et al., 2003 |
| $H_2O$ | cavity ring-down spectroscopy | 5 % | Kwok et al., 2015 |
| actinic flux | spectral radiometer | 10 % | Meusel et al., 2016 |

The *Kommandor Iona* Research and Survey Vessel sailed whenever possible with the wind coming from the bow to avoid contamination by stack emissions. However, based on the relative wind direction, the variability in NO as well as the temporal evolution of $NO_x$, $SO_2$, and $O_3$ sections of data in which the air mass was contaminated by the ship's stack were identified. All data used here to calculate $RO_2$ and NOPR have been filtered to remove contaminated air masses. Altogether, 21 % of the sampling time was potentially contaminated by the ship exhaust of the KI of which 87 % occurred on the first leg. During the second leg the ship sailed against the wind and most of the data was free of stack contamination. Our analysis is based on a 5-minute running mean for each data set, whereby only averages that have been calculated at a temporal coverage greater than 30 % have been used. A time series of the NO, $NO_2$ (both CLD), $O_3$ and $j(NO_2)$ measurements is given in the supplements Figures S2 and S3.

NO and $NO_2$ were measured from July 03[rd] to August 31[st], $O_3$ was measured from June 22[nd] to September 01[st], HCHO from July 01[st] to August 31[st] and OH and $HO_2$ from July 18[th] to August 31[st]. For the analysis of peroxy radicals $RO_2$ and NOPR around the Arabian Peninsula we have removed data measured during the stop-overs in Jeddah (July 11[th] to July 13[th]), Kuwait (July 31[st] to August 03[rd]) and during bunkering at Fujairah City (August 06[th], 07:00 – 15:00 UTC). Due to $HO_x$ data being available from July 18[th] onward, we have limited the net ozone production analysis to the period after this date.





## 2.4 Methods

The so-called $NO_x$-$O_3$-null cycle represents a rapid daytime cycling between NO, $NO_2$ and $O_3$. Solar UV radiation photolyzes $NO_2$ to NO and $O(^3P)$ (R1) which will reform $O_3$ in the subsequent reaction with molecular oxygen $O_2$ (R2)
(Leighton, 1961). NO and $O_3$ react to form $NO_2$ and $O_2$ (R3). R1, R2 and R3 constitute a so called null cycle which establishes photostationary steady state (PSS) for both $NO_x$ and $O_3$ in mid latitudes during noon time on a time scale of ~100 s (Thornton et al., 2002; Mannschreck et al., 2004).

$$NO_2 + h\nu \; (\lambda < 424 \text{ nm}) \; \rightarrow NO + O(^3P) \tag{R1}$$
$$O_2 + O(^3P) + M \rightarrow O_3 + M \tag{R2}$$
$$NO + O_3 \rightarrow NO_2 + O_2 \tag{R3}$$

Under the assumption of PSS, the Leighton Ratio $\varphi$ is unity (Leighton, 1961)

$$\varphi = \frac{j(NO_2) \cdot [NO_2]}{k_{NO+O_3} \cdot [NO][O_3]} = 1 \tag{1}$$

with $j(NO_2)$ being the $NO_2$ photolysis rate [s$^{-1}$]. In low $NO_x$ environments (< 100 ppt$_v$) previous studies have indicated that further NO oxidizing trace gases such as peroxy radicals ($HO_2$, $RO_2$) and halogen monoxides (XO) may result in a deviation
from unity (Nakamura et al., 2003; Hosaynali Beygi et al., 2011; Reed et al., 2016).

$$NO + HO_2 \rightarrow NO_2 + OH \tag{R4}$$
$$NO + RO_2 \rightarrow NO_2 + RO \tag{R5}$$
$$NO + XO \; \rightarrow NO_2 + X \tag{R6}$$

In the present study we include $HO_2$ and $R_iO_2$ into the production term for $NO_2$.

$$j(NO_2) \cdot [NO_2] = k_{NO+O_3} \cdot [NO][O_3] + k_{NO+HO_2} \cdot [NO][HO_2] + [NO] \cdot \sum_i k_{NO+R_iO_2} \cdot [R_iO_2] \tag{2}$$

Assuming that the temperature-dependent rate coefficient for the reaction of each particular peroxy radical $R_iO_2$ with NO equals the rate $k_{NO+HO_2}$ for Reaction R4 (Hauglustaine et al., 1996; Cantrell et al., 1997; Thornton et al., 2002), we can combine $HO_2$ and the sum of all organic peroxy radicals $R_iO_2$ to the entity $RO_2$ that can be estimated using the steady state equation

$$[RO_2] = \frac{j(NO_2) \cdot [NO_2] - k_{NO+O_3} \cdot [NO][O_3]}{k_{NO+HO_2} \cdot [NO]}. \tag{3}$$



However, the steady state assumption is not valid if the sampled air parcel is affected by fresh emissions or fast changes in the actinic flux (Thornton et al., 2002). After sampling a fresh emission e.g. a ship plume, for which $NO_x$ went up typically to values of several tens of $ppb_v$ with simultaneous titration in $O_3$, we assume that PSS is re-established on a time scale of 2 minutes (Thornton et al., 2002; Mannschreck et al., 2004). To best approximate PSS in our analysis we have restricted the

estimation of $RO_2$ on time frames $\pm$ 2 h around noontime for which we expect the smallest relative changes in the actinic flux. Noontime for each day was determined as the centre of a Gaussian fit that was applied to the actinic flux data. We applied a Gaussian Fit to the actinic flux data as this fitting method is sufficient to estimate the centre of the diurnal actinic flux. To further limit the effect of periods for which PSS is not fulfilled, we use the median instead of the average that is often disproportionately biased by strong $NO_x$ sources nearby. See supplements Tables ST2, ST4 and ST6 for detailed

statistics and a further motivation on regional averages and median values. See supplements Figure S1 for a detailed illustration of the calculation of the fraction of the noontime integral.

A further part of the analysis will be the investigation of NOPR. Ozone production is initiated by reactions that produce $HO_x$, for which primary production is from the photolysis of ozone, formaldehyde, nitrous acid (HONO) and hydrogen peroxide ($H_2O_2$) (Thornton et al., 2002; Hens et al., 2014; Lu et al., 2016; Mallik et al., 2018). The production of ozone can be

approximated by the rate of oxidation of NO with $RO_2$ ($HO_2 + \Sigma_i R_i O_2$) to form $NO_2$ that will rapidly form $O_3$ (R1-R2) (Bozem et al., 2017). For $RO_2$ we use the result from Eq. 3 that incorporates $HO_2$ and the sum of all further peroxy radicals $\sum_i R_i O_2$ (Parrish et al., 1986; Thornton et al., 2002).

$$P(O_3) = k_{NO+HO_2} \cdot [NO][RO_2] \tag{4}$$

Photochemical $O_3$ loss is mainly due to photolysis ($\lambda < 340$ nm) in the presence of water vapour and the reactions of ozone

with OH and $HO_2$ (Bozem et al., 2017).

$$O_3 + h\nu \ (\lambda < 340 \ \text{nm}) \ \rightarrow O_2 + O(1D) \tag{R7.1}$$

$$O(1D) + H_2O \ \rightarrow 2OH \tag{R7.2}$$

$$O_3 + OH \ \rightarrow HO_2 + O_2 \tag{R8}$$

$$O_3 + HO_2 \ \rightarrow OH + 2O_2 \tag{R9}$$

$\alpha$, the fraction of O(1D) that reacts with $H_2O$

$$\alpha = \frac{k_{O(1D)+ H_2O}[H_2O]}{k_{O(1D)+ H_2O}[H_2O] + k_{O(1D)+M}[M]} \tag{5}$$

was $(10.6 \pm 2.2)$ % during AQABA with a quasi linear dependence on water concentrations. Furthermore, ozone is lost due to reactions with alkenes (R12) and halogen radicals (R13).





$$O_3 + \text{alkenes} \rightarrow \text{radicals} \tag{R12}$$

$\quad O_3 + X \rightarrow O_2 + XO \tag{R13}$

We find that the loss rate is dominated by the photolysis of ozone with subsequent reaction of $O(^1D)$ with $H_2O$, was $60 - 80$ % of the total loss rate, followed by the reaction of $O_3$ with $HO_2$, which makes up $10 - 30$ % (note that the uncertainty in $HO_2$ radical concentrations mentioned above has no significant influence on the total $O_3$ loss rate, due to its small contribution). The remaining fraction (10-30 %) is due to the reaction of $O_3$ with OH. The reaction of ozone with ethene is
$\quad$ on average $0.005 - 0.01$ ppb$_v$ h$^{-1}$ and therefore generally less than 2 % of the total ozone loss rate (Bourtsoukidis et al., 2019). The reaction of $O_3$ with all alkenes will hence be neglected. Halogen radicals were not measured during AQABA and will not be incorporated into our study. Therefore the noon-time chemical ozone loss rate can be summarized by

$$L(O_3) = [O_3] \cdot \left(\alpha \cdot j(O1D) + k_{OH+O_3} \cdot [OH] + k_{HO_2+O_3}[HO_2]\right). \tag{6}$$

NOPR presented in this study is finally calculated as the difference of Eq. 4 and Eq. 6.

$\quad \text{NOPR} = k_{NO+RO_2}[NO][RO_2] - [O_3] \cdot \left(\alpha \cdot j(O1D) + k_{OH+O_3}[OH] + k_{HO_2+O_3}[HO_2]\right). \tag{7}$

Under the assumption of constant chemical composition for a given day, the NOPR is expected to have a diel cycle following the measured actinic flux. Hence integrating the estimated NOPR over the course of a day based on the particular fractional noontime integral of $j(NO_2)$ will yield a diurnal value for NOPR. A detailed calculation of the diurnal fractional integrals is given in the supplements Figure S1. Note that all reaction rate constants used are from the IUPAC Task Force on
$\quad$ Atmospheric Chemistry Chemical Kinetic Data Evaluation (Atkinson et al., 2004). Indications whether a chemical regime is $NO_x$-limited or VOC-limited can be derived from the ratio of HCHO to $NO_2$. Former studies have derived HCHO/$NO_2$-ratios from satellite measurements to establish whether ozone production is $NO_x$-limited or VOCs-limited. The results indicate $NO_x$-limitation for HCHO/$NO_2 > 2$ and prevailing VOC-limitation for HCHO/$NO_2 < 1$ (Duncan et al., 2010).

**2.5 ECHAM/MESSy Atmospheric Chemistry (EMAC) model**

$\quad$ EMAC is a 3D general circulation model that includes a variety of sub-models to describe numerous processes in the troposphere, their interaction with oceans and land surfaces and incorporates anthropogenic influences. Here we use the second development cycle of the Modular Earth Submodel System (MESSy2) (Jöckel et al., 2010) and ECHAM5 (Röckner et al., 2006) which is the fifth generation European Centre Hamburg general circulation model in the T106L31 resolution (corresponding to a quadratic grid of roughly 1.1° and 1.1°). The model has 31 vertical pressure levels and involves the
$\quad$ complex organic chemistry mechanism MOM (Mainz Organic Mechanism) as presented by Sander et al. (2019) that includes further developments of the version used by Lelieveld et al. (2016b). Here we use the lowest pressure level in a terrain following coordinates (equivalent to the surface level) and simulations of NO, $NO_2$, $O_3$, OH, $HO_2$, $j(NO_2)$ and $j(O^1D)$.





The sum of peroxy radicals was estimated as the sum of all radicals $R_iO_2$ with less than 4 carbon atoms. Net ozone production based on data retrieved from EMAC was estimated as

$$\text{NOPR} = [\text{NO}] \cdot \left( k_{\text{NO}+\text{HO}_2}[\text{HO}_2] + \sum_i k_{\text{NO}+\text{R}_i\text{O}_2}[\text{R}_i\text{O}_2] \right) - [\text{O}_3] \cdot \left( \alpha \cdot j(\text{O1D}) + k_{\text{OH}+\text{O}_3}[\text{OH}] + k_{\text{HO}_2+\text{O}_3}[\text{HO}_2] \right). \qquad (8)$$

A list of all included peroxy radicals $R_iO_2$ for the reaction with NO is given in the supplementary Table ST9.

### 3 Results and discussions

### 3.1 $NO_x$ and $O_3$ in the MBL around the Arabian Peninsula

During AQABA $NO_x$ mixing ratios varied over five orders of magnitude with lowest values of less than 50 $ppt_v$ observed in
relatively pristine regions and highest values of several hundred $ppb_v$ found in the vicinity of megacities or nearby passing ships. Ozone mixing ratios ranged from values of less than 20 $ppb_v$, detected over the Arabian Sea, to more than 150 $ppb_v$ during episodes of severe pollution. Figures 3a) and 3b) show distributions of $NO_x$ measured during the first and second leg of the campaign (range from 0.1 $ppb_v$ to 20 $ppb_v$) while Figure 3c) and 3d) show corresponding ozone mixing ratios covering a range from 20 $ppb_v$ to 100 $ppb_v$, respectively. A classification of the different regions based on Box-Whisker-Plots,
including the 25-75-percentile interval (box) and whiskers for the 10-90-percentile interval, is shown in Figure 4 and Figure 5 for $NO_x$ and $O_3$, respectively. As average $NO_x$ is often influenced by fresh, localized emissions, we have included the median (black bar) instead of the average in the Box-Whisker-Plot for $NO_x$, which is less sensitive to extreme values. For $O_3$, although the difference between median and mean is mostly negligible, we also use the median in Figure 5. $NO_x$ and $O_3$ averages, medians, standard deviations, 1st and 3rd quantiles and the number of data points quantified per region are given in
the supplementary Table ST2. See supplementary Figure S4 for OH and $HO_2$ mixing ratios around the Arabian Peninsula.



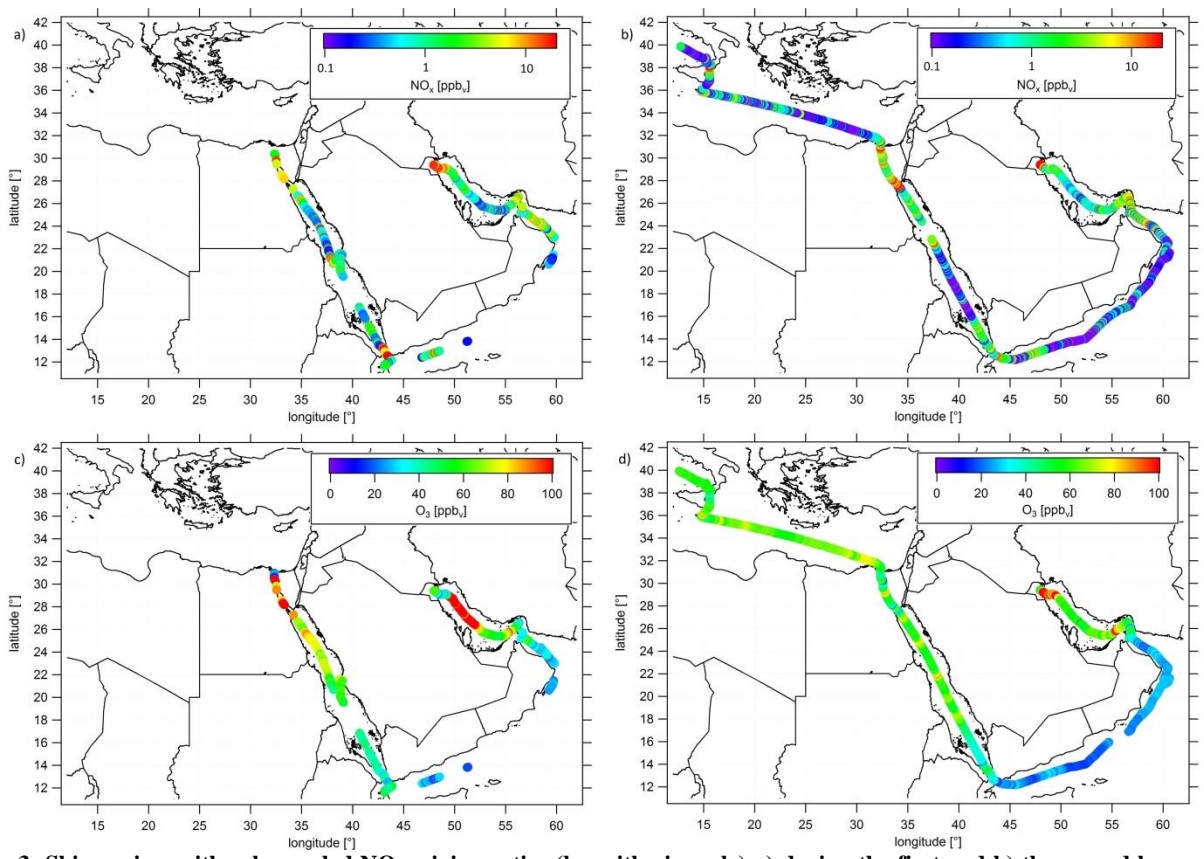

**Figure 3: Ship cruises with color-scaled NO$_x$ mixing ratios (logarithmic scale) a) during the first and b) the second leg and color-scaled O$_3$ mixing ratios (linear scale) c) during the first and d) during the second leg. Note that both NO$_x$ and O$_3$ has been filtered for own stack contamination.**



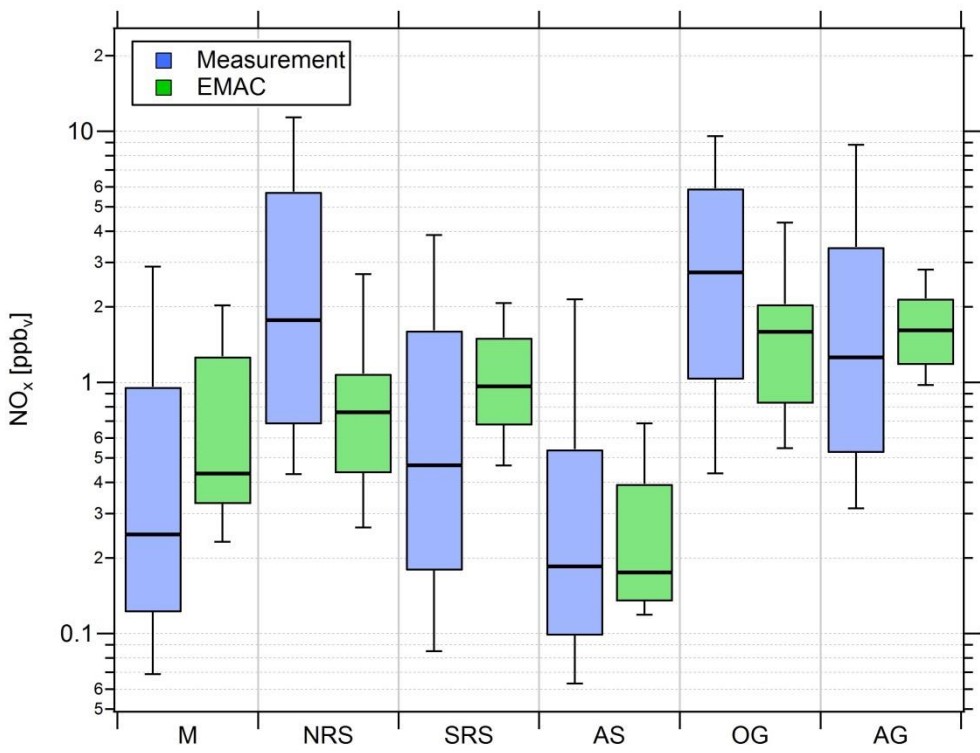


**Figure 4: Comparison of measured (blue) and simulated (green) NO$_x$ mixing ratios in the six different regions investigated during AQABA. The horizontal black bar indicates the median value, the box the 25- and 75-percentiles and the whiskers the 10- and 90-percentiles.**



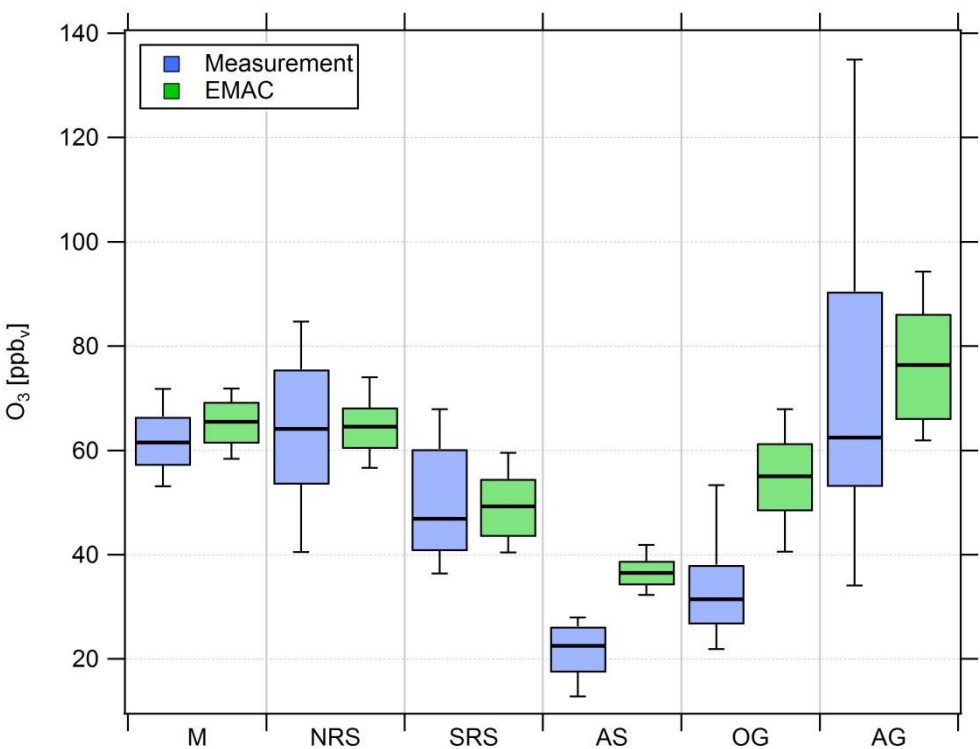

**Figure 5: Comparison of measured (blue) and simulated (green) O$_3$ mixing ratios in the six different regions investigated during AQABA. The horizontal black bar indicates the median value, the box the 25- and 75-percentiles and the whiskers the 10- and 90-percentiles.**

Overall, we find that NO$_x$ mixing ratios over the Northern Red Sea, the Gulf of Oman and the Arabian Gulf are approximately one order of magnitude higher than in the other three regions (Southern Red Sea, Arabian Sea, Mediterranean). NO$_x$ medians over the Arabian Gulf, the Northern Red Sea and the Gulf of Oman are 1.26 ppb$_v$, 1.76 ppb$_v$ and 2.74 ppb$_v$, respectively. Lower median NO$_x$ mixing ratios were measured over the Southern Red Sea (0.46 ppb$_v$), the Mediterranean (0.25 ppb$_v$) and the Arabian Sea (0.19 ppb$_v$). With respect to observed O$_3$ mixing ratios, the Arabian Sea is the only region representing remote MBL conditions with lowest median and average O$_3$ of 21.5 ppb$_v$ and 22.5 ppb$_v$ respectively, followed by the Gulf of Oman where median and mean O$_3$ were 31.5 ppb$_v$ and 34 ppb$_v$, respectively. The low O$_3$ mixing ratios over the Arabian Sea were accompanied by the smallest variability (whisker-interval: 15.1 ppb$_v$). However, a significantly larger whisker-interval of 31.4 ppb$_v$ over the Gulf of Oman indicates increasing amounts of pollution and advection from the Arabian Gulf where extreme events of ozone were observed several times during the campaign with maximum mixing ratios of up to 170 ppb$_v$. The whisker-interval over the Arabian Gulf was 100.9 ppb$_v$, more than six times higher than that over the Arabian Sea. Over the Mediterranean, the Northern Red Sea and the Southern Red Sea, median ozone was 61.5 ppb$_v$, 64.2 ppb$_v$ and 46.9 ppb$_v$, respectively. The whisker-intervals over the Northern Red Sea and the Southern Red Sea were 44.2 ppb$_v$ and 31.6 ppb$_v$, respectively. Photochemically aged air masses over the Mediterranean lead





to a rather small whisker-interval of 18.7 ppb$_v$. In summary, median NO$_x$ over the Oman Gulf was 56 % and 117 % higher

than over the Northern Red Sea and the Arabian Gulf, respectively. However, the highest NO$_x$ average was measured over the Northern Red Sea at 4.69 ppb$_v$, similar to the values observed over the Oman Gulf (4.16 ppb$_v$) and the Arabian Gulf (3.65 ppb$_v$). Note that highest NO$_x$ mixing ratios over the Oman Gulf and over the Northern Red Sea are not always associated with high O$_3$ mixing ratios. We find that average ozone was highest over the Arabian Gulf with 74 ppb$_v$ followed by the Northern Red Sea region (63.4 ppb$_v$). The average ozone mixing ratio over the Oman Gulf was 34 ppb$_v$, which corresponds to 46 % of the value observed over the Arabian Gulf. Photochemically aged air masses over the Mediterranean

Basin show an ozone average of 61.6 ppb$_v$ and air masses encountered over the Northern Red Sea (O$_3$ median of 64.2 ppb$_v$, O$_3$ average of 63.4 ppb$_v$) are comparable to the Arabian Gulf.

Due to a number of large pollution sources in the region around the Arabian Peninsula such as passing ships, highly urbanized areas as well as on- and off-shore petrochemical processing, NO$_x$ levels were rarely as low as those found in remote locations such as over the South Atlantic (Fischer et al., 2015) where NO$_x$ levels may be under 20 ppt$_v$. Apart for a

few occasions where NO$_x$ was below 50 ppt$_v$ for short periods (Arabian Sea, the Southern Red Sea and the Mediterranean), NO$_x$ levels during AQABA generally ranged from 100 ppt$_v$ up to several ppb$_v$. The campaign NO$_x$ median of 0.65 ppb$_v$ and mean value of $(2.51 \pm 5.84)$ ppb$_v$ is comparable to urban sites (Kleinman et al., 2005). A detailed emission density analysis performed by Johansson et al. (2017) shows that NO$_x$ emissions in and around the Arabian Peninsula are amongst the highest worldwide, which could explain the rather high NO$_x$ level in the MBL around the peninsula (Johansson et al., 2017;

Pfannerstill et al., 2019). O$_3$ mixing ratios measured during AQABA were also very variable with O$_3$ mixing ratios ranging between less than 20 ppb$_v$ in the remote MBL (Fischer et al., 2015) to 60-70 ppb$_v$ in the Mediterranean (consistent with previous ship-based measurements in the region (Velchev et al., 2011) and as high as 150 ppb$_v$ measured over the Arabian Gulf region. The latter are consistent with O$_3$ mixing ratios reported from regions influenced by oil and gas processing (Edwards et al., 2014; Pfannerstill et al., 2019) and narrow shipping lanes such as the Houston Ship Channel (Kleinman et

al., 2005; Zhou et al., 2014).

Figure 4 also shows that the general trend for NO$_x$ mixing ratios in the different regions is widely reproduced by the EMAC model. We find that the median NO$_x$(model)/NO$_x$(measurement)-ratio throughout the whole campaign is 0.91, indicating that the model underestimates NO$_x$ by roughly 10 %. The average ratio and its standard deviation are significantly larger at 2.57 and 5.71, respectively, indicating that single modeled data points strongly exceed the measurements, especially during

periods of low in situ NO$_x$ (see supplementary Figure S5). Particularly over the Arabian Sea and the Southern Red Sea, the model generally simulates NO$_x$ mixing ratios higher than 100 and 200 ppt$_v$, respectively while the measurements indicate mixing ratios of less than 50 ppt$_v$ for certain periods. Furthermore, as expected, the model is not able to reproduce point sources such as passing ships for which we observe a significant underestimation of the measured NO$_x$. For ozone we find that the median O$_3$(model)/O$_3$(measurement)-ratio throughout the campaign is 1.23, indicating that over the course of the





campaign the model overestimates $O_3$ by about 23 %. This could partly be related to the same limitation, i.e. the inability of the model to resolve point sources in which $O_3$ is locally reduced due to titration by NO. While the model is in rather good agreement with the measurements over the Mediterranean, the Northern Red Sea and Southern Red Sea, large deviations are found over the Arabian Sea and the Oman Gulf, where the model overestimation with respect to the regional median is 63 % and 75 %, respectively. A possible explanation for the overestimation of both ozone and $NO_x$ in pristine regions such as over

the Arabian Sea and the Oman Gulf could be related to the model resolution of 1.1° x 1.1°. Interpolation of model simulations along the *Kommandor Iona* ship track close to the coast at this resolution will most likely incorporate contributions from nearby land areas, affected by anthropogenic emissions. See supplementary Table ST2 and Table ST3 for further information and Figure S5 and S6 for additional scatterplots of measured and simulated regional median $NO_x$ and $O_3$, respectively.

**3.2 Estimation of $RO_2$ around the Arabian Peninsula**

Noontime $RO_2$ was estimated based on Eq. 3. As the steady state assumption will not hold for air masses originating from fresh emissions (times to acquire steady state estimated from the inverse sum of the loss and production terms for $NO_2$ typically ranged from 1-2 minutes during AQABA) and for fast changes in the actinic flux, we have calculated Box-Whisker-Plots for $\pm$ 2 h around noontime for which we expect relatively minor changes in the actinic flux (Fig. 6). The

noontime of each day was approximated by applying a Gaussian fit routine to the measured $j(NO_2)$ values whereas $j(NO_2)$ values being less than $10^{-3}$ s$^{-1}$ were neglected. Due to the availability of OH and $HO_2$ data from July 18$^{th}$ 2017 onwards, we have limited the analysis to this period. Note that there are no noontime $RO_2$ estimates from July 18$^{th}$ to July 21$^{st}$ due to contamination by the ship exhaust and on August 24$^{th}$ 2017 due to missing data. The black bar in Fig. 6 indicates the median value, with the Box-interval marking the 25- and 75-percentile and the whisker showing the 10- and 90-percentile. Figure 7

shows summarized regional trends of the $RO_2$ estimates for measured and simulated data.



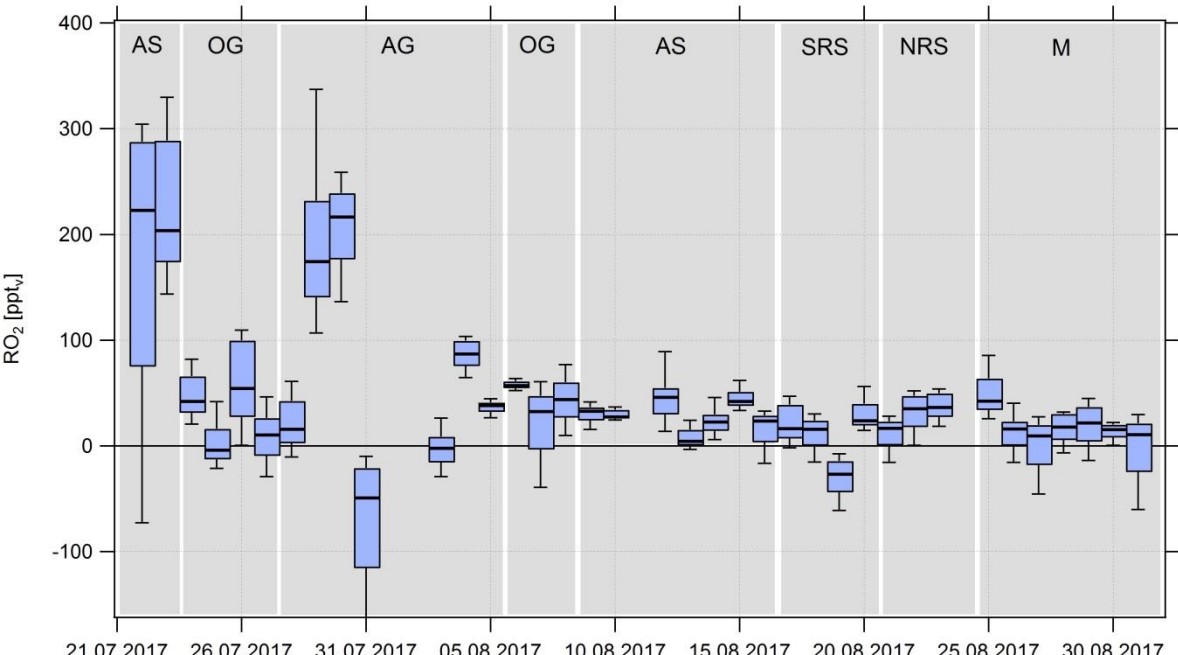

**Figure 6: Timeline of median RO$_2$ noontime estimates from July 22$^{nd}$ to August 31$^{st}$ 2017. Due to contamination by the ship exhaust itself, there is no data from the July 18$^{th}$ to July 21$^{st}$ 2017. See annotations for the classification of the different regions.**






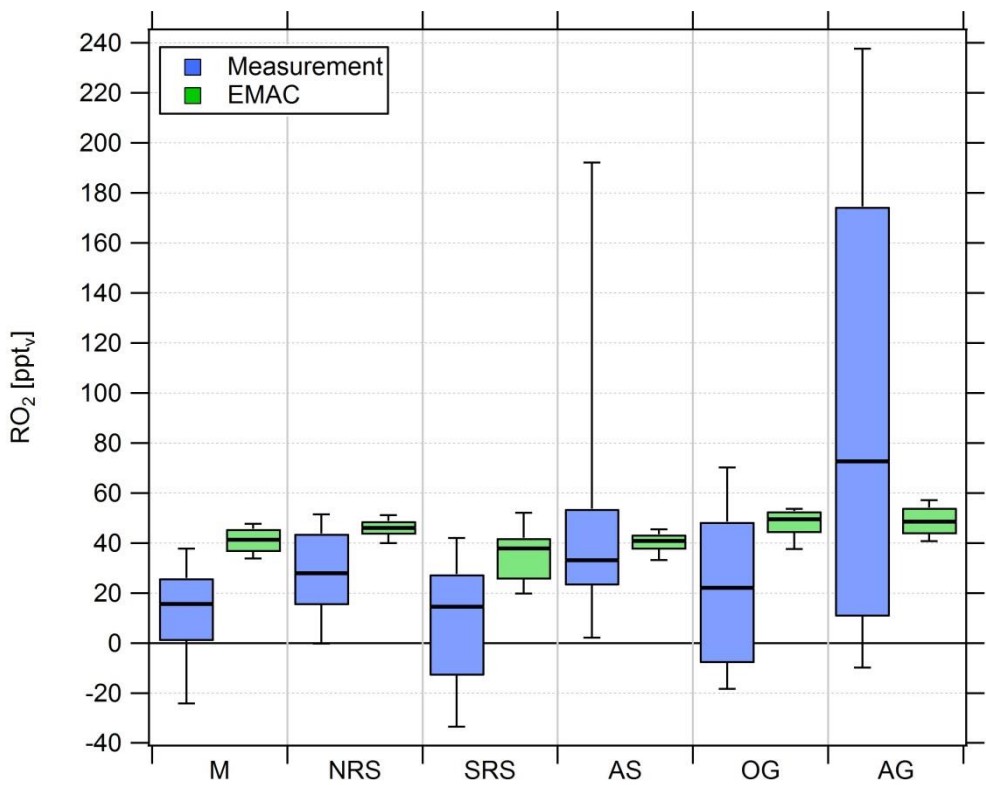

**Figure 7: Comparison of Box-Whisker-Plots of the regional estimated noontime RO₂ median based on estimated and simulated data for the period from July 18th 2017 onwards.**

We find median noontime $RO_2$ mixing ratios over the Mediterranean, the Northern Red Sea, the Southern Red Sea, the Arabian Sea and Oman Gulf of 16 ppt$_v$, 28 ppt$_v$, 15 ppt$_v$, 33 ppt$_v$ and 22 ppt$_v$, respectively, with each respective 75-percentile $RO_2$ being equal or less than 54 ppt$_v$. Based on the total measurement uncertainties of the measured quantities in Eq. 3, the uncertainty in the $RO_2$ estimates is estimated at 14 %. Only over the Arabian Gulf, the $RO_2$ estimate yields a median noontime mixing ratio of 73 ppt$_v$ accompanied by the largest variations in the box-interval of the whole campaign. While the

box-interval of the $RO_2$ estimate in the other regions is 25-57 ppt$_v$, the box-interval over the Arabian Gulf is significantly higher at 165 ppt$_v$. Negative values for all regions are regularly found in the vicinity of fresh emissions and air masses not in photochemical equilibrium. The elevated 90-percentile over the Arabian Sea is due to high $RO_2$ estimates during the first leg on July 22nd and 23rd.

Estimated $RO_2$ mixing ratios based on measured tracer data are in general agreement with previous studies performed in

marine boundary layer environments which report maximum mixing ratios between 30 and 55 ppt$_v$ around noontime (Hernandez et al., 2001). As peroxy radicals are short-lived molecules generated from the oxidation of VOCs, enhanced $RO_2$




concentrations observed over the Arabian Gulf are most likely due to high VOC emissions from intense oil and gas activities in the region (Bourtsoukidis et al., 2019; Pfannerstill et al., 2019). Bourtsoukidis et al. report that spatial volume mixing ratios of ethane and propane over the Arabian Gulf were about a factor of 10-15 times higher than over the Arabian Sea and

the Southern Red Sea (Bourtsoukidis et al., 2019). We find that the median noontime $RO_2$(estimated)/$HO_2$(measurement)-ratio throughout the whole campaign is 1.54, indicating that throughout the campaign about one third of the sum of all peroxy radicals was represented by organic peroxy radicals. Note that during single days, $HO_2$ may be higher than the $RO_2$ estimate, which is within the uncertainty of the $RO_2$ estimate.

EMAC modelled, median noontime $RO_2$ mixing ratios estimated as the sum of simulated $HO_2$ and all simulated peroxy

radicals with less than four carbon molecules are 41 $ppt_v$, 46 $ppt_v$, 38 $ppt_v$, 41 $ppt_v$, 50 $ppt_v$ and 49 $ppt_v$ over the Mediterranean, the Northern Red Sea, the Southern Red Sea, the Arabian Sea, the Oman Gulf and the Arabian Gulf, respectively. The observation based $RO_2$ estimate yields 16 $ppt_v$, 28 $ppt_v$, 15 $ppt_v$, 33 $ppt_v$, 22 $ppt_v$ and 73 $ppt_v$ respectively. We find that the median point by point $RO_2$(model)/$RO_2$(measurement)-ratio from July 18[th] onward is 1.05 so that, on average, the model overestimates the measurement by 5 %. Please note that the observational variability is much higher than

the modeled one and that the median of 1.05 is accompanied by a larger average (1.84) and a large variability (42.51). See supplementary Table ST4 and ST5 for further information and Figure S7 for an additional scatterplot of measured and simulated regional median $RO_2$.

### 3.3 Net ozone production rates around the Arabian Peninsula

In the following, net ozone production rates (at noon) are calculated based on Eq. 7 for the different regions. These noontime

values are then scaled to diurnal production rates (Fig. 8) based the integrated actinic flux as photochemical net ozone production is in good approximation linear with actinic flux. Due to contamination by the ship exhaust and due to the availability of OH and $HO_2$ data only from July 18[th] 2017 onwards, we have limited the analysis to the period from July 22[nd] 2017 to August 31[st] 2017. A comparison of NOPR estimated based on measured and simulated data for the different regions is shown in Fig. 9.





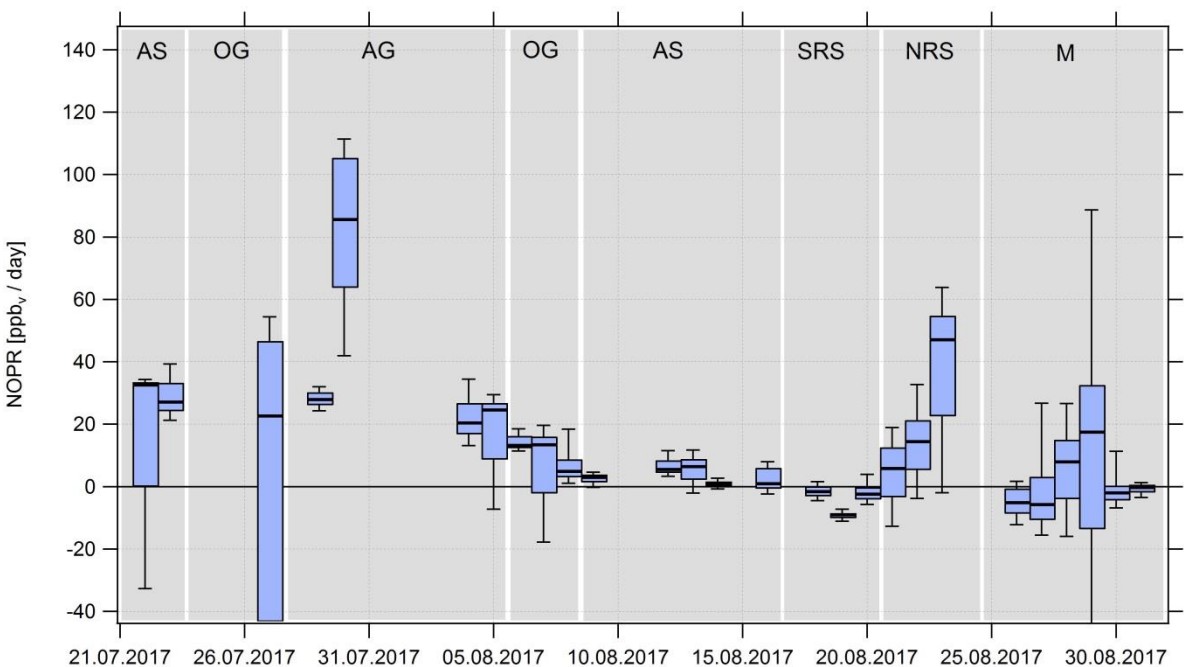


**Figure 8: Timeline of the diurnal NOPR from July 22$^{nd}$ to August 31$^{st}$ 2017. NOPR calculations are limited to the time period from July 22$^{nd}$ onwards due to missing HO$_x$ data and contamination before this period. See annotations for the classification of the different regions.**

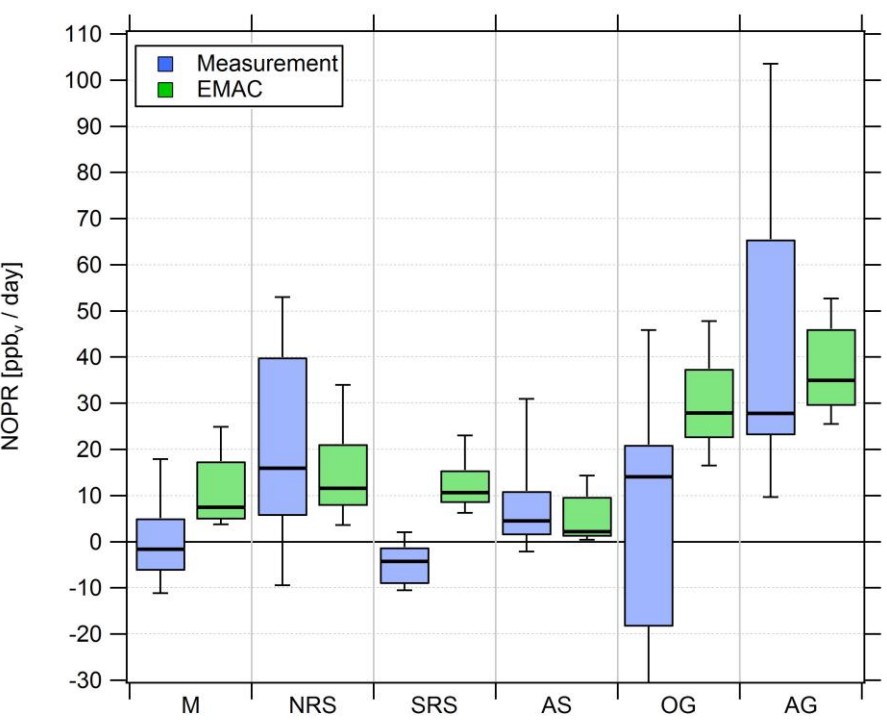






**Figure 9: Diurnal net ozone production rates in the different regions. Related to the magnitude of pollution sources, the lower whisker of the NOPR estimate over the Oman Gulf is -324 ppb day⁻¹.**

Over the Mediterranean and the Southern Red Sea, NOPR values do not significantly deviate from zero (production equals loss) within the atmospheric variability (the uncertainty of the regional NOPR is 40 % which has been estimated by error propagation). However, the best estimate indicates slight net ozone destruction for the Mediterranean and Southern Red Sea (- 2 ppb day⁻¹) and (- 4 ppb day⁻¹) respectively, and slight net production for the Arabian Sea (5 ppb day⁻¹), which is significantly positive within the variability of the box-interval. Variations in NOPR calculated as the width of the 25-75-percentile-box yield comparable values of 9-11 ppb day⁻¹ for these three regions. Substantial net ozone production was inferred over the Oman Gulf, the Northern Red Sea, and the Arabian Gulf with the respective median values being 14 ppb day⁻¹, 16 ppb day⁻¹ and 28 ppb day⁻¹, respectively. Especially over the Red Sea we find a strong latitudinal gradient in net ozone production rates with higher values towards the northern end, while slight net ozone destruction of -4 ppb day⁻¹ is reported over the southern part.

NOPR estimates for the Oman Gulf, the Northern Red Sea and the Arabian Gulf are comparable to results reported for dense traffic shipping routes such as the Houston Ship Channel with NOPR of a few tens of ppb h⁻¹ for periods of severe pollution (Zhou et al., 2014). Similar net ozone production rates have been reported for regions of Beijing in summer 2006 (Lu et al., 2010). For regions with low anthropogenic influence such as the Southern Red Sea and the Arabian Sea we estimate net ozone production that does not differ significantly from zero. This is due to the rather low $NO_x$ mixing ratios in the clean marine boundary layer (Bozem et al., 2017). Note that we calculated net ozone destruction only for a few days over the Southern Red Sea and the Arabian Sea, indicating that the marine boundary layer around the Arabian Peninsula is rarely free from anthropogenic influence owing to the multitude of on- and off-shore anthropogenic activities.

We find that model-calculated estimates of NOPR reproduce the trends observed for NOPR calculated from in situ measurements except over the Mediterranean and the Southern Red Sea. Although EMAC predicts high ozone levels over the Arabian Sea, it also reports the lowest NOPR in this region. Deviations between model-calculated estimate and the estimate based on measured tracer data over the Mediterranean and over the Southern Red Sea could be linked to $NO_x$ being overestimated in the model in these regions. As these regions have been classified as $NO_x$-limited ozone production regimes in this study, an increase in $NO_x$ is expected to increase NOPR. In the model, pollution emissions, especially over the Oman Gulf and the Arabian Gulf, seem to be averaged over a large (1.1° grid size) region. High background concentrations of ozone precursors hence contribute to net ozone production rates that compare to conditions in highly polluted marine boundary air such as in the Houston Ship Channel (Zhou et al., 2014). Even in the more pristine regions such as over the Southern Red Sea and the Arabian Sea, the model is not able to reproduce net ozone destruction, which is consistent with the fact the ozone is generally too high and that $NO_x$ levels below 0.1 $ppb_v$ are not found in the model. See supplementary Table ST6 and ST7 for further information and Figure S8 for an additional scatterplot of measured and simulated regional NOPR.





## 3.4 VOC and $NO_x$ relationship to net ozone production

Ozone is photochemically formed when the precursors $NO_x$ and VOCs are abundant in the presence of sunlight (Bozem et al., 2017; Jaffe et al., 2018). In order to determine whether a chemical system is $NO_x$- or VOC-limited or in a transition between those two regimes, one has to estimate the total amount of OH reactivity towards VOCs and towards $NO_x$. Therefore the VOC/$NO_x$-ratio is an important indicator of the behavior of $NO_x$, VOCs and $O_3$ in a system. Since it is not feasible to precisely define all ambient VOCs (could be thousands), formaldehyde mixing ratios have been used as a proxy

for the OH reactivity towards VOCs since it is a short-lived oxidation product of many VOCs that is often positively correlated with peroxy radicals (Sillman et al., 1995; Duncan et al., 2010). Sillman et al. first used afternoon concentrations of indicator species such as HCHO and total reactive nitrogen ($NO_y$) to determine the sensitivity of ozone production to VOCs or $NO_x$ (Sillman et al., 1995). Their approach was later successfully transferred to space-based satellite observations by using the ratio of tropospheric columns of HCHO and $NO_2$ to determine the sensitivity of ozone production (Martin et al.,

2004). Here we use HCHO/$NO_2$-ratios (referred to as "Ratio") deduced by Duncan et al. as indicators for the sensitivity of ozone production to $NO_x$- and VOC-limitations in megacities in the United States with large amounts of anthropogenic $NO_x$ and VOC emissions (Duncan et al., 2010). The Ratio is an indicator of surface photochemistry as most of the atmospheric column of HCHO and $NO_2$ is located in the planetary boundary layer (Duncan et al., 2010). Duncan et al. have derived $NO_x$-limited ozone production regimes for HCHO/$NO_2 > 2$ and VOC-limited ozone production for HCHO/$NO_2 < 1$ (Duncan et

al., 2010). For $1 <$ HCHO/$NO_2 < 2$ both $NO_x$ and VOC emission reductions may lead to a reduction in ozone. Figure 10 shows the Box-Whisker-Plot classification of the HCHO/$NO_2$-ratio of the different regions during noontime.





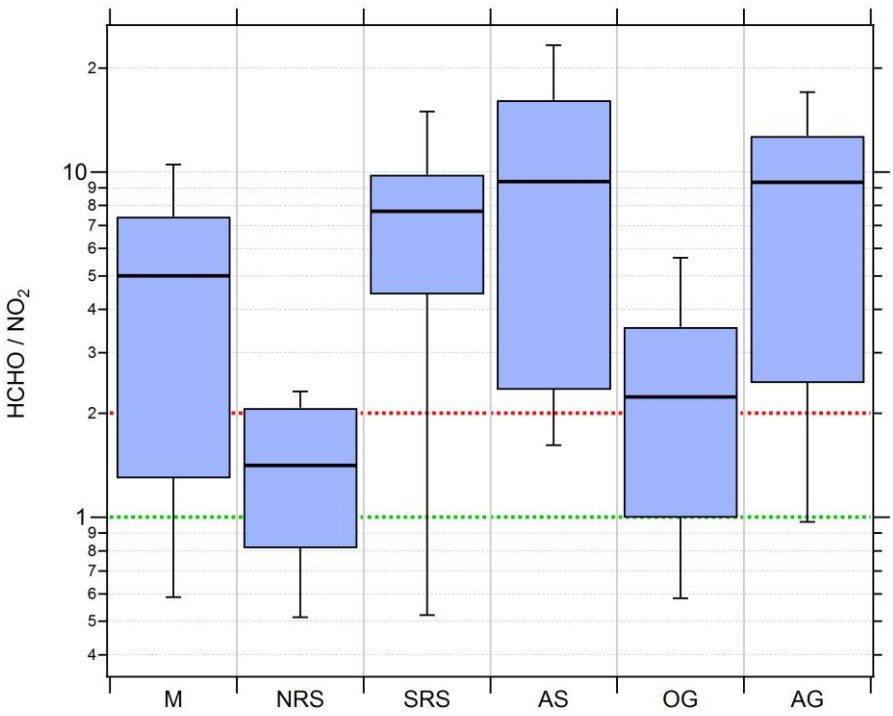

**Figure 10: Box-Whisker-Plots of the HCHO/NO₂-ratio for the different regions with the black bar indicating the median value. Red (ratio = 2) and green (ratio = 1) lines indicate the limits for HCHO/NO₂ deduced by Duncan et al. (2010) for NOₓ-limitation and VOC-limitation, respectively.**

Median HCHO/NO₂-ratios of 5, 7.7, 9.4 and 9.3 over the Mediterranean, the Southern Red Sea, the Arabian Sea and the Arabian Gulf respectively indicate tendencies towards NOₓ-limited regimes. In a previous study based on measured OH reactivity, Pfannerstill et al. classified these regions as being mostly in a transition between NOₓ- and VOC-limitation, with a tendency towards NOₓ-limitation (2019). Median HCHO/NO₂-ratios of 1.4 and 2.2 estimated over the Northern Red Sea and the Oman Gulf signify tendencies towards VOC-limitation. However, none of the medians of the six regions falls below the VOC-limit deduced by Duncan et al. (2010).

Over the Red Sea we find a latitudinal gradient in the HCHO/NO₂-ratio, similar to the gradients for NOₓ and NOPR. Due to very low NOₓ over the Southern Red Sea, O₃ production is NOₓ-limited, changing into a more VOC-limited regime over the Northern Red Sea. Photochemical aging of the air over the Mediterranean leads to a NOₓ-limited regime. NOₓ limitation is also inferred for the relatively clean Arabian Sea and the polluted Arabian Gulf atmosphere. Note that a future increase in emissions of NOₓ from increased shipping in the Arabian Gulf would thus lead to higher NOPR further increasing ozone pollution in the region (Johansson et al., 2017). See Table ST8 for detailed statistics on regional HCHO/NO₂-ratios.







## 4 Conclusion

In situ observations of NO, $NO_2$, $O_3$, HCHO, OH, $HO_2$, absolute humidity, actinic flux, temperature and pressure were carried out in the marine boundary layer around the Arabian Peninsula during the AQABA ship campaign from late June to early September 2017. Concentration ranges of both $NO_x$ and $O_3$ clearly showed anthropogenic influence in the MBL. $NO_x$

was highest over the Arabian Gulf, the Northern Red Sea and the Oman Gulf. Lowest $NO_x$ was observed over the Arabian Sea and over the Southern Red Sea during the second leg. $O_3$ mixing ratios were highest over the Arabian Gulf. We observed a latitudinal gradient in $O_3$ concentrations with higher values towards the northern part of the Red Sea. Although comparable $O_3$ averages were measured over the Northern Red Sea and over the Mediterranean, lower variability over the Mediterranean towards the end of August 2017 indicates photochemically more extensively aged air masses. The lowest regional $O_3$ mixing

ratio average was detected over the Arabian Sea, which is broadly comparable to remote marine boundary layer conditions in the Northern Hemisphere.

Noontime $RO_2$ estimates based on deviations from the Leighton Ratio yield median values around the Arabian Peninsula amount to $15-33$ $ppt_v$ for all regions except over the Arabian Gulf where the median is 73 $ppt_v$. The uncertainty due to the missing up-welling actinic flux portion is expected to be insignificant. Furthermore, we estimated noontime and diurnal

NOPR based on Eq. 6 and the integral over the actinic flux. Highest diurnal NOPR were observed over the Oman Gulf, the Northern Red Sea and the Arabian Gulf with median values of 14 $ppb_v$ $day^{-1}$, 16 $ppb_v$ $day^{-1}$ and 28 $ppb_v$ $day^{-1}$, respectively, which is in agreement with previous studies that predicted net photochemical $O_3$ formation conditions in the region. Net ozone destruction was only observed for a few days with clean conditions over the Arabian Sea and the Southern Red Sea. Based on HCHO/$NO_2$-ratios our analysis suggests tendencies towards $NO_x$-limitation over the Mediterranean, the Southern

Red Sea, the Arabian Sea and the Arabian Gulf and VOC-limitation over the Northern Red Sea and the Oman Gulf, which reproduces the trends observed by Pfannerstill et al. (2019).

$NO_x$ results from the atmospheric chemistry – general circulation model EMAC underestimate the measurement data by 10 % whereas median modeled $O_3$ overestimates the measurement by 23 %, the latter being related to limitations in model resolution in coastal proximity and near shipping lanes. Although EMAC generally reproduces regional $NO_x$ and $O_3$

medians, the scatter when comparing both data sets is large. $NO_x$ is generally too low as it does not resolve local point sources and too high for clean regions. Lowest $NO_x$ of less than 0.1 $ppb_v$ found in the in situ measurements is not reproduced by the model as emissions are averaged over a large area (1.1°). Median noontime $RO_2$ retrieved from the EMAC model are ~ 5 % higher than $RO_2$ estimates based on measurement data, however, the $RO_2$ sum deduced from EMAC is sometimes about a factor of 2 higher than the regional $RO_2$ estimate based on the Leighton Ratio and measured tracer data. NOPR

estimates based on modeled data reproduce the tendencies derived from the measurements very well. However, the model does not reproduce observed net ozone destruction along some clean parts of the ship cruise.

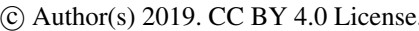


**Data availability**

Data used in this study is available to all scientists agreeing to the AQABA protocol at https://doi.org/10.5281/zenodo.3531501.

**Author contributions**

IT, HF and JL designed the study. UP and IT performed the CLD NO and $NO_2$ measurements and processed the data. JC and PE performed the $O_3$ measurements, JS performed the actinic flux measurements. JS performed cavity ring-down spectroscopy measurements of $NO_2$. DD and BH performed the HCHO measurements. HH, MM, RR, ST performed the OH and $HO_2$ measurements. J-DP was responsible for the $H_2O$ measurements. Model simulations were made by AP. All authors

have contributed to writing this manuscript

**Competing interests**

The authors declare no conflict of interest.

**Acknowledgements**

We acknowledge the collaborations with King Abdullah University of Science and Technology (KAUST), The Cyprus
Institute (CyI) and the Kuwait Institute for Scientific Research (KISR). We would like to thank Marcel Dorf and Claus Koeppel for the organization of the campaign. We would like to thank Hays Ships Ltd. and the ship crew for caring about the physical well-being of the scientific participants and for an unforgettable time on board *Kommandor Iona*. Last but not least we would like to thank the whole AQABA community for a successful campaign.




## Appendix: Acronyms and abbreviations

### General

| | |
|---|---|
| AQABA | **A**ir **Q**uality and Climate in the **A**rabian **Ba**sin campaign |
| CyI | The **Cy**prus **I**nstitute |
| KAUST | **K**ing **A**bdullah **U**niversity of **S**cience and **T**echnology |
| KISR | **K**uwait **I**nstitute for **S**cientific **R**esearch |


### Regions

| | |
|---|---|
| AG | **A**rabian **G**ulf |
| AS | **A**rabian **S**ea |
| M | **M**editerranean Sea |
| NRS | **N**orthern **R**ed **S**ea |
| OG | **O**man **G**ulf |
| SRS | **S**outhern **R**ed **S**ea |


### Scientific

| | |
|---|---|
| CLD | **C**hemi**l**uminescence **d**etector |
| CRDS | **C**avity **r**ing-**d**own **s**pectroscopy |
| ECHAM5 | Fifth generation **E**uropean **C**entre **Ham**burg general circulation model |
| EMAC | **E**CHAM/**M**ESSy **A**tmospheric **C**hemistry model |
| FWHM | **F**ull **w**idth at **h**alf **m**aximum |
| GC-FID | **G**as **c**hromatography – **f**lame **i**onization **d**etector |
| HORUS | **H**ydr**O**xyl **R**adical measurement **U**nit based on fluorescence **S**pectroscopy instrument |
| HO$_x$ | OH + HO$_2$ |
| LED | **L**ight **e**mitting **d**iode |
| LIF | **L**aser **i**nduced **f**luorescence |
| MBL | **M**arine **b**oundary **l**ayer |
| MESSy | **M**odular **E**arth **S**ubmodel **Sy**stem |
| NOPR | **N**et **o**zone **p**roduction **r**ate |
| NO$_x$ | NO + NO$_2$ |
| PFA | **Per**fluoro**a**lkoxy |
| PSS | **P**hotostationary **s**teady **s**tate |
| PTFE | **P**oly**t**etra**f**luoro**e**thylene |
| SLM | **S**tandard **l**iter per **m**inute |
| STEAM3 | **S**hip **T**raffic **E**mission **A**ssessment **M**odel 3 |
| TMU | **T**otal **m**easurement **u**ncertainty |
| VOC | **V**olatile **o**rganic **c**ompounds |
| UV | **U**ltra**v**iolet |











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
