# Peer review of "Net ozone production and its relationship to NOx and VOCs in the marine boundary layer around the Arabian Peninsula"

_Atmospheric Chemistry and Physics, 2019_

## Referee Comment (RC1) · Anonymous Referee #1 · 30 Jan 2020

The authors present an analysis of net O3 production calculated from ship-borne trace gas measurements obtained during the AQABA campaign, mostly in July and August 2017. In the Oman Gulf, the Northern Red Sea, and the Arabian Gulf the authors found the highest values ranging from 14 ppb day-1 to 28 ppb day-1. Based on HCHO/NO2 ratios, in most areas O3 formation was NOx limited apart from the Northern Red Sea, which was located in the VOC-NOx limitation transition zone. The Arabian Sea and Arabian Gulf areas showed maximum HCHO/NO2 values, clearly indicating NOx limitation. This paper shows some interesting data from an area, which often lacks robust air quality data, but which is also an area with significant (and increasing) anthropogenic emissions, mostly related to oil and gas exploration. The paper is well-written and

deserves publication. However, I was hoping the authors could address some of my concerns.

Page 1 L22-24: According to Fig 10 net O3 formation actually mostly occurs in NOx limitation regimes, not in the transition regime, as the authors mention here. Also, it seems Fig 10 does not support the findings by Pfannerstill et al. (2019) as stated by the authors on page 25 (L509-511). All, but one median, is above the threshold of HCHO/NO2 > 2. I also doubt that a HCHO/NO2 median of 2.2 (for OG) would signify a tendency towards VOC limitation (L511-512).

Page 3, L58-59: The reference Zhou et al (2014) shows up a couple of times in this paper. While I agree that it makes sense to compare the authors' Middle East study area with the Houston area (e.g. some similar emissions; similar latitude) there are better publications for the Houston case which include direct measurements of OH, HO2, and also O3 production itself among many other measurements (e.g. Chen et al., 2010; Mao et al, 2010; Ren et al. 2013). Also, I doubt that the term marine environment shows up in Zhou et al.

Page 3, L80-86: This explanation is confusing. It seems there is a main sampling inlet (not sure what instruments were connected), but for HCHO and NOx sampling was done via a 1/2" PFA tubing (not sure how long the tube was), and for the other trace gas measurements (what were those?) it was done in a different way. May be a chart showing the experimental design would be helpful.

Page 6, L144: What was the flow of the calibration gas into the zero air?

Page 6, L157: What do the authors mean by "notably high"?

Page 11, L259: I guess the authors mean Lu et al. (2010) here.

Page 12, L276-277: The loss mechanism through H2O is important. Also, it seems to vary a lot. Some parts of the ship cruising legs might have already been exposed high humidity due to the Indian monsoon system. It would be good to see the absolute

humidity variation along the legs similar to Figs 3 and S4.

Page 12, L281-282: At least, the authors want to include an estimate for the potential contribution of halogens.

Page 12, Eq. 7 and Fig 9: It would be nice to see a break-down of the different terms in Eq. 7 for different legs as shown in Fig. 9 to evaluate what processes might be most relevant/different in those different legs.

Page 13, L310: Authors mention NOx values of several hundred ppbs. Where do they show up in Figs. 3, S2, and S3? What were the megacities along the cruising legs? I could think about Cairo, but according to Fig 3 NOx values do not show extremely high values.

Page 13, L311-312: During the first leg very high O3 values are found in the Arabian Gulf and potentially in the area of the Suez Canal. In the second leg those high O3 values are pretty much gone. I doubt emissions have changed. I also doubt that weather conditions have changed drastically. What were the reasons for those distinct changes?

Page 16, L334-355: This section should include some more explanations: it seems there is a huge variation in NOx and O3 in AG (also a huge variation in NOPR as shown in Fig 9). What is the major driver of this: point sources from ships? Why are the highest NOx values in OG and why are some of the lowest O3 values found in OG? Why would you consider air masses over the Mediterranean as photochemically aged air masses due to the small whisker-interval, while the whisker-plots for AS and OG show pretty much the same with, but at much lower absolute O3 ranges. There are no emission sources in that area of the Mediterranean?

Page 17, L367: As I understand it Velchev et al. (2011) show O3 data from the Western Mediterranean. A reference looking closer to the area the authors studied would be Kouvarakis et al (2002).

Page 17, L369: Edwards et al. (2014) is not a good reference here. Edwards et al report wintertime O3 in cold-pool conditions, i.e. extremely low boundary layer heights. The meteorological conditions reported in Edwards et al are pretty different from the ones observed in the Middle East during summertime. Also, there is no word on the impact of narrow shipping lanes in Kleinman et al (2005) and Zhou et al (2014). O3 in Houston is predominantly driven by emissions from all kinds of petrochemical industries (including refineries, but no oil exploration) located in the Ship Channel area.

Page 25, L516-517: Actually, Figure 10 shows that in almost all areas O3 formation is NOx limited. However, the authors say that this is typical for photochemically aged air masses over the Mediterranean. As already mentioned further above, why do the authors explicitly consider the Mediterranean area having aged air masses? It is even more surprising as the results for the Mediterranean area in Figure 10 indicate that the Box-Whisker plot stretches into the transition between NOx and VOC limitation.

Page 25. L517-519: Why would higher NOx lead to higher O3 pollution? For instance, according to Figure 4, OG has the highest NOx values, but also pretty low O3 values. With regard to NOPR, the Box-Whisker plot for OG shows positive, but also large negative values. In any case NOPR values are significantly lower than for AG, for instance.

Table 1: This table can go into the supplement.

Figure 7: The legend mentions "Measurements", the figure captions says "estimated". From Eq 3 I understand that RO2 was neither measured nor estimated, but calculated. Also, what would be the interpretation of the negative RO2 concentrations (blue Box-Whisker plots) when calculated from Eq 3?

Figures S2 and S3: It would be nice to see the time series of OH and HO2 here as well

References:

Chen et al (2010): A comparison of chemical mechanisms based on TRAMP-2006 field data, Atmos. Environ., 44, 4116-4125, doi:10.1016/j.atmosenv.2009.05.027

Kouvarakis et al (2002): Spatial and temporal variability of tropospheric ozone (O3) in the boundary layer above the Aegean Sea (Eastern Mediterranean), J. Geophys. Res., 107, 8173, 10.1029/2000JD000081

Mao et al (2010): Atmospheric Oxidation Capacity in the Summer of Houston 2006: Comparison with Summer Measurements in other Metropolitan Studies, Atmos. Environ., 44, 4107-4115, doi:10.1016/j.atmosenv.2009.01.013

Ren et al (2013): Atmospheric Oxidation Chemistry and Ozone Production: Results from SHARP 2009 in Houston, Texas, J. Geophys. Res., 118, 5770-5780, doi:10.1002/jgrd.50342

---

## Referee Comment (RC2) · Anonymous Referee #2 · 14 Feb 2020

Review of "Net ozone production and its relationship to NOy and VOCs in the marine boundary layer around the Arabian Peninsula" by Tadic et al

The authors have a very interesting data set. I'm not familiar with the chemistry of this region but I assume there are few high quality NOx measurements and perhaps no radical measurements. Analysis is through the use of deviations from the Leighton photostationary state to get peroxy radical concentrations which are then used in conjunction with measured NO to obtain ozone production rates. Observed OH and HO2 are presented only via a color-coded ship track. As far as I can tell, the only use of the observed OH and HO2 is to determine loss rates of ozone which when added to the

PSS ozone production rate yields a net ozone production rate. The ratio of HCHO to NOx (both observed quantities) is used as an indicator ratio to predict regions in which O3 production is either NOx or VOC sensitive.

Comparisons are made with a Eulerian model for NOx, ozone, and RO2 mixing ratios as well as for net O3 production rate. This article contains the quantum of information to warrant publication. I can only guess that either they want to write up the radical and HCHO measurements in a separate study or that they are uninterruptable for one reason or another.

I found the article difficult to follow in places. It is my belief that the authors have under-estimated the uncertainty of the instruments used to determine peroxy radicals. The authors need to re-examine their error propagation formula. I am calling this a major revision as it affects the most prominent results in the paper. In practice it can be done in an afternoon. They could also compare the PSS value of RO2 with measured HO2 (which is stated to be preliminary due to an up to 20% interference by RO2). This is a reality test.

1.  line 124 "UV-induced positive bias in the NO2 measurements due to photolysis of HONO, BrONO2, NO3 and ClNO2 to produce NO was characterized ahead of the campaign to be 7.7 %, 7.2 %, 5.6 % and 1.5 % of the respective ambient concentration of HONO, BrONO2, NO3 and ClNO2 respectively,"

(Italics mine). No where in the paper is it mentioned that these species are measured. Have interferences ben determined based on model-calculated or typical concentrations? Or do these figures represent the percent interference if the interferent has the same concentration as NO?

2. line 181 NO2 was further measured by cavity ring-down spectroscopy (Sobanski et al., 2016) and used for correcting the instrumental background of the CLD NO2 data, as described above (the correction was taken as the ultimate absolute measurement uncertainty in the CLD NO2 data).

The uncertainty of the NO2 chemiluminescent measurement is not equal to that of the cavity ring-down instrument. The chemiluminescent NO2 is the difference between two measurements, one of which gets divided by 0.294 to take into account photolysis efficiency. In order to add errors in quadrature, I need to know the NO to NO2 ratio. I'm guessing that the relative uncertainty of NO2 will be at least twice that of NO. The NO2 chemiluminescent measurement is affected adversely by the relatively low photolysis efficiency. That accounts for random errors in NO2. Comparison with the cavity ring-down may take care of the NO2 instrumental background but how does it take care of the random errors?

Line 205 Total Measurement Uncertainty. All of these numbers appear very optimistic. In a previous comment, I gave my reasons why the TMU of NO2 appeared low. I do not know why the TMU of HO2 is not larger than OH. There are many sources of uncertainty in the conversion of HO2 to OH which is the quantity actually measured. The authors state a 20% bias due to RO2 chemistry. I do not know to what extent quantification of HO2 is made easier than that of OH because there is more HO2 than OH and hence a larger signal.

3. Line 272 "In low NOx environments (< 100 pptv) previous studies have indicated that further NO oxidizing trace gases such as peroxy radicals (HO2, RO2) and halogen monoxides (XO) may result in a deviation from unity (Nakamura et al., 2003; Hosaynali Beygi et al., 2011; Reed et al., 2016)."

Deviations from the Leighton relation can also be important at higher NO. Departures of the Leighton ratio from unity depend (primarily) on the competition between HO2+NO and O3+NO. In polluted environments, HO2 concentrations can increase, remain steady, or decrease only slowly as NO is increased.

Line 250 and following. Does Fig. S1 show the ratio of actinic flux in the 4 hour window centered around noon to the total measured actinic flux? Or is Fig. S1 a ratio obtained by fitting a Gaussian, between zeroes in the AM and PM?

[Figure]

4. Lines 266-270, typos for O singlet D. Elsewhere, the "1" is in its proper place as a superscript.

5. Line 285. Am I correct that insofar as ozone production is concerned the only use of the HO2 and OH measurements is their contribution to the loss rate of ozone (and hence their effect on net ozone production)?

6. Line 373. "We find that the median NOx(model)/NOx(measurement)-ratio throughout the whole campaign is 0.91, indicating that the model underestimates NOx by roughly 10 %."

The median of what? Could you please specify what items you are taking the median of; i.e., what are the data points. Ratios by Region? Days? Individual data points. I may have missed it; how long are data points? Equal to the 5 minute instrument averaging time?

7. Line 391. "Noontime RO2 was estimated based on Eq. 3. As the steady state assumption will not hold for air masses originating from fresh emissions (times to acquire steady state estimated from the inverse sum of the loss and production terms for NO2 typically ranged from 1-2 minutes during AQABA) and for fast changes in the actinic flux,"

What I think you want to say is mis-stated. As written, it says: We can't use samples that had fresh emissions, so we used samples taken when actinic flux was slowly changing. Was the data screened to eliminate time periods in which NOx (or less likely O3) was rapidly varying? From the looks of the actinic flux plot you did not have many clouds giving rapid variations in jNO2. I would be surprised if the time window around noon could not have been wider. How much does jNO2 change between, say noon-3 hours and noon – 175 minutes and what change in HO2 does that produce?

8. Figure 6. Obtaining peroxy radical concentrations from photostationary state calculations is not easy.

Line 413 "the total uncertainty in the RO2 estimates is estimated at 14%. This is a way too optimistic estimate of the uncertainty of RO2. Instrument precision is too high and no account (except for NO2) is taken of biases. Even still, the PSS is a difference in two numbers, often of comparable magnitude. I'm not certain the error propagation was correctly performed. I would like to see the formula that represents "errors added in quadrature". Compare those results with a simple Monte Carlo calculation that can be done on a spreadsheet. And keep in mind that the result will not take into account errors in rate constants.

9. Line 416. "Negative values for all regions are regularly found in the vicinity of fresh emissions and air masses not in photochemical equilibrium"

That might be the explanation. It would useful to quantify this point. HO2 and RO2 concentrations will be low in a high NOx environment. The negative values may reflect the measurement accuracy needed to distinguish, for example, – 5 ppt RO2 from zero.

10. Figure 7. Here the RO2 data looks much better. There are differences mentioned in the text. but for this reader could you please provide a concise reason why Fig. 7. looks so much better than Fig. 6. Is it the data groupings? I assume that in both Fig. 6 and 7. the blue RO2 data is from Eq. 3. I am not totally positive because you were measuring HO2 and some fraction of RO2.

11. Line 421. As peroxy radicals are short-lived molecules generated from the oxidation of VOCs, enhanced RO2 concentrations observed over the Arabian Gulf are most likely due to high VOC observed over the Arabian Gulf are most likely due to high VOC emissions from intense oil and gas activities in the region.

High HO2 can also occur in aged air masses in which NOx and VOCs have reacted away but still have significant O3 and (perhaps) HCHO. Photolysis could then yield peroxy radicals.

12. Line 440. Regarding the extrapolated actinic flux curve to get a daily ozone pro-

duction. j(1OD) decreases early and late in the day faster than J(NO2). Not sure how much difference it makes. I would be very leery of this extrapolation over land; I'm assuming that you are far enough away that varying traffic and boundary layer heights are not a concern.

13. Line 455 "the uncertainty of the regional NOPR is 40 % which has been estimated by error propagation"

I don't disagree with this value. Merely surprised at its magnitude compared to a 15% uncertainty for RO2 from Eq. 3. . 14. Line 473. "Although EMAC predicts high ozone levels over the Arabian Sea, it also reports the lowest NOPR in this region. Deviations between model-calculated estimate and the estimate based on measured tracer data over the Mediterranean and over the Southern Red Sea could be linked to NOx being overestimated in the model in these regions."

I'm not following. There is a low net ozone production rate. which to me implies that the model has a too low NO concentration, but you say that the model overpredicts NOx.

15. Section 3.4 VOC and NOx sensitivity. Makes sense.

---

## Author Response (AR1)

**Reply to Reviewer Report 1**

*In the following the comments of the reviewer are presented (black) alongside with our replies (in blue) and changes made to the manuscript (in red).*

General statement: The authors present an analysis of net O3 production calculated from ship-borne trace gas measurements obtained during the AQABA campaign, mostly in July and August 2017. In the Oman Gulf, the Northern Red Sea, and the Arabian Gulf the authors found the highest values ranging from 14 ppb day-1 to 28 ppb day-1. Based on HCHO/NO2 ratios, in most areas O3 formation was NOx limited apart from the Northern Red Sea, which was located in the VOC-NOx limitation transition zone. The Arabian Sea and Arabian Gulf areas showed maximum HCHO/NO2 values, clearly indicating NOx limitation. This paper shows some interesting data from an area, which often lacks robust air quality data, but which is also an area with significant (and increasing) anthropogenic emissions, mostly related to oil and gas exploration. The paper is well-written and deserves publication. However, I was hoping the authors could address some of my concerns.

Dear reviewer, thank you very much for reviewing our manuscript and for the insightful comments. Below we provide detailed responses to your comments. Please find revised graphs at the end of the document, which were compiled based on your comments.

Comment 1: Page 1 L22-24: According to Fig 10 net O3 formation actually mostly occurs in NOx limitation regimes, not in the transition regime, as the authors mention here. Also, it seems Fig 10 does not support the findings by Pfannerstill et al. (2019) as stated by the authors on page 25 (L509-511). All, but one median, is above the threshold of HCHO/NO2 > 2. I also doubt that a HCHO/NO2 median of 2.2 (for OG) would signify a tendency towards VOC limitation (L511-512).

Indeed a median HCHO/$NO_2$-ratio of 2.2 does not fall within plain VOC limitation as deduced by Duncan et al. (2010), which needs to be re-written. Although Pfannerstill et al. (2019) highlight the scatter in the attribution of ozone production to $NO_x$- and VOC-limitation, their study shows that ozone production over the Arabian Sea over the Arabian Gulf is rather $NO_x$-limited, whereas the Gulf of Suez is characterized by a strong VOC-limitation (page 11516 and page 11517, Figure 9 [Pfannerstill et al., 2019]). We have revised Page 1 L21-24 to: Constrained by HCHO/$NO_2$-ratios, our photochemistry calculations show that net ozone production in the MBL around the Arabian Peninsula occurs mostly in $NO_x$-limitation regimes with a significant share of ozone production occurring in the tranisition regime between $NO_x$- and VOC-limitation over the Mediterranean and more significantly over the Northern Red Sea and Oman Gulf.

Comment 2: Page 3, L58-59: The reference Zhou et al (2014) shows up a couple of times in this paper. While I agree that it makes sense to compare the authors' Middle East study area with the Houston area (e.g. some similar emissions; similar latitude) there are better publications for the Houston case which include direct measurements of OH, HO2, and also O3 production itself among many other measurements (e.g. Chen et al., 2010; Mao et al, 2010; Ren et al. 2013).Also, I doubt that the term marine environment shows up in Zhou et al.

We have removed the term "marine environment" in the context of Zhou et al. (2014) and we have revised the manuscript as suggested on Page 2, L55-56 to: Measurements performed in the Houston Ship Channel revealed NOPR of the order of several tens of ppb h$^{-1}$ (Chen et al., 2010; Mao et al., 2010; Ren et al., 2013).

Comment 3: Page 3, L80-86: This explanation is confusing. It seems there is a main sampling inlet (not sure what instruments were connected), but for HCHO and NOx sampling was done via a 1/2" PFA tubing (not sure how long the tube was), and for the other trace gas measurements (what were those?) it was done in a different way. May be a chart showing the experimental design would be helpful.

Please note that air was drawn from the stainless steel common inlet into each measurement container via bypass systems. $NO_x$ (CLD measurements) and HCHO measurements were located in one lab container (both sampling air from the same bypass) and $NO_2$ (CRDS measurements) and $O_3$ in another lab container. We have revised Page 3, L78-84 to: A 6 m high, 20 cm diameter cylindrical stainless steel common inlet was installed on the front deck of the vessel to sample air at a total mass flow rate of 10,000 SLM. NO and $NO_2$ chemiluminescence measurements were obtained at a total bypass flow rate of 28.5 SLM sampling air from the common inlet with a residence time in the tubing of ~3 s. HCHO, $NO_2$ cavity ring-down spectroscopy and $O_3$ measurements were obtained with similar bypass systems sampling air from the common inlet. $H_2O$ vapor was measured on the top of the ship mast in the front.

Comment 4: Page 6, L144: What was the flow of the calibration gas into the zero air?
The NO calibration flow into the zero air was 4.5 sccm. We have revised the manuscript. Now it says on Page 6, L 140-142: Zero air measurements and NO calibrations were performed with a total mass flow of 3.44 SLM achieving an overflow of 0.44 SLM to guarantee ambient air free standard measurements. The calibration gas was added at 4.5 sccm to the zero air flow.

Comment 5: Page 6, L157: What do the authors mean by "notably high"?
Alkenes can react with ozone to produce a chemiluminescent signal which will bias NO measurement obtained by chemiluminescence. To subtract such interferences, the CLD has been equipped with a prechamber to which the sampled air can be directed during prechamber measurements. As the reaction of ozone with alkenes will be much slower than the reaction of NO and with ozone, the CLD will be measuring only the signal from the reaction of alkenes during prechamber measurements. It should be noted that the signal from the reaction of alkenes with ozone observed during a prechamber measurement will also slightly decrease. Regions where alkenes are strongly varying in time and magnitude might be plagued by a measurement offset. We have revised Page 6, L148-150: However, in regions where alkene concentrations are strongly varying in time and magnitude, the CLD is prone to enhanced backgrounds due to the interference of alkenes with ozone in the instrument.

Comment 6: Page 11, L259: I guess the authors mean Lu et al. (2010) here.
Thank you very much for noticing this typo. Now it says Lu et al. (2010) instead of Lu et al. (2016) on page 11, L260.

Comment 7: Page 12, L276-277: The loss mechanism through H2O is important. Also, it seems to vary a lot. Some parts of the ship cruising legs might have already been exposed high humidity due to the Indian monsoon system. It would be good to see the absolute humidity variation along the legs similar to Figs 3 and S4.
Figures showing the absolute variation along both legs have been added to the supplement. On P13 L328 now it says: Supplementary Figure S5 shows the variation of the absolute humidity around the Arabian Peninsula.

Comment 8: Page 12, L281-282: At least, the authors want to include an estimate for the potential contribution of halogens.
Based on oxidative pairs, Bourtsoukidis et al. found that the majority of the samples they collected during AQABA were characterized by a OH/Cl-ratio of ~ 200:1 (Bourtsoukidis et al., 2019, Non-methane hydrocarbon ($C_2$-$C_8$) sources and sinks around the Arabian Peninsula, doi:10.5194/acp-19-7209-2019). Measured daytime OH concentrations were of the order of $5 \cdot 10^6$ molec cm$^{-3}$, hence the Cl radical concentration can be estimated at $2.5 \cdot 10^4$ molec cm$^{-3}$. Incorporating an ozone loss due to the reaction of $O_3$ with Cl (at Cl concentrations of $2.5 \cdot 10^4$ molec cm$^{-3}$) into the NOPR (Eq. 7) would decrease the diurnal net ozone production rates by roughly 0.2

ppb$_v$ day$^{-1}$ over the Arabian Sea and at most 0.6 ppb$_v$ day$^{-1}$ over the other regions. We have revised the manuscript as follows on page 12, L285-289: Based on oxidative pairs, Bourtsoukidis et al. (2019) have classified the majority of their samples collected during AQABA by an OH/Cl-ratio of 200:1. As measured daytime OH concentrations were of the order of 5·10$^6$ molecule cm$^{-3}$, the estimate would yield a Cl concentration of 2.5·10$^4$ molecule cm$^{-3}$, which would decrease the estimated diurnal net ozone production rates by roughly 0.2 ppb$_v$ day$^{-1}$ over the Arabian Sea and at most 0.6 ppb$_v$ day$^{-1}$ over the other regions, which does not substantially alter the here presented findings. The noontime chemical ozone loss rate can be summarized by

Comment 9: Page 12, Eq. 7 and Fig 9: It would be nice to see a break-down of the different terms in Eq. 7 for different legs as shown in Fig. 9 to evaluate what processes might be most relevant/different in those different legs.

We have included regional box-whisker plots (according to Figure 9) of the four terms of Eq. 7 in the supplements. We have revised the manuscript on Page 21, L460-461: A break-down of the different terms of Eq. 7 in the six regions is included in the supplementary Figures S10 – S13.

Comment 10: Page 13, L310: Authors mention NOx values of several hundred ppbs. Where do they show up in Figs. 3, S2, and S3? What were the megacities along the cruising legs? I could think about Cairo, but according to Fig 3 NOx values do not show extremely high values.

Values of up to several hundred of ppb$_v$ NO$_x$ were observed when measuring own ship stack, stack of bypassing ships or when being at anchor in the direct vicinity of Jeddah or Kuwait City. Data measured during these time periods have been removed from the final data set as contamination by the ship exhaust itself could not be excluded for these period. Megacities along the cruising legs generally include Cairo, Kuwait City, megacities in the UAE and Jeddah (Saudi Arabia).

Comment 11: Page 13, L311-312: During the first leg very high O3 values are found in the Arabian Gulf and potentially in the area of the Suez Canal. In the second leg those high O3 values are pretty much gone. I doubt emissions have changed. I also doubt that weather conditions have changed drastically. What were the reasons for those distinct changes?

Based on back trajectories, previous studies (Pfannerstill et al., 2019) also already highlighted different air mass origins during the two legs for the air sampled e.g. over the Arabian Gulf. While during the first leg northwestern wind from Kuwait/Iraq was encountered, northeastern winds from Iran were encountered during the second leg over the Arabian Gulf. For the area of Suez, data coverage during the first leg (due to instrumental mal function and applied stack filter) is not as exhaustive as during the second leg. We would classify these changes of concentrations in the Suez region as insignificant (see also Figure S2 and the data coverage for the Suez region before July 06$^{th}$ 2017). We have revised the manuscript on Page 16, L351-355 as follows: However, a significantly larger whisker-interval of observed ozone of 31.4 ppb$_v$ over the Gulf of Oman indicates increasing amounts of pollution and advection from the Arabian Gulf where extreme events of ozone were observed several times during the campaign with maximum mixing ratios of up to 170 ppb$_v$ when wind was coming from Kuwait/Iraq. Please note that during the second leg wind was coming from Iran (Pfannerstill et al., 2019).

Comment 12: Page 16, L334-355: This section should include some more explanations: it seems there is a huge variation in NOx and O3 in AG (also a huge variation in NOPR as shown in Fig 9). What is the major driver of this: point sources from ships? Why are the highest NOx values in OG and why are some of the lowest O3 values found in OG? Why would you consider air masses over the Mediterranean as photochemically aged air masses due to the small whisker-interval, while the whisker-plots for AS and OG show pretty much the same with, but at much lower absolute O3 ranges. There are no emission sources in that area of the Mediterranean?

Reasons for large variations of $NO_x$ and $O_3$ over the AG are point sources (ship, oil and gas processing) as well as a change in the general wind direction observed during both legs. $NO_x$ are high over the OG due to the magnitude of emissions from vessels. $O_3$ is generally very low over the OG because it has been partly characterized as VOC-limited and high $NO_x$ values may contribute to net ozone destruction. Air over the Mediterranean has previously been characterized as photochemically processed emissions from Eastern Europe (Turkey, Greece) (Destroff et al., 2017; Pfannerstill et al., 2019). We have included Destroff et al., "Volatile organic compounds (VOCs) in photochemically aged air from the eastern and western Mediterranean, 2017, doi:10.5194/acp-17-9547-2017 as a reference. Page 16, L348-364 following now say (please note that only the underlined sentences have been changed): **The low $O_3$ mixing ratios over the Arabian Sea was accompanied by the smallest variability (whisker-interval: 15.1 $ppb_v$). Although observing highest $NO_x$ over the Oman Gulf, $O_3$ observed over the Oman Gulf was amongst the lowest detected throughout the whole campaign, which can be partly explained by the fact that high $NO_x$ lead to low ozone production or even net ozone destruction. However, a significantly larger whisker-interval of observed ozone of 31.4 $ppb_v$ over the Gulf of Oman indicates increasing amounts of pollution and advection from the Arabian Gulf where extreme events of ozone were observed several times during the campaign with maximum mixing ratios of up to 170 $ppb_v$ when wind was coming from Kuwait/Iraq. Please note that during the second leg wind was coming from Iran (Pfannerstill et al., 2019).** The whisker-interval over the Arabian Gulf was 100.9 $ppb_v$, more than six times higher than that over the Arabian Sea. **Reasons for large variations of both $NO_x$ and $O_3$ over the Arabian Gulf were a multitude of point sources as well as a change in the observed wind direction with air masses coming from Iraq/Kuwait area during the first leg and air masses coming from Iran during the second leg (Pfannerstill et al., 2019).** Over the Mediterranean, the Northern Red Sea and the Southern Red Sea, median ozone was 61.5 $ppb_v$, 64.2 $ppb_v$ and 46.9 $ppb_v$, respectively. The whisker-interval over the Northern Red Sea and the Southern Red Sea were 44.2 $ppb_v$ and 31.6 $ppb_v$, respectively. **Air masses over the Mediterranean were characterized as photochemically aged due to their impact by northerly winds (Etesians) which bring processed/oxidated air from eastern Europe (Turkey, Greece) to the Mediterranean area (Derstroff et al., 2017; Pfannerstill et al., 2019). This photochemical ageing/oxidation over the Mediterranean leads to a rather small whisker-interval of 18.7 $ppb_v$ in ozone.**

Comment 13: Page 17, L367: As I understand it Velchev et al. (2011) show O3 data from the Western Mediterranean. A reference looking closer to the area the authors studied would be Kouvarakis et al (2002).
We have now replaced (Velchev et al., 2011) by (Kouvarakis et al., 2002) on page 17, L382.

Comment 14: Page 17, L369: Edwards et al. (2014) is not a good reference here. Edwards et al report wintertime O3 in cold-pool conditions, i.e. extremely low boundary layer heights. The meteorological conditions reported in Edwards et al are pretty different from the ones observed in the Middle East during summertime. Also, there is no word on the impact of narrow shipping lanes in Kleinman et al (2005) and Zhou et al (2014). O3 in Houston is predominantly driven by emissions from all kinds of petrochemical industries (including refineries, but no oil exploration) located in the Ship Channel area.
We agree that Edwards et al., 2014 is not the best reference in here, which has been removed in this context. Also Kleinman et al. (2005) and Zhou et al. (2014) have been removed in the context of shipping lanes. Instead we have included Mazzuca et al., 2016 "Ozone production and its sensitivity to $NO_x$ and VOCs: results from the DISCOVER-AQ field experiment, Houston 2013", doi:10.5194/acp-16-14463-2016. On page 17, L382-384 now it says: The latter are consistent with $O_3$ mixing ratios reported from regions influenced by oil and gas processing (Pfannerstill et al., 2019) and shipping lanes such as the Houston Ship Channel (Mazzuca et al., 2016).

Comment 15: Page 25, L516-517: Actually, Figure 10 shows that in almost all areas O3 formation is NOx limited. However, the authors say that this is typical for photochemically aged air masses over the Mediterranean. As

already mentioned further above, why do the authors explicitly consider the Mediterranean area having aged air masses? It is even more surprising as the results for the Mediterranean area in Figure 10 indicate that the Box-Whisker plot stretches into the transition between NOx and VOC limitation.

The Mediterranean area is considered having aged air masses as it is impacted by processed emissions from eastern Europe (Destroff et al., 2017; Pfannerstill et al., 2019). We have removed the statement that photochemical ageing of air masses over the Mediterranean leads to $NO_x$-limitation, however it should be noted that on 29 August 2017 we were lying at anchor in front of Malta with a magnitude of pollution sources nearby. This day is characterized by a low HCHO/$NO_2$-ratio and explains why the Box-Whisker-Plot for the Mediterranean stretches into the transition between $NO_x$- and VOC-limitation. Page 25, L534-538 now says: Ozone production over the Mediterranean was classified as rather $NO_x$-limited, however partly being in the transition regime between $NO_x$- and VOC-limitation, which can be explained by measurements obtained on 29 August 2017 when laying at anchor in front of Malta with a multitude of ($NO_x$)-emissions from nearby situated vessels. Average noontime $NO_x$ on that particular day was about three times as large as the regional average noontime $NO_x$ observed over the whole Mediterranean area.

Comment 16: Page 25. L517-519: Why would higher NOx lead to higher O3 pollution? For instance, according to Figure 4, OG has the highest NOx values, but also pretty low O3 values. With regard to NOPR, the Box-Whisker plot for OG shows positive, but also large negative values. In any case NOPR values are significantly lower than for AG, for instance.

Indeed, the passage needs more characterization. We have re-written the passage. Now it says on P25 L539-544: Note that a further increase in $NO_x$-emissions from increased shipping in the Arabian Gulf may initially lead to higher ozone production. However, a further increase in $NO_x$ might eventually lead to a change from $NO_x$- to VOC-sensitivity and a decrease in ozone production for this region, as observed for the Oman Gulf (median HCHO/$NO_2$-ratio of 2.2 and average $O_3$ of 34 $ppb_v$). See supplementary Table ST9 for detailed statistics on regional HCHO/$NO_2$-ratios.

Comment 17: Table 1: This table can go into the supplement.

We have moved the table into the supplement and numbered the tables in the manuscript and supplement accordingly. In the manuscript (Line 99-101) now it says: See supplementary Table ST1 for the range of latitudinal and longitudinal coordinates of the different regions and supplementary Table ST2 for a detailed day to day description of the route.

Comment 18: Figure 7: The legend mentions "Measurements", the figure captions says "estimated". From Eq 3 I understand that RO2 was neither measured nor estimated, but calculated. Also, what would be the interpretation of the negative RO2 concentrations (blue Box- Whisker plots) when calculated from Eq 3?

The legends (Figure 7) now says Estimated based on measured data and EMAC. Also the legend for Figure 9 now says Estimated based on measured data and Estimated based on simulated data.

Comment 19: Figures S2 and S3: It would be nice to see the time series of OH and HO2 here as well.

We have added the time series of OH and preliminary $HO_2$ to Figures S2 and S3.

[Figure]

**Figure S2: Timeline of NO, NO₂ (both CLD), O₃, OH, HO₂ preliminary and $j$(NO₂) data during the first leg. See Table ST1 for additional information on the ship cruise. Note that HO₂ data are prelimary.**

[Figure]

245

**Figure S3: Timeline of NO, NO₂ (both CLD), O₃, OH, HO₂ preliminary and $j$(NO₂) data during the second leg. See Table ST1 for additional information on the ship cruise. Note that HO₂ data are preliminary.**

[Figure]

250   **Figure S5: Ship cruises with color-scaled absolute humidity a) during the first and b) the second leg.**

[Figure]

**Figure S10: Comparison of the regional, absolute contribution of $k_{NO+HO_2}[NO][RO_2]$ to NOPR in the six different regions investigated during AQABA. The horizontal black bar indicates the median value, the box the 25- and 75-percentiles and the whiskers the 10- and 90-percentiles.**

[Figure]

**Figure S11: Comparison of the regional, absolute contribution of** $-j(O^1D) \cdot \alpha \cdot [O_3]$ **to NOPR in the six different regions investigated during AQABA. The horizontal black bar indicates the median value, the box the 25- and 75-percentiles and the whiskers the 10- and 90-percentiles.**

[Figure]

**Figure S12: Comparison of the regional, absolute contribution of $-k_{HO_2+O_3}[HO_2][O_3]$ to NOPR in the six different regions investigated during AQABA. The horizontal black bar indicates the median value, the box the 25- and 75-percentiles and the whiskers the 10- and 90-percentiles.**

270

[Figure]

**Figure S13: Comparison of the regional, absolute contribution of $-k_{OH+O_3}[OH][O_3]$ to NOPR in the six different regions investigated during AQABA. The horizontal black bar indicates the median value, the box the 25- and 75-percentiles and the whiskers the 10- and 90-percentiles.**

275

**Reply to Reviewer Report 2**

*In the following the comments of the reviewer are presented (black) alongside with our replies (in blue) and changes made to the manuscript (in red).*

280 General statement: The authors have a very interesting data set. I'm not familiar with the chemistry of this region but I assume there are few high quality NOx measurements and perhaps no radical measurements. Analysis is through the use of deviations from the Leighton
photostationary state to get peroxy radical concentrations which are then used in conjunction
with measured NO to obtain ozone production rates. Observed OH and HO2 are presented only via a color-
285 coded ship track. As far as I can tell, the only use of the observed OH and HO2 is to determine loss rates of ozone which when added to the PSS ozone production rate yields a net ozone production rate. The ratio of HCHO to NOx (both observed quantities) is used as an indicator ratio to predict regions in which O3 production is either NOx or VOC sensitive.

290 Comparisons are made with a Eulerian model for NOx, ozone, and RO2 mixing ratios as well as for net O3 production rate. This article contains the quantum of information to warrant publication. I can only guess that either they want to write up the radical and HCHO measurements in a separate study or that they are uninterruptable for one reason or another.

295 I found the article difficult to follow in places. It is my belief that the authors have underestimated the uncertainty of the instruments used to determine peroxy radicals. The authors need to re-examine their error propagation formula. I am calling this a major revision as it affects the most prominent results in the paper. In practice it can be done in an afternoon. They could also compare the PSS value of RO2 with measured HO2 (which is stated to be preliminary due to an up to 20% interference by RO2). This is a reality test.
300 Dear reviewer, thank you very much for reviewing our manuscript and for the insightful comments. Below we provide detailed responses to your comments. Please note that PSS RO$_2$ has been compared to HO$_2$ in Sect. 3.2, P 21 L442.

305 Comment 1: line 124 "UV-induced positive bias in the NO2 measurements due to photolysis of HONO, BrONO2, NO3 and ClNO2 to produce NO was characterized ahead of the campaign to be 7.7 %, 7.2 %, 5.6 % and 1.5 % of the respective ambient concentration of HONO, BrONO2, NO3 and ClNO2 respectively,"
(Italics mine). No where in the paper is it mentioned that these species are measured. Have interferences been determined based on model-calculated or typical concentrations? Or do these figures represent the percent
310 interference if the interferent has the same concentration as NO?
We have used absorption cross sections from the MPI-Mainz UV/VIS Spectral Atlas of Gaseous Molecules (Keller-Rudek et al., 2013) for the estimation of the parallel interference. The particular photolysis rate of NO$_2$ and the regarded NO$_y$-compound was calculated by (Eq. 1)

315 $j = \int F(\lambda, T) \cdot \sigma(\lambda, T) \cdot \varphi(\lambda) \, \mathrm{d}\lambda,$                  (1)

where $F(\lambda, T)$ is the spectral emission of the UV LEDs, $\sigma(\lambda, T)$ the absorption cross section and $\varphi(\lambda)$ the quantum yield. For the calculation of the photolysis rate of the particular NO$_y$-compound the respective quantum yield was conservatively estimated to be equal 1, yielding an upper limit for the interference. On Page 5, L 120
320 now it says: The UV-induced positive bias in the NO$_2$-measurement due to photolysis of BrONO$_2$, HONO, NO$_3$ and ClNO$_2$ to produce NO was estimated at 6.1 %, 2.8 %, 2.7 % and 1.2 %, respectively, based on the absorption cross sections from the MPI-Mainz UV/VIS Spectral Atlas of Gaseous Molecules (Keller-Rudek et al., 2013). These values represent upper limits for the interference of the respective NO$_y$ compound as the respective molecular quantum yield was estimated conservatively at 1. Note that the values represent percent interferences

325 if the interferent had the same concentration as NO₂. Due to small daytime concentrations of these molecules in the MBL, a UV-induced bias was neglected for the observations in this study.
Please note that the values have been revised to 6.1 %, 2.8%, 2.7 % and 1.2 % for $BrONO_2$, HONO, $NO_3$ and $ClNO_2$, respectively. See further changes.

330

Comment 2: line 181 NO2 was further measured by cavity ring-down spectroscopy (Sobanski et al., 2016) and used for correcting the instrumental background of the CLD NO2 data, as described above (the correction was taken as the ultimate absolute measurement uncertainty in the CLD NO2 data).
The uncertainty of the NO2 chemiluminescent measurement is not equal to that of the cavity ring-down
335 instrument. The chemiluminescent NO2 is the difference between two measurements, one of which gets divided by 0.294 to take into account photolysis efficiency. In order to add errors in quadrature, I need to know the NO to NO2 ratio. I'm guessing that the relative uncertainty of NO2 will be at least twice that of NO. The NO2 chemiluminescent measurement is affected adversely by the relatively low photolysis efficiency. That accounts for random errors in NO2. Comparison with the cavity ringdown may take care of the NO2 instrumental
340 background but how does it take care of the random errors?
Random errors are taken up by the calculation of a one-minute average before the instrumental background was estimated from the difference in the two NO₂ measurements. Note that random errors in the data are expected to be reduced as our final data analysis is based on a five minute average. NO₂ is calculated by

345 $$NO_2 = \frac{NO_c - NO}{K_e} \quad ,$$

where $NO_c$ is the signal of the channel equipped with the photolytic converter, NO the signal of the NO-channel and $K_e$ the NO₂ conversion efficiency. We agree that the relative uncertainty in the NO₂ has to be estimated by means of the largest error possible from the relative uncertainties of NO (6 %), $NO_c$ (6 %) and the conversion
350 efficiency $K_e$ (3 %) rather than only from the uncertainty of the $NO_c$-signal and the uncertainty of the conversion efficiency. The revised TMU for NO₂ follows as:

$$\Delta NO_2 = \sqrt{\Delta NO^2 + \Delta NO_c^2 + \Delta K_e^2} = \sqrt{6\%^2 + 6\%^2 + 3\%^2} = 9\ \%.$$

355

We also agree that the absolute error of a single NO₂ data point might exceed 9 % when NO₂ is calculated from two high numbers (e.g. if we assume 100 ppbᵥ NO and 101 ppbᵥ $NO_c$, NO2 would be ~ 4 ppbᵥ), the error in NO₂ would be underestimated by 9 %. However the average and the median relative uncertainty of each data point of the campaign at 5 minute integration time are 13.6 % and 11.8 %, respectively, which is indeed slightly higher
360 than the statistical estimate of 9 %. However the above average of 13.6 % yields a standard deviation of 6.9 %, which yields that both the average and the median relative uncertainty of NO₂ are significantly not different from the statistical estimate of 9 %. As we finally calculate median values for days and regions, it is more practicable to estimate a statistical error for the NO₂ measurement at 9 %. The manuscript has been explicitly revised on Page 7, L 158-160. Now it says: The TMU in NO₂ has been estimated by means of the largest error possible from
365 error propagation at
$$\Delta NO_2 = \sqrt{\Delta NO^2 + \Delta NO_c^2 + \Delta K_e^2} = \sqrt{6\%^2 + 6\%^2 + 3\%^2} = 9\ \%$$
at a confidence level of $1\sigma$ and an integration time of five minutes.
Also the TMU of NO₂ in Table 1 has been corrected to 9 %.

370

Line 205 Total Measurement Uncertainty. All of these numbers appear very optimistic. In a previous comment, I gave my reasons why the TMU of NO2 appeared low. I do not know why the TMU of HO2 is not larger than OH. There are many sources of uncertainty in the conversion of HO2 to OH which is the quantity actually measured.

The authors state a 20% bias due to RO2 chemistry. I do not know to what extent quantification of HO2 is made easier than that of OH because there is more HO2 than OH and hence a larger signal.

The 1 sigma accuracy of OH is 20 % and the average precision is $3.4 \cdot 10^5$ molec cm$^{-3}$. The accuracy of HO$_2$ is 20 % and the largest uncertainty due to interference by contribution of RO$_2$ is 7 % or 3 ppt$_v$ whichever is higher. The average precision of HO$_2$ is 3 ppt$_v$. The uncertainty in OH is estimated as the 1 sigma accuracy whereas the uncertainty in HO$_2$ is estimated at $\sqrt{(20\ \%)^2 + (7\ \%)^2} \approx 21\ \%$. Note that HO$_2$ is calculated as the difference from two signals: HO$_2$ = (HO$_2$ + OH) − OH, whereas the signal (HO$_2$+OH) is about two order of magnitude larger than the signal from OH, which means that the uncertainty in OH can be neglected in the determination of the uncertainty in HO$_2$. On page 8, L 190-193 now it says: HO$_2$ data used in this study is still preliminary due to not yet corrected interference of organic peroxy radicals RO$_2$. The largest uncertainty due to interference by contribution of RO$_2$ is 7 % or 3 ppt$_v$, whichever is higher. The 1 sigma accuracy of both OH and HO$_2$ is 20 %. The uncertainty in the OH data is here estimated as the 1 sigma accuracy of the data set at 20 %, whereas the uncertainty in HO$_2$ is estimated at $\sqrt{(20\ \%)^2 + (7\ \%)^2} \approx 21\ \%$. The measurement uncertainty for HO$_2$ has been corrected to 21 % in Table 1.

Comment 3: Line 272 "In low NOx environments (< 100 pptv) previous studies have indicated that further NO oxidizing trace gases such as peroxy radicals (HO2, RO2) and halogen monoxides (XO) may result in a deviation from unity (Nakamura et al., 2003; Hosaynali Beygi et al., 2011; Reed et al., 2016)."

Deviations from the Leighton relation can also be important at higher NO. Departures of the Leighton ratio from unity depend (primarily) on the competition between HO2+NO and O3+NO. In polluted environments, HO2 concentrations can increase, remain steady, or decrease only slowly as NO is increased.

Several studies address deviations from the Leighton Ratio. However in high NO$_x$ regimes gaps between measurement and estimate can be closed by incorporating higher oxidations (HO$_2$, halogen monoxides). However, in low NO$_x$ regimes, even addressing these further oxidants does not close the gap between observation and estimate (see for instance Hosaynali Beygi et al., 2011). We have added the following sentence Page 10, L 238: Deviations from expected NO/NO$_2$-ratios at low NO$_x$ generally refer to missing oxidants converting NO to NO$_2$ (Hosaynali Beygi et al., 2011; Reed et al., 2016) or to a measurement error due to an instrumental background or a positive interference from thermal labile NO$_x$ reservoir species (Reed et al., 2016; Silvern et al., 2018).

Line 250 and following. Does Fig. S1 show the ratio of actinic flux in the 4 hour window centered around noon to the total measured actinic flux? Or is Fig. S1 a ratio obtained by fitting a Gaussian, between zeroes in the AM and PM?

You are right, Figure S1 shows the ratio of the actinic flux in the 4 hour window (centered to noon) to the total measured actinic flux of that particular day. The Gaussian Fit was only used to estimate noontime of each particular day. We have revised caption of Figure S1. Now the first sentence of the caption of supplementary Figure S1 says: Ratio of the noontime actinic flux ($\pm$ 2h around noon) with regard to the total actinic flux $j$(NO$_2$) of that particular day.

Comment 4: Lines 266-270, typos for O singlet D. Elsewhere, the "1" is in its proper place as a superscript.

Thank you for noticing. Now it says on Page 11, Line 269: $O(^1D) + H_2O \rightarrow 2OH$

Comment 5: Line 285. Am I correct that insofar as ozone production is concerned the only use of the HO2 and OH measurements is their contribution to the loss rate of ozone (and hence their effect on net ozone production)?

HO$_x$ data have been used to calculate net ozone production rates, however HO$_2$ has also been used as a reality test: The peroxy radical estimate based on Eq. 3 was compared to measured HO$_2$ in Section 3.2 (Page 21, L 442).

Comment 6: Line 373. "We find that the median NOx(model)/NOx(measurement)-ratio throughout the whole campaign is 0.91, indicating that the model underestimates NOx by roughly 10 %." The median of what? Could you please specify what items you are taking the median of; i.e., what are the data points. Ratios by Region? Days? Individual data points. I may have missed it; how long are data points? Equal to the 5 minute instrument averaging time?

The median $NO_x$(model)/$NO_x$(measurement) has been calculated as the median of five minute individual data points of the whole data set of the campaign. We have cleared out the misunderstanding in the text, now it says on Page 17, L386-387 following: We find that the median $NO_x$(model)/$NO_x$(measurement)-ratio of all five minute averaged data points of the whole campaign is 0.91, indicating that the model underestimates $NO_x$ by roughly 10 %.

Comment 7: Line 391. "Noontime RO2 was estimated based on Eq. 3. As the steady state assumption will not hold for air masses originating from fresh emissions (times to acquire steady state estimated from the inverse sum of the loss and production terms for NO2 typically ranged from 1-2 minutes during AQABA) and for fast changes in the actinic flux," What I think you want to say is mis-stated. As written, it says: We can't use samples that had fresh emissions, so we used samples taken when actinic flux was slowly changing. Was the data screened to eliminate time periods in which NOx (or less likely O3) was rapidly varying? From the looks of the actinic flux plot you did not have many clouds giving rapid variations in jNO2. I would be surprised if the time window around noon could not have been wider. How much does jNO2 change between, say noon-3 hours and noon – 175 minutes and what change in HO2 does that produce?

Please note that before performing photochemistry calculation, the whole data set was filtered based on a stack filter which was established based on parameters such as wind direction, wind speed, variability in NO data, $O_3$, $SO_2$ (see Lines 209-211). However rapid changes in the actinic flux $j(NO_2)$ were observed e.g. after sunrise and before sunset. For $\pm$ 2h around noon the variation of the actinic flux (within the 4 hour period) was about 7 %, whereas the variation of the actinic flux within a 6-hour window ($\pm$ 3h around noon) would be at least 20 % and assumptions that the actinic flux does not change within this time frame would not have been supportable. Differences in the actinic flux between noon-3 hours and noon-175 minutes would already be about 2 % of the maximum noontime value (and about the same change in $HO_2$). Differences in the actinic flux between noon-2 hours and noon–115 minutes would also be about 1-2 % of the maximum noontime value (slightly less than for 6 hours), but however expecting to decrease to zero towards noontime. Constant $RO_2$ and NOPR can be rather assumed for a 4-hour window around noon than for a 6-hour window around noon.

Comment 8: Figure 6. Obtaining peroxy radical concentrations from photostationary state calculations is not easy. Line 413 "the total uncertainty in the RO2 estimates is estimated at 14%. This is a way too optimistic estimate of the uncertainty of RO2. Instrument precision is too high and no account (except for NO2) is taken of biases. Even still, the PSS is a difference in two numbers, often of comparable magnitude. I'm not certain the error propagation was correctly performed. I would like to see the formula that represents "errors added in quadrature". Compare those results with a simple Monte Carlo calculation that can be done on a spreadsheet. And keep in mind that the result will not take into account errors in rate constants.

We agree that obtaining $RO_2$ from PSS calculations is not trivial and very much depends on the errors of the used quantities. The TMU in $RO_2$ was estimated based on Gaussian error propagation by means of the estimate of the largest error possible. $RO_2$ is calculated by

$$[RO_2] = \frac{j(NO_2) \cdot [NO_2] - k_{NO+O_3} \cdot [NO][O_3]}{k_{NO+HO_2} \cdot [NO]}.$$

The relative uncertainty in the RO₂ data has to be calculated from the relative uncertainty of NO, NO₂, O₃ and $j$(NO₂) at roughly 6 %, 9 % (instead of 7 %), 2 % and 10 %, respectively, which yields a relative uncertainty of 15 % for RO₂.

475

$$\Delta RO_2 = \sqrt{\Delta NO^2 + \Delta NO_2^2 + \Delta O_3^2 + \Delta j(NO_2)^2} = \sqrt{6\%^2 + 9\%^2 + 2\%^2 + 10\%^2} \approx 15\%$$

The relative uncertainty in RO₂ has been revised in the manuscript, now it says on Page 20 line 425-428: Based on the total measurement uncertainties of the measured quantities in Eq. 3, the uncertainty in RO₂ is estimated

480 by means of the largest error possible at 15 %.

$$\Delta RO_2 = \sqrt{\Delta NO^2 + \Delta NO_2^2 + \Delta O_3^2 + \Delta j(NO_2)^2} = \sqrt{6\%^2 + 9\%^2 + 2\%^2 + 10\%^2} \approx 15\% \qquad (9)$$

Note that our calculation assumes that errors in the used rate coefficients are negligible.

485 Comment 9: Line 416. "Negative values for all regions are regularly found in the vicinity of fresh emissions and air masses not in photochemical equilibrium" That might be the explanation. It would useful to quantify this point. HO2 and RO2 concentrations will be low in a high NOx environment. The negative values may reflect the measurement accuracy needed to distinguish, for example, – 5 ppt RO2 from zero.
Note that the uncertainty of the best estimates of RO₂ for each day/each region is estimated at 15 %. A reason
490 why negative values were regularly found in the vicinity of fresh emissions is that the assumption of photostationary steady state (PSS) might not be fulfilled for fresh emissions. Time to acquire PSS was estimated as the inverse sum of the loss and production terms of NO₂ and was of the order of 1-2 minutes during AQABA, depending on the particular conditions (temperature, radiation, ozone concentration). The assumption that the regarded air masses were in PSS might not be true in the vicinity of fresh emissions (with transport time from
495 source to instrument of less than 2 minutes). For these air masses production of NO₂ (oxidation of NO) prevailed over photolysis of NO₂ leading to $j(NO_2) \cdot [NO_2] - k_{NO+O_3} \cdot [NO][O_3] < 0$. Negative values might hence simply reflect that the regarded air mass has not reached PSS.

500 Comment 10: Figure 7. Here the RO2 data looks much better. There are differences mentioned in the text. but for this reader could you please provide a concise reason why Fig. 7. looks so much better than Fig. 6. Is it the data groupings? I assume that in both Fig. 6 and 7. the blue RO2 data is from Eq. 3. I am not totally positive because you were measuring HO2 and some fraction of RO2.
Figure 7 has been processed based on five minute data points in each region whereas Figure 6 has been
505 processed for single days which might be more biased by single data outliers. Grouping into regions rather than for single days allows for a statistically more established estimate.

Comment 11: Line 421. As peroxy radicals are short-lived molecules generated from the oxidation of VOCs,
510 enhanced RO2 concentrations observed over the Arabian Gulf are most likely due to high VOC observed over the Arabian Gulf are most likely due to high VOC emissions from intense oil and gas activities in the region. High HO2 can also occur in aged air masses in which NOx and VOCs have reacted away but still have significant O3 and (perhaps) HCHO. Photolysis could then yield peroxy radicals.
We have added the comment to the manuscript on Page 21, L. 438-439. However high HO₂ and RO₂ can also
515 occur in aged air massed with low NOₓ and VOCs but still significant O₃ (and perhaps HCHO whose photolysis would then yield peroxy radicals).

Comment 12: Line 440. Regarding the extrapolated actinic flux curve to get a daily ozone production. j(1OD) decreases early and late in the day faster than J(NO2). Not sure how much difference it makes. I would be very leery of this extrapolation over land; I'm assuming that you are far enough away that varying traffic and boundary layer heights are not a concern.

Except for $j(O^1D)*\alpha*[O_3]$ all other terms will scale with $j(NO_2)$. However errors due to a different diurnal profiles of $j(O^1D)$ will remain insignificant with respect to the relative uncertainty of NOPR, as NOPR are mainly determined by $k*[NO]*[RO_2]$ and loss reactions due to OH and $HO_2$.

Comment 13: Line 455 "the uncertainty of the regional NOPR is 40 % which has been estimated by error propagation" I don't disagree with this value. Merely surprised at its magnitude compared to a 15% uncertainty for RO2 from Eq. 3.

The uncertainty in NOPR was estimated by means of the largest error possible and is mainly driven by the high uncertainty in OH and $HO_2$ (20 % and 21 %, respectively) and by the uncertainty in $RO_2$ (15 %). Diurnal net ozone production rates were estimated by scaling the median hourly noontime value to a diurnal value (by multiplying by the hourly value by 4 due to a 4-hour window around noon and by further dividing by 0.461 (noontime fraction ($\pm$ 2h) of $j$NO$_2$). Strictly speaking, the relative uncertainty (6 %) in the division by 0.461, referred to as $\Delta s$ (see Supplements Figure 1), also needs to be accounted for when estimating the uncertainty in NOPR. Also the error in $\alpha$ (Eq. 5) is rather determined by the error in $H_2O$ (5 %) than from its relative variation over the campaign (relative variation ~ 21 %). This yields a measuremrent uncertainty for NOPR of

$$\Delta\text{NOPR} = \sqrt{\Delta\text{NO}^2 + \Delta\text{NO}_2^2 + \Delta\text{O}_3^2 + \Delta j(\text{NO}_2)^2 + \Delta j(\text{O}^1\text{D})^2 + \Delta\alpha^2 + \Delta\text{RO}_2^2 + \Delta\text{OH}^2 + \Delta\text{HO}_2^2 + \Delta s^2}$$

$$= \sqrt{6\%^2 + 9\%^2 + 2\%^2 + 10\%^2 + 10\%^2 + 5\%^2 + 15\%^2 + 20\%^2 + 21\%^2 + 6\%^2} \approx 38\%.$$

We have corrected the error in NOPR from 40 % to 38 % in the manuscript and specified error calculation in detail also for NOPR in the manuscript.

On Page 12, L274 now it says: The error in $\alpha$ is mainly determined by the error of $H_2O$ at 5 %.
On Page 21, L453 now it says: These noontime values are scaled to diurnal production rates (Figure 8). As photochemical net ozone production is in good approximation linear with actinic flux $j$(NO$_2$) and as on average (46.1 $\pm$ 2.8) % of the total $j$(NO$_2$) occurred $\pm$ 2h around noon, the median noontime NOPR estimate was multiplied by 4/0.461 $\approx$ 8.68 to obtain a diurnal value. The error in the total actinic flux located $\pm$ 2h around noon is estimated from the standard deviation of the best estimate of 0.461 at $\Delta s \approx$ 6 %.
On Page 23, L 472 now it says: Based on the total measurement uncertainties of the measured quantities in Eq. 3, the systematic error in NOPR is estimated from error propagation by means of the largest error possible at 38 %.

$$\Delta\text{NOPR} = \sqrt{\Delta\text{NO}^2 + \Delta\text{NO}_2^2 + \Delta\text{O}_3^2 + \Delta j(\text{NO}_2)^2 + \Delta j(\text{O}^1\text{D})^2 + \Delta\alpha^2 + \Delta\text{RO}_2^2 + \Delta\text{OH}^2 + \Delta\text{HO}_2^2 + \Delta s^2}$$

$$= \sqrt{6\%^2 + 9\%^2 + 2\%^2 + 10\%^2 + 10\%^2 + 5\%^2 + 15\%^2 + 20\%^2 + 21\%^2 + 6\%^2} \approx 38\%.$$

Comment 14: Line 473. "Although EMAC predicts high ozone levels over the Arabian Sea, it also reports the lowest NOPR in this region. Deviations between model-calculated estimate and the estimate based on measured

560    tracer data over the Mediterranean and over the Southern Red Sea could be linked to NOx being overestimated
       in the model in these regions." I'm not following. There is a low net ozone production rate. which to me implies
       that the model has a too low NO concentration, but you say that the model overpredicts NOx.
       As demonstrated in Figure 4, the model overpredicts the $NO_x$ measurement over the Mediterranean and the
       Southern Red Sea. As these regions are classified as rather $NO_x$ sensitive regimes (see Figure 10), higher $NO_x$
565    (in the model) will likewise imply higher NOPR (under stagnant chemistry, i.e. $NO_x$ sensitivity). The text is not fully
       clear and will be rewritten on P. 23 L492-495 as: Although EMAC predicts high ozone levels over the Arabian
       Sea, it also reports the lowest NOPR in this region. On the other side, the large overestimation of the model-
       calculated estimate NOPR against the one based on measured tracer data over the Mediterranean and over the
       Southern Red Sea could be linked to $NO_x$ being overestimated in the model in these regions.

570

       Comment 15: Section 3.4 VOC and NOx sensitivity. Makes sense.
       We have changed the title of chapter 3.4 to VOC- and $NO_x$-sensitivity.

575

**Further changes**

We have further made the following changes in the manuscript:

580     Change 1: There has been an updated version of the preliminary $HO_2$ data, which were used to revise NOPR in the different regions yielding 16 $ppb_v$ $day^{-1}$ over the Oman Gulf (instead of 14 $ppb_v$ $day^{-1}$), 32 $ppb_v$ $day^{-1}$ over the Arabian Gulf (instead of 28 $ppb_v$ $day^{-1}$) and -1 $ppb_v$ $day^{-1}$ over the Mediterranean (instead of -2 $ppb_v$ $day^{-1}$). The values have been corrected in the manuscript and the supplement, e.g. in Line 17 now it says: Net ozone production rates (NOPR) were greatest with 16 $ppb_v$ $day^{-1}$ over both the Gulf of Oman and the Northern Red Sea and with 32 $ppb_v$ $day^{-1}$ over the Arabian Gulf.

585     In the manuscript in Line 490 and Line 569 the values have also been revised. We have also revised the median noontime $RO_2$(measurement estimate)/$HO_2$(estimate)-ratio to 1.88 in Line 450-451: We find that the median noontime $RO_2$(measurement estimate)/$HO_2$(estimate)-ratio throughout the whole campaign is 1.88.

    A new file with updated $HO_2$ data has been uploaded to zenodo and a new DOI has been generated. The data is now available at https://doi.org/10.5281/zenodo.3693988.

590

    Change 2: Each end of the photolytic converter features 200 UV LED units instead of 50 UV LED units. In Line 122, 123 now it says: The photolytic $NO_2$ converter features a set of 200 UV LED units attached to each end of the converter.

    Change 3: The calculation of the UV-induced positive bias was revised, which reduced the estimated maximum interferences

595     of $BrONO_2$, HONO, $NO_3$ and $ClNO_2$ to 6.1 %, 2.8 %, 2.7 % and 1.2 %, respectively (instead of 7.7 %, 7.2 %, 5.6 % and 1.5 % for HONO, $BrONO_2$, $NO_3$ and $ClNO_2$, respectively). Now it says in Line 124-127: The UV-induced positive bias in the $NO_2$-measurement due to photolysis of $BrONO_2$, HONO, $NO_3$ and $ClNO_2$ to produce NO was estimated at 6.1 %, 2.8 %, 2.7 % and 1.2 %, respectively, based on the absorption cross sections from the MPI-Mainz UV/VIS Spectral Atlas of Gaseous Molecules (Keller-Rudek et al., 2013).

600

    Change 4: In the manuscript it says "supplementary Figure" and "supplementary Table" instead of "supplements Figure" and "supplements Table", respectively. Also we now write "Figure" and "Table" instead of "Fig." and "Tab.", respectively, and we have revised the used date format in the manuscript to, e.g. 21 July 2017 instead of July 21st 2017.

605     Change 5: Table 1 was moved into the supplement. Figure S5 was also added to the supplement. All other figures and tables have been numbered accordingly in the manuscript and supplement.

    Change 6: In the caption of supplementary Figures S6-S9 now it says "The error bars represent the 25-75-percentile variation" instead of "The error bars are represented by the 25-75-percentile variation"

610

    Changes made to the final manuscript and supplements are written in red.

[revised manuscript text omitted]

Figure S1 shows a day to day calculation of the correction factor to scale the fractional noontime integral to a diurnal value.
840 An average $\pm$ standard deviation of (46.1 $\pm$ 2.8) % of the diurnal integral within $\pm$ 2h around noontime was estimated.

[Figure]

**Figure S1: Ratio of the noontime actinic flux ($\pm$ 2h around noon) with regard to the total actinic flux $j$(NO$_2$) of that particular day. Dashed line represents the campaign average of (46.1 $\pm$ 2.8) %. The errors bars are represented by the relative uncertainty (6 %) of the campaign average.**

[Figure]

845

**Figure S2: Timeline of NO, NO₂ (both CLD), O₃, OH, HO₂ preliminary and *j*(NO₂) data during the first leg. See Table ST2 for additional information on the ship cruise. Note that HO₂ data are preliminary.**

[Figure]

850 **Figure S3: Timeline of NO, NO₂ (both CLD), O₃, OH, HO₂ preliminary and *j*(NO₂) data during the second leg. See Table ST2 for additional information on the ship cruise. Note that HO₂ data are preliminary.**

[Figure]

**Figure S4: Ship cruises with color-scaled OH mixing ratios a) during the first and b) the second leg and color-scaled HO₂ mixing ratios (preliminary) c) during the first and d) the second leg. Note that OH and preliminary HO₂ data have been filtered for own stack contamination.**

[Figure]

**Figure S5: Ship cruises with color-scaled absolute humidity a) during the first and b) the second leg.**

[Figure]

**Figure S6: Scatterplot of simulated and measured regional NOx median in ppbv. 1:1 line added for orientation. The error bars represent the 25-75-percentile variation.**

[Figure]

865    **Figure S7: Scatterplot of simulated and measured regional O₃ median in ppbᵥ. 1:1 line added for orientation. The error bars represent the 25-75-percentile variation.**

[Figure]

**Figure S8: Scatterplot of simulated regional RO₂ median and regional RO₂ median estimated based on measured tracer data in pptᵥ. 1:1 line added for orientation. The error bars represent the 25-75-percentile variation.**

[Figure]

870

**Figure S9: Scatterplot of median NOPR in ppb$_v$ day$^{-1}$ estimated based on simulated and measured tracer data. 1:1 line added for orientation. The error bars represent the 25-75-percentile variation.**

[Figure]

875

**Figure S10: Comparison of the regional, absolute contribution of $k_{NO+HO_2}$[NO][RO$_2$] to NOPR in the six different regions investigated during AQABA. The horizontal black bar indicates the median value, the box the 25- and 75-percentiles and the whiskers the 10- and 90-percentiles.**

[Figure]

880 **Figure S11: Comparison of the regional, absolute contribution of $-j(O^1D) \cdot \alpha \cdot [O_3]$ to NOPR in the six different regions investigated during AQABA. The horizontal black bar indicates the median value, the box the 25- and 75-percentiles and the whiskers the 10- and 90-percentiles.**

[Figure]

885

**Figure S12: Comparison of the regional, absolute contribution of $-k_{\mathrm{HO_2+O_3}}[\mathrm{HO_2}][\mathrm{O_3}]$ to NOPR in the six different regions investigated during AQABA. The horizontal black bar indicates the median value, the box the 25- and 75-percentiles and the whiskers the 10- and 90-percentiles.**

[Figure]

890

**Figure S13: Comparison of the regional, absolute contribution of** $-k_{OH+O_3}[OH][O_3]$ **to NOPR in the six different regions investigated during AQABA. The horizontal black bar indicates the median value, the box the 25- and 75-percentiles and the whiskers the 10- and 90-percentiles.**

895 **Table ST1: Range of latitudinal and longitudinal coordinates and dates during both legs of the different regions.**

| region (abbreviation) | latitudinal range | longitudinal range | Date (1st leg) | Date (2nd leg) |
|---|---|---|---|---|
| Mediterranean (M) | 31.810° N-39.923° N | 12.620° E-31.850° E | --- | 25.08.2017 – 31.08.2017 |
| Northern Red Sea (NRS) | 23.343° N-30.986° N | 32.305° E-37.085° E | 03.07.2017 – 08.07.2017 | 21.08.2017 – 24.08.2017 |
| Southern Red Sea (SRS) | 12.672° N-22.494° N | 37.411° E-43.327° E | 09.07.2017 – 16.07.2017 | 17.08.2017 – 20.08.2017 |
| Arabian Sea (AS) | 11.797° N-22.782° N | 44.035° E-60.636° E | 18.07.2017 – 24.07.2017 | 07.08.2017 – 16.08.2017 |
| Oman Gulf (OG) | 23.050° N-25.622° N | 56.492° E-59.913° E | 24.07.2017 – 27.07.2017 | 05.08.2017 – 07.08.2017 |
| Arabian Gulf (AG) | 25.396° N-29.425° N | 47.920° E-56.772° E | 27.07.2017 – 31.07.2017 | 03.08.2017 – 05.08.2017 |

**Table ST2: Overview of the time spent in the particular regions during AQABA. Red color indicates periods with KI at anchor that are not included in the data analysis. Data measured during bunkering at Fujairah City (06 August, 07:00-15:00 UTC) were also not included in the analysis.**

| Date | Region | Date | Region |
|---|---|---|---|
| 03.07.2017 | Northern Red Sea | 02.08.2017 | Kuwait port |
| 04.07.2017 | Northern Red Sea | 03.08.2017 | Kuwait port/Arabian Gulf |
| 05.07.2017 | Northern Red Sea | 04.08.2017 | Arabian Gulf |
| 06.07.2017 | Northern Red Sea | 05.08.2017 | Arabian Gulf/Oman Gulf |
| 07.07.2017 | Northern Red Sea | 06.08.2017 | Oman Gulf |
| 08.07.2017 | Northern Red Sea | 07.08.2017 | Oman Gulf/Arabian Sea |
| 09.07.2017 | Southern Red Sea | 08.08.2017 | Arabian Sea |
| 10.07.2017 | Southern Red Sea | 09.08.2017 | Arabian Sea |
| 11.07.2017 | Southern Red Sea/Jeddah port | 10.08.2017 | Arabian Sea |
| 12.07.2017 | Jeddah port | 11.08.2017 | Arabian Sea |
| 13.07.2017 | Jeddah port/Southern Red Sea | 12.08.2017 | Arabian Sea |
| 14.07.2017 | Southern Red Sea | 13.08.2017 | Arabian Sea |
| 15.07.2017 | Southern Red Sea | 14.08.2017 | Arabian Sea |
| 16.07.2017 | Southern Red Sea | 15.08.2017 | Arabian Sea |
| 17.07.2017 | Djibouti port | 16.08.2017 | Arabian Sea |
| 18.07.2017 | Arabian Sea | 17.08.2017 | Southern Red Sea |
| 19.07.2017 | Arabian Sea | 18.08.2017 | Southern Red Sea |
| 20.07.2017 | Arabian Sea | 19.08.2017 | Southern Red Sea |
| 21.07.2017 | Arabian Sea | 20.08.2017 | Southern Red Sea |
| 22.07.2017 | Arabian Sea | 21.08.2017 | Northern Red Sea |
| 23.07.2017 | Arabian Sea | 22.08.2017 | Northern Red Sea |
| 24.07.2017 | Arabian Sea/Oman Gulf | 23.08.2017 | Northern Red Sea |
| 25.07.2017 | Oman Gulf | 24.08.2017 | Northern Red Sea |
| 26.07.2017 | Oman Gulf | 25.08.2017 | Mediterranean |
| 27.07.2017 | Oman Gulf/Arabian Gulf | 26.08.2017 | Mediterranean |
| 28.07.2017 | Arabian Gulf | 27.08.2017 | Mediterranean |
| 29.07.2017 | Arabian Gulf | 28.08.2017 | Mediterranean |
| 30.07.2017 | Arabian Gulf | 29.08.2017 | Mediterranean |
| 31.07.2017 | Arabian Gulf/Kuwait port | 30.08.2017 | Mediterranean |
| 01.08.2017 | Kuwait port | 31.08.2017 | Mediterranean |

**Table ST3: Overview of measured NO$_x$ (upper table) and measured O$_3$ (lower table) spatial volume mixing ratio average, standard deviation, 1st quantile, median, 3rd quantile (all in ppb$_v$) and number of considered data points.**

| NO$_x$ (upper), O$_3$ (lower) | Mediterranean | Northern Red Sea | Southern Red Sea | Arabian Sea | Oman Gulf | Arabian Gulf |
|---|---|---|---|---|---|---|
| data points | 1767 | 1694 | 1755 | 2656 | 1056 | 1539 |
| average | 1.24 | 4.69 | 1.62 | 0.95 | 4.16 | 3.65 |
| stdev | 3.34 | 7.9 | 3.7 | 3.15 | 4.33 | 9.24 |
| 1st quantile | 0.12 | 0.68 | 0.18 | 0.10 | 1.03 | 0.52 |
| median | **0.25** | **1.76** | **0.46** | **0.19** | **2.74** | **1.26** |
| 3rd quantile | 0.96 | 5.68 | 1.6 | 0.54 | 5.92 | 3.47 |
| data points | 2010 | 2717 | 2307 | 4130 | 1249 | 1809 |
| average | **61.56** | **63.39** | **50.35** | **21.53** | **34.04** | **73.99** |
| stdev | 8.25 | 18.45 | 12.96 | 6.8 | 11.27 | 35.68 |
| 1st quantile | 57.05 | 53.51 | 40.68 | 17.45 | 26.66 | 53.08 |
| median | **61.54** | **64.16** | **46.93** | **22.52** | **31.5** | **62.5** |
| 3rd quantile | 66.48 | 75.51 | 60.28 | 26.19 | 38.03 | 90.42 |

**Table ST4: Overview of simulated NO$_x$ (upper table) and simulated O$_3$ (lower table) spatial volume mixing ratio average, standard deviation, 1st quantile, median, 3rd quantile (all in ppb$_v$) and number of considered data points.**

| NO$_x$ (upper), O$_3$ (lower) | Mediterranean | Northern Red Sea | Southern Red Sea | Arabian Sea | Oman Gulf | Arabian Gulf |
|---|---|---|---|---|---|---|
| data points | 2012 | 2719 | 2310 | 4464 | 1253 | 1810 |
| average | 0.84 | 1.27 | 1.13 | 0.31 | 1.88 | 1.91 |
| stdev | 0.75 | 1.97 | 0.62 | 0.29 | 1.47 | 1.37 |
| 1st quantile | 0.33 | 0.43 | 0.67 | 0.14 | 0.82 | 1.17 |
| median | **0.43** | **0.76** | **0.97** | **0.18** | **1.59** | **1.61** |
| 3rd quantile | 1.27 | 1.08 | 1.51 | 0.39 | 2.05 | 2.16 |
| data points | 2012 | 2719 | 2310 | 4464 | 1253 | 1810 |
| average | **65.15** | **64.76** | **49.5** | **36.91** | **55** | **76.85** |
| stdev | 4.99 | 7.7 | 7.21 | 3.87 | 9.94 | 12.8 |
| 1st quantile | 61.35 | 60.34 | 43.47 | 34.12 | 48.37 | 65.88 |
| median | **65.53** | **64.6** | **49.33** | **36.54** | **55.01** | **76.39** |
| 3rd quantile | 69.33 | 68.25 | 54.58 | 38.85 | 61.42 | 86.22 |

**Table ST5: Overview of noontime RO$_2$ spatial volume mixing ratio average, standard deviation, 1st quantile, median, 3rd quantile (all in ppt$_v$) estimated based on measured tracer data. Number of considered data points added in the first line.**

| RO$_2$ | Mediterranean | Northern Red Sea | Southern Red Sea | Arabian Sea | Oman Gulf | Arabian Gulf |
|---|---|---|---|---|---|---|
| data points | 288 | 126 | 190 | 338 | 166 | 242 |
| average | 13 | 27 | 7 | 64 | 23 | 94 |
| stdev | 24 | 20 | 34 | 83 | 42 | 113 |
| 1st quantile | 1 | 15 | -13 | 23 | -8 | 11 |
| median | **16** | **28** | **15** | **33** | **22** | **73** |
| 3rd quantile | 26 | 44 | 27 | 54 | 49 | 176 |

910 **Table ST6: Overview of simulated noontime RO$_2$ spatial volume mixing ratio average, standard deviation, 1st quantile, median, 3rd quantile (all in ppt$_v$). Number of considered data points added in the first line.**

| RO$_2$ | Mediterranean | Northern Red Sea | Southern Red Sea | Arabian Sea | Oman Gulf | Arabian Gulf |
|---|---|---|---|---|---|---|
| data points | 336 | 192 | 192 | 720 | 203 | 293 |
| average | 41 | 46 | 36 | 40 | 48 | 49 |
| stdev | 5 | 4 | 11 | 5 | 6 | 7 |
| 1st quantile | 36 | 43 | 25 | 38 | 44 | 44 |
| median | **41** | **46** | **38** | **41** | **50** | **49** |
| 3rd quantile | 46 | 49 | 42 | 43 | 53 | 54 |

**Table ST7: Overview of NOPR average, standard deviation, 1st quantile, median, 3rd quantile (all in ppb$_v$ day$^{-1}$) estimated based on measured tracer data. Number of considered data points added in the first line.**

| NOPR | Mediterranean | Northern Red Sea | Southern Red Sea | Arabian Sea | Oman Gulf | Arabian Gulf |
|---|---|---|---|---|---|---|
| data points | 148 | 114 | 89 | 187 | 84 | 111 |
| average | -3 | 16 | -5 | -67 | -105 | -24 |
| stdev | 43 | 39 | 12 | 576 | 362 | 449 |
| 1st quantile | -5 | 6 | -9 | 1 | -18 | 25 |
| median | **-1** | **16** | **-4** | **5** | **16** | **32** |
| 3rd quantile | 8 | 40 | -1 | 11 | 23 | 65 |

915

**Table ST8: Overview of NOPR average, standard deviation, 1st quantile, median, 3rd quantile (all in ppb$_v$ day$^{-1}$) estimated based on simulated tracer data. Number of considered data points added in the first line.**

| NOPR | Mediterranean | Northern Red Sea | Southern Red Sea | Arabian Sea | Oman Gulf | Arabian Gulf |
|---|---|---|---|---|---|---|
| data points | 336 | 192 | 192 | 720 | 203 | 293 |
| average | 11 | 16 | 12 | 5 | 30 | 38 |
| stdev | 8 | 12 | 6 | 6 | 11 | 10 |
| 1st quantile | 5 | 8 | 8 | 1 | 22 | 30 |
| median | **8** | **12** | **11** | **2** | **28** | **35** |
| 3rd quantile | 17 | 21 | 15 | 10 | 37 | 46 |

920

**Table ST9: Overview of measured HCHO/NO$_2$-ratio average, standard deviation, 1st quantile, median, 3rd quantile and number of considered data points.**

| Ratio | Mediterranean | Northern Red Sea | Southern Red Sea | Arabian Sea | Oman Gulf | Arabian Gulf |
|---|---|---|---|---|---|---|
| data points | 203 | 79 | 48 | 252 | 108 | 122 |
| average | 5.4 | 1.5 | 8.5 | 11.1 | 2.7 | 9 |
| stdev | 4.7 | 0.7 | 6.5 | 8.9 | 2.1 | 6.4 |
| 1st quantile | 1.3 | 0.8 | 4.4 | 2.3 | 1 | 2.5 |
| median | **5** | **1.4** | **7.7** | **9.4** | **2.2** | **9.3** |
| 3rd quantile | 7.4 | 2.1 | 9.8 | 16.1 | 3.6 | 12.7 |

925 **Table ST10: List of included peroxy radicals (with less than four carbon atoms) for the reaction with NO as recommended by Sander et al. (2019).**

| Species |
| --- |
| $HO_2$ |
| $CH_3O_2$ |
| $C_2H_5O2$ |
| $C_2H_5CO_3$ |
| $CH_3CO_3$ |
| C3DIALO2 ($C_3H_3O_4$) |
| $CH_3CHOHO_2$ |
| $CH_3COCH_2O_2$ |
| $CH_3COCO_3$ |
| $CHOCOCH_2O_2$ |
| $CO_2H_3CO_3$ |
| $HCOCH_2CO_3$ |
| $HCOCH_2O_2$ |
| $HCOCO_3$ |
| $HCOCOHCO_3$ |
| $HOC_2H_4CO_3$ |
| $HOCH_2CH_2O_2$ |
| $HOCH_2CO_3$ |
| $HOCH_2COCH_2O_2$ |
| $HOCH_2O_2$ |
| $CH_3CHO_2CH_2OH$ |
| IC3H7O2 (isopropylperoxy radical) |
| NC3H7O2 (propylperoxy radical) |
| $NCCH_2O_2$ |
| $NO_3CH_2CO_3$ |
| $CH_3CHO_2CH_2ONO_2$ |

---

## Editor Decision (ED1)

Editor Comments (Andreas Hofzumahaus)

I am pleased that the authors have addressed the referees' comments in detail. With respect to the main comment by referee #2, the treatment of error propagation has been adequately corrected. Nevertheless, I am not fully satisfied with the error estimates.

- The CLD instrument measures NO and NOc by chemiluminescence. The NO detection involves photon counting and the statistical noise of the signals should follow Poisson statistics. The absolute $1\sigma$ precision should scale with the square root of the NO signal for a given integration time, whereas the relative precision (in percent) is expected to decrease with the square root of the NO concentration. In line 156 of their revised manuscript, the authors specify a general constant value for the relative precision (5%). How can that be? Same arguments apply to the measurement of NOc.

- In line 157, the authors calculate a total relative uncertainty of NO from the signal precision (5%) and the error of the calibration gas mixture (3%). The same is done for NOc. In the calculation of the total uncertainty for NO2, the total uncertainties for NO and NOc are treated as statistically independent. This is not correct, because the calibrations of NO and NOc rely on the same calibration standard. The correct procedure to calculate the uncertainty of NO2 would be to calculate first the precision of NO2 from the precisions of NO and NOc. In a second step, the precision of NO2 can be combined with the uncertainty from the calibration mixture.

- In line 166, the authors assume that the error of the NO measurement is zero when no NO is present. However, when NO approaches zero, the signal noise will be dominated by the instrumental background (e.g., dark signal) which is responsible for the limit of detection. The role of background noise should be clarified.

Further technical comments.

- In the paper, HO2 data are called "preliminary" because it has not been corrected for interferences from organic peroxy radicals. Since the discovery of possible RO2 interferences in the detection of HO2 by LIF, uncorrected HO2 is generally called HO2* (e.g., Lu et al., Atmos. Chem. Phys., 12, 1541–1569, 2012). I recommend to adopt this nomenclature.

- In line 199, the possible RO2 interference is treated as a statistical error when calculating the uncertainty of HO2. This is not appropriate. The interference is an additive positive bias. If the interference is 7%, then all HO2* values will be too high by this amount. Please clarify: where does the value of 7% come from? Why don't you correct the HO2* data by this amount?

- In Table 1, I have the impression that TMU is used either for precision, or accuracy, or a combination of both. I suggest to specify instead the limit of detection and accuracy (usually related to calibration) for each substance in separate columns.

- Equation (4) in line 271 looks weird. I find the re-definition of RO2 as the total sum of peroxy radicals (including HO2) confusing and inconsistent with other parts of the paper (e.g., line 196, line 239, reactions R4-R6). According to the general nomenclature, "RO2" should be used for the sum of organic peroxy radicals without HO2. I suggest to modify the equation accordingly:
  $$P(O3) = k_{NO+RO2} [NO] [RO2] + k_{NO+HO2} [NO] [HO2]) \cong k_{NO+HO2} [NO] ([RO2] + [HO2])$$

---

## Author Response (AR2)

**Reply to Reviewer Report 1**

*In the following the comments of the reviewer are presented (black) alongside with our replies (in blue) and changes made to the manuscript (in red).*

General statement: Review of Tadic et al revision of "Net ozone production and its relationship to NOx and VOCs in the marine boundary layer around the Arabian Peninsula" for ACP

Many of my comments concern uncertainty estimates. The authors add errors in quadrature, as appropriate for determining the variance of a quantity that depends on several variables. However, it is not the variance that is used in the paper; derived variables are presented with relative uncertainties as a percent of the variable value.

Dear reviewer, thank you very much for reviewing our revised manuscript and for the insightful comments. Below we provide the demanded corrections.

I will use the error analysis for NO2 as an example. I have written this text with the expectation that Greek characters, partial derivative symbols, and exponents will not survive cut and paste: d's are partial derivative, ^2 means squared, and sigma is written out. The same analysis applies to other quantities:

$$x = f(u, v, ...) \quad (1)$$

If errors are normally distributed,

$$\text{sigma}(x)^2 = \text{sigma}(u)^2 (dx/du)^2 + \text{sigma}(v)^2 (dx/dv)^2 + ... \quad (2)$$

NO and NO2 are determined from the chemiluminescent reaction NO+O3 -> NO2; NO is measured directly; in a second channel, called NOc, a light source converts a fraction K of NO2 into NO. As noted in the response the concentration of NO2 is given by

$$NO2 = (NOc-NO)/K \quad (3)$$

The relative uncertainties of NO. NOc, and K are given as 6%, 6% and 3% respectively. Applying Eq 2 to NO2, gives

$$\text{sigma}(NO2)^2 = \text{sigma}(NOc)^2 (dNO2/dNOc)^2 + \text{sigma}(NO)^2 (dNO2/dNO)^2 + \text{sigma}(K)^2 (dNO2/dK)^2$$

Evaluating the partial derivatives, gives

$$\text{sigma}(NO2)^2 = \text{sigma}(NOc)^2 *(1/K)^2 + \text{sigma}(NO)^2 *(1/K)^2 + \text{sigma}(K)^2 *((NOc-NO)/K)^2$$

Define the relative uncertainty in x as R(x) = sigma(x)/x. Then,

$$R(NO2)^2 = R(NOc)^2 *(NOc/NO2)^2 *(1/K)^2 + R(NO)^2 *(NO/NO2)^2 * (1/K)^2 + R(K)^2 * (K/NO2)^2 * ((NOc-NO)/K^2)^2$$

The uncertainty of NO2 depends on the concentration of NO and NO2. As an example, NO2 = 4ppb, NO = 1 ppb, giving NOc = 2.2 ppb. For convenience, I have rounded K to 0.3. With these concentrations, the above formula based on adding errors in quadrature gives R(NO2) = 12.4%. Equation (2), properly applied can give uncertainties that are greater or less than that obtained by the addition-in-quadrature formulas of the revised paper. In this case, the uncertainty is greater than given in the paper. I would hope that the input uncertainties, R(NO), R(NOc), and

R(K) were determined as in Eqs 1-2, rather than by the formulas in the paper, or worse still by manufacturers recommendations.

Similar considerations apply to the formulas in the manuscript for Delta(HO2), Delta(RO2), and Delta(NOPR).

Regarding the measurement of HO2, my understanding is that HO2 is converted to OH and OH is measured by LIF. The point that I was trying to make is that if the uncertainty in OH is 20% (independent of concentration), then the uncertainty of HO2 has to be greater, as there are uncertainties associated with the conversion.

The relative uncertainties of the signal of the two channels R(NO) and R(NO$_c$) have been estimated by adding errors in quadrature, analogously to Beygi et al. (2011). In the case of NO, the relative uncertainty (total measurement uncertainty) is calculated at 6 % by adding the errors of the calibration gas mixture concentration and the precision in quadrature.

$$\text{TMU}([\text{NO}]) = \sqrt{(5\ \%)^2 + (3\ \%)^2} \approx 6\ \%$$

Here 5 % represents the precision of the NO-channel and 3 % the uncertainty of the calibration gas mixture. The relative uncertainty associated with the NO$_c$ data (R(NO$_c$)) has been analogously calculated at 6 %. The conversion efficiency $K_e$ has been estimated as the average of four gas phase titration measurements performed during the campaign. The relative uncertainty associated with $K_e$ (3 %) has been estimated as the standard deviation of the averages. So far we would like to stay with the input uncertainties.

To be correct and as recommended, we have revised the estimation of relative uncertainties associated with NO$_2$, RO$_2$ and NOPR. NO$_2$ is calculated by

$$[\text{NO}_2] = \frac{[\text{NO}_c] - [\text{NO}]}{K_e} \tag{1}$$

As you have correctly derived, following error propagation the relative uncertainty in NO$_2$ is calculated by

$$\text{TMU}([\text{NO}_2]) = \frac{1}{[\text{NO}_2]} \cdot \sqrt{\left(\frac{\Delta[\text{NO}_c]}{K_e}\right)^2 + \left(\frac{\Delta[\text{NO}]}{K_e}\right)^2 + \left(\frac{\Delta K_e \cdot ([\text{NO}_c] - [\text{NO}])}{K_e^2}\right)^2} \tag{2}$$

Note that for convenience, we have separated the factor 1/[NO$_2$] from the square root. Eq. 2 yields a relative error for each single data point. A statistic over the course of the campaign yields a median relative uncertainty of 8 % and an average of 16 % of the NO$_2$ data set. The median is lower than the average as NO is practically zero during nighttime. During nighttime the relative error of the NO$_2$ data points is about 6.7 %. Note for convenience that is exactly the value calculated from errors in quadrature (if NO is practically zero during nighttime): $\sqrt{6\%^2 + 0\%^2 + 3\%^2} \approx 6.7\ \%$. However, the total measurement uncertainty (relative uncertainty) in NO$_2$ has been conservatively estimated at 16 % as the average of the relative uncertainty of all data points obtained during AQABA, which will correctly address (and possibly also overestimate) most of the errors of single data points.

Similar calculations apply for [RO$_2$], which is calculated based on in situ measurements of NO, NO$_2$, O$_3$ (and $j$(NO$_2$)).

$$[\text{RO}_2] = \frac{j(\text{NO}_2) \cdot [\text{NO}_2] - k_{\text{NO}+\text{O}_3} \cdot [\text{NO}][\text{O}_3]}{k_{\text{NO}+\text{HO}_2} \cdot [\text{NO}]} \tag{3}$$

The relative error in RO$_2$ is calculated by error propagation (Eq. 4) below

$$R([RO_2]) = \frac{1}{[RO_2]} \cdot \sqrt{\begin{array}{c} \left(\frac{\Delta j(NO_2) \cdot [NO_2]}{k_{NO+HO_2} \cdot [NO]}\right)^2 + \left(\frac{\Delta[NO_2] \cdot j(NO_2)]}{k_{NO+HO_2} \cdot [NO]}\right)^2 + \left(\frac{\Delta[O_3] \cdot k_{NO+O_3} \cdot [NO]}{k_{NO+HO_2} \cdot [NO]}\right)^2 + \\ \left(\Delta[NO] \cdot \left(\frac{-k_{NO+O_3} \cdot [O_3] \cdot k_{NO+HO_2} \cdot [NO] - k_{NO+HO_2} \cdot (j(NO_2) \cdot [NO_2] - k_{NO+O_3} \cdot [O_3] \cdot [NO])}{\left(k_{NO+HO_2} \cdot [NO]\right)^2}\right)\right) \end{array}}$$

(4). Again a statistic over the relative errors of each $RO_2$ data point has been calculated: the median of the relative $RO_2$ error of all data points obtained during AQABA is 56 %, the average with 132 % is not reliable as it seems to be biased by single data outliers. The relative error associated with the $RO_2$ calculation is hence estimated at 56 % (instead of 15 %).

Similar calculations apply for NOPR, which is calculated by

$$NOPR = k_{NO+RO_2}[NO][RO_2] - [O_3] \cdot (\alpha \cdot j(O^1D) + k_{OH+O_3}[OH] + k_{HO_2+O_3}[HO_2]). \quad (5)$$

The relative error in NOPR is calculated by error propagation (Eq. 6) below

$$R(NOPR) = \frac{1}{NOPR} \cdot \sqrt{\begin{array}{c} \left(k_{NO+HO_2} \cdot [NO] \cdot \Delta[RO_2]\right)^2 + \left(k_{NO+HO_2} \cdot \Delta[NO] \cdot [RO_2]\right)^2 + \\ \left(\Delta[O_3] \cdot \left(\alpha \cdot j(O^1D) + k_{OH+O_3} \cdot [OH] + k_{HO_2+O_3} \cdot [HO_2]\right)\right)^2 + \\ (\Delta j(O^1D) \cdot [O_3] \cdot \alpha)^2 + (j(O^1D) \cdot [O_3] \cdot \Delta\alpha)^2 + \left(\Delta[OH] \cdot k_{OH+O_3} \cdot [O_3]\right)^2 + \\ \left(\Delta[HO_2] \cdot k_{HO_2+O_3} \cdot [O_3]\right)^2 \end{array}}$$

(6). Incorporating a relative error of 16 % associated with $NO_2$ and a relative error of 56 % associated with $RO_2$, the median of the relative NOPR error of all data points obtained during AQABA is 69 %. The average relative uncertainty of NOPR is 15 % and strongly biased by single data outliers, which are in the case of NOPR significantly negative (due to fresh emissions and titration of $O_3$ by NO). Again the median is a more representative measure for the general uncertainty associated with the NOPR calculations. The relative error associated with the NOPR estimates based on measured data is hence estimated at 69 %.

As stated by the data owners, the 1 sigma accuracy for both OH and $HO_2$ is equally 20 %. The additional uncertainty for $HO_2$ (associated with the NO titration) is the interference by contribution of $RO_2$ which is 7 % or 3 $ppt_v$, whichever is higher. The uncertainty in $HO_2$ is hence estimated by adding these errors in quadrature at $\sqrt{(20\,\%)^2 + (7\,\%)^2} \approx$ 21 %.

We now present the calculation of the TMU in NO. The manuscript has been revised on Page 7, L155: The total measurement uncertainty (TMU) in NO has been calculated at 6 % at an integration time of 5 minutes and a confidence level of $1\sigma$ by adding the precision (5 %) and the error of the calibration gas mixture concentration (3 %) in quadrature: $TMU([NO]) = \sqrt{(5\,\%)^2 + (3\,\%)^2} \approx 6\,\%$.
The manuscript has been also revised on Page 7, L158: The TMU in $NO_2$ has been estimated by error propagation.

$$R([NO_2]) = \frac{1}{[NO_2]} \cdot \sqrt{\left(\frac{\Delta[NO_c]}{K_e}\right)^2 + \left(\frac{\Delta[NO]}{K_e}\right)^2 + \left(\frac{\Delta K_e \cdot ([NO_c] - [NO])}{K_e^2}\right)^2}$$

Note that the total measurement uncertainty of the $NO_c$-channel data has also been calculated at 6 % at an integration time of 5 minutes and a confidence level of $1\sigma$ by adding the precision and the error of the calibration gas mixture in quadrature. Over the course of the campaign the median and the average relative uncertainty of $NO_2$ are 8 % and 16 %, respectively. The median is lower than the average as NO is practically zero during

nighttime. During nighttime the relative error of the $NO_2$ data points is about 6.7 %. Note for convenience that is exactly the value calculated from errors in quadrature (if NO is practically zero during nighttime): $\sqrt{6\%^2 + 0\%^2 + 3\%^2} \approx 6.7$ %. The relative uncertainty in $NO_2$ has been estimated as a conservative upper limit at 16 % as the average of the relative uncertainty of all data points obtained during AQABA.

The manuscript has also been revised on Page 20, L425: The relative uncertainty associated with the $RO_2$ estimate has been calculated by error propagation of Eq. 3.

$$R([RO_2]) = \frac{1}{[RO_2]} \cdot \sqrt{\begin{array}{l} \left(\frac{\Delta j(NO_2) \cdot [NO_2]}{k_{NO+HO_2} \cdot [NO]}\right)^2 + \left(\frac{\Delta[NO_2] \cdot j(NO_2)]}{k_{NO+HO_2} \cdot [NO]}\right)^2 + \left(\frac{\Delta[O_3] \cdot k_{NO+O_3} \cdot [NO]}{k_{NO+HO_2} \cdot [NO]}\right)^2 + \\ \left(\Delta[NO]\left(\frac{-k_{NO+O_3} \cdot [O_3] \cdot k_{NO+HO_2} \cdot [NO] - k_{NO+HO_2} \cdot (j(NO_2) \cdot [NO_2] - k_{NO+O_3} \cdot [O_3][NO])}{\left(k_{NO+HO_2} \cdot [NO]\right)^2}\right)\right)^2 \end{array}}$$

Over the course of the campaign, the median relative $RO_2$ uncertainty is 56 %. The average is 132 % and heavily biased by single data outliers and therefore not representative. The relative error associated with the $RO_2$ calculation is hence estimated at 56 %.

The manuscript has also been revised on Page 23, L470: The relative uncertainty associated with the NOPR estimate has been calculated by error propagation of Eq. 7.

$$R(NOPR) = \frac{1}{NOPR} \cdot \sqrt{\begin{array}{l} \left(k_{NO+HO_2} \cdot [NO] \cdot \Delta[RO_2]\right)^2 + \left(k_{NO+HO_2} \cdot \Delta[NO] \cdot [RO_2]\right)^2 + \\ \left(\Delta[O_3] \cdot \left(\alpha \cdot j(O^1D) + k_{OH+O_3} \cdot [OH] + k_{HO_2+O_3} \cdot [HO_2]\right)\right)^2 + \\ \left(\Delta j(O^1D) \cdot [O_3] \cdot \alpha\right)^2 + \left(j(O^1D) \cdot [O_3] \cdot \Delta\alpha\right)^2 + \\ \left(\Delta[OH] \cdot k_{OH+O_3} \cdot [O_3]\right)^2 + \left(\Delta[HO_2] \cdot k_{HO_2+O_3} \cdot [O_3]\right)^2 \end{array}}$$

Incorporating a relative error of 56 % associated with $RO_2$, the median of the relative NOPR error of all data points obtained during AQABA is 69 %. The average relative uncertainty of NOPR is 15 % and strongly biased by single data outliers, which are in the case of NOPR significantly negative (due to fresh emissions and titration of $O_3$ by NO). Again the median is a more representative measure for the general uncertainty associated with the NOPR calculations. The relative error associated with the NOPR estimates based on measured data is hence estimated at 69 %.

Comment 7. Your response was confusing. NOPR is expected to vary between noon and noon – 3 hours. In order to calculate NOPR, jNO2, O3, NO, and NO2 should be reasonably constant over a 5-minute period. One should look at the change in jNO2 over 5 minutes relative to its mean value over that 5-minute period (not relative to its mean value at noon).

The variation of $j(NO_2)$ over 5 minutes relative to its mean value of that 5-minute period is generally less than 1 % within a time frame of $\pm 2$ h around noon. The variation of $j(NO_2)$ within a time frame $\pm 3$ h around noon increases to about 2 %, for particular days also more than 2 %. Both values represent sufficiently slow changes. However we would like to stress that for our calculation it is more important to have constant actinic flux conditions than slow changes, which will yield a substantial deviation from the daytime value within a $\pm 3$ h window from noon (the difference is about 20 %) compared to a change of less than 7 % within the $\pm 2$ h window (with respect to the maximum noontime value).

**Reply to Reviewer Report 2**

*In the following the comments of the reviewer are presented (black) alongside with our replies (in blue) and changes made to the manuscript (in red).*

General statement: The authors have responded adequately to most of my concerns. However, there are still some issues which have not been properly addressed. I list my initial concern for those sections which the authors have not responded properly and then include my comment [new remark] to the authors' reply.

Dear reviewer, we appreciate your reviewing our revised manuscript and providing insightful comments. Below we provide detailed responses to your new remarks.

It would be nice to see a break-down of the different terms in Eq. 7 for different legs as shown in Fig. 9 to evaluate what processes might be most relevant/different in those different legs. **New remark 1:** I appreciate the new figures S10-S13. It would be good to include some discussion associated with these figures, as at times some notable deviation of "Estimated based on measured data" from "Estimated based on simulated data" occurs (e.q. in S12 and S13).

An additional paragraph has been added to section 3.3. The manuscript has been revised on Page24 L502f: Measured OH and $HO_2$ as well as $RO_2$ estimated based on measured data are generally underestimating the concurrent simulated data. Speaking in terms of absolute amounts, we find that the break-down loss and productions terms of Eq. 7 (NOPR) based on measured data are generally underestimating the results based on simulated data. The deviations between measurement and model pretty much represent the differences observed in the noontime concentrations of the mentioned tracers. Largest deviations of the break-down loss terms, associated with reactions of $O_3$ with OH and $HO_2$, are found over the OG and AG, where also OH and $HO_2$ is significantly overestimated in the model. In the case of $j(O^1D) \cdot \alpha \cdot [O_3]$ a slight overestimation by the estimate based on simulated data compared to the estimate based on measured data is observed. This is due to simulated absolute humidity being slightly larger than the concurrent measured data. Also we find that the break-down production term $k_{NO+HO_2} \cdot [NO] \cdot [RO_2]$ estimated based on simulated data is generally larger than the estimate based on measured data. This pretty much reflects that noontime $RO_2$ is overestimated in the model by a factor of 2, except for the Arabian Gulf where fair agreement is found.

The loss mechanism through H2O is important. Also, it seems to vary a lot. Some parts of the ship cruising legs might have already been exposed high humidity due to the Indian monsoon system. It would be good to see the absolute humidity variation along the legs similar to Figs 3 and S4.

**New remark 2:** I appreciate the new figure S5, but I am missing some discussion related to S5.

Some discussion related to the absolute humidity variation has been added on Page 14, L 331. The manuscript now says: Figure S5 shows that absolute humidity observed during AQABA ranges from lowest values of less than 1 % observed in the Suez Golf during the first leg to about 3 % observed during both legs in the southeastern part of the Arabian Gulf and in the Strait of Hormuz. Although observing highest absolute humidity on both legs in the southeastern part of the Arabian Gulf, absolute humidity was very low on the first leg near Kuwait, where absolute humidity was about 1 %. These air masses were brought from the Kuwait/Iraq into the MBL of the Arabian Gulf on the first leg, whereas a change of wind direction for the second leg resulted in winds coming from Iran area with moister air. For the rest of the cruise, absolute humidity mixing ratio was about 1.5 % with variations being generally less than 0.5 %.

Authors mention NOx values of several hundred ppbs. Where do they show up in Figs. 3, S2, and S3? What were the megacities along the cruising legs? I could think about Cairo, but according to Fig 3 NOx values do not show extremely high values.

**New remark 3:** I did not see changes in the text. (1) The authors still mention several hundred of ppbv NOx, although in their response they say they have removed those from the final data set, as contamination from the ship exhaust could not be excluded. (2) I am not sure what Megacity definition the authors are thinking about, but I am used to a definition of a Megacity as having at least 10 million inhabitants. I assume this only applies to Cairo.

220 Keeping these statements the way the authors wrote is misleading.

You are right. The manuscript has been revised. Now it says on P13, L316: During AQABA NO$_x$ mixing ratios varied over three orders of magnitude with lowest values of less than 50 ppt$_v$ observed in relatively pristine regions and highest values of more than 10 ppb$_v$ found in the vicinity of areas with strong anthropogenic influence or nearby passing ships.

225

This section should include some more explanations: it seems there is a huge variation in NOx and O3 in AG (also a huge variation in NOPR as shown in Fig 9). What is the major driver of this: point sources from ships? Why are the highest NOx values in OG and why are some of the lowest O3 values found in OG? Why would you consider

230 air masses over the Mediterranean as photochemically aged air masses due to the small whisker-interval, while the whisker-plots for AS and OG show pretty much the same with, but at much lower absolute O3 ranges. There are no emission sources in that area of the Mediterranean?

**New remark 4:** Still it is not clear, why there are highest NOx values in OG. It should be spelled explicitly what sources those might have been, even if this information might have already been given in other papers. Here it is

235 critical to mention/repeat this information, as it obviously has a major impact on O3.

Major drivers of high NO$_x$ over the Oman Gulf were ship point sources. At this point we can only guess why NO$_x$ was highest over the OG, and this can be partly explained by the immediate vicinity of point sources in this region, which lead to higher NO$_x$ (before it is lost by reaction with OH and deposition to the surface) and titration of O$_3$ (note the relatively low regional O$_3$ median of 31.5 ppb$_v$). The manuscript has been revised. Now it says on P16,

240 L359: Although observing highest NO$_x$ over the Oman Gulf, O$_3$ observed over the Oman Gulf was amongst the lowest detected throughout the whole campaign, which can partly be explained the fact that high NO$_x$ eventually leads to ozone destruction. The immediate vicinity of point sources in this region, which leads to higher NO$_x$ (before it is lost by reaction with OH and deposition to the surface) and titration of O$_3$ (note the low regional O$_3$ median of 31.5 ppb$_v$), may partly explain why NO$_x$ was highest over the Gulf of Oman.

245

Actually, Figure 10 shows that in almost all areas O3 formation is NOx limited. However, the authors say that this is typical for photochemically aged air masses over the Mediterranean. As already mentioned further above, why do the authors explicitly consider the Mediterranean area having aged air masses? It is even more surprising as

250 the results for the Mediterranean area in Figure 10 indicate that the Box-Whisker plot stretches into the transition between NOx and VOC limitation.

**New remark 5:** I think these lengthy discussion about local differences of NOx-VOC limitations vs conditions of the larger Mediterranean area do not provide new insights. It pretty much resembles studies at any other location, i.e. the closer to a fossil fuel combustion emission source the fresher and least photochemically aged processed the

255 pollution plume is. Here, it is about ship point sources. So what?

The regional size of the Mediterranean Basin is significantly larger than that of the other investigated regions except for the OG and AS, which can be considered open towards the Indian Ocean. Air masses observed along the ship cruise in the Mediterranean hence do not only include ship point sources from nearby and from larger distances in the Mediterranean, but also air masses with continental influence. Deduced from the relatively low NO$_x$ in the

260 Mediterranean, the integral effect of ship point sources is rather small. Instead and this directly reproduces and constrains previous studies (Destroff et al., 2017), Etesian winds bring air masses from the broader Southeastern Europe into the marine boundary layer of the Mediterranean. Note that most of the time measuring in the Mediterranean Basin was spent in the eastern part which is directly influenced by these regional wind patterns.

265

Why would higher NOx lead to higher O3 pollution? For instance, according to Figure 4, OG has the highest NOx values, but also pretty low O3 values. With regard to NOPR, the Box-Whisker plot for OG shows positive, but also large negative values. In any case NOPR values are significantly lower than for AG, for instance.

**New remark 6:** This somehow ties into the above question about where the high NOx in OG comes from. According to the authors it is from "increased shipping in the Arabian Gulf". A few follow-up questions: Why is shipping "increased" in the Arabian Gulf (is it higher than during any other time)? Why is there a further increase of NOx in OG? Are there more ships than in the Arabian Gulf? I think it should be the same number of ships, assuming that (1) most of them are oil-tankers and (2) the number of those ships entering and leaving the Strait of Hormuz would be the same.

You are right. The number of ships in the Arabian Gulf is equal to the number of ships in the Oman Gulf, which is also reproduced in the $NO_x$ distributions for these two regions. The term "increased shipping" on Page 26, L 539 over the AG is a clear double statement, which is misleading. In the manuscript it says at the moment: "Note that a further increase in $NO_x$-emissions from "increased shipping" in the Arabian Gulf." The term "increased" in front of "shipping" has been removed. This sentence picks up the $NO_x$ sensitivity of the Arabian Gulf and that an increase in $NO_x$-emissions may initially lead to higher ozone production, before ozone production decrease and the chemistry changes into VOC-sensitivity. Now it on says on Page 26, L539f: Note that a further increase in $NO_x$-emissions from shipping in the Arabian Gulf may initially lead to higher ozone production.

The legend mentions "Measurements", the figure captions says "estimated". From Eq 3 I understand that RO2 was neither measured nor estimated, but calculated. Also, what would be the interpretation of the negative RO2 concentrations (blue Box- Whisker plots) when calculated from Eq 3?

**New remark 7:** Still, it is not clear. Why is "estimated", when it is either measured or simulated?

You are right. $RO_2$ has been estimated based on measured data. Simulated $RO_2$ is the sum of all peroxy radicals $R_iO_2$ with less than four carbon atoms as given in the supplements. The caption of Figure 7 has been revised to: Comparison of Box-Whisker-Plots of the regional estimated noontime $RO_2$ median based on measured data and simulated $RO_2$ data for the period from 18 July 2017 onwards.

**Further changes**

295     **Change 1:** The TMU of $NO_2$ has been revised to 16 % in Table 1.

    **Change 2**: A dot in the middle of the sentence on Page 26, L546 has been removed.

[revised manuscript text omitted]

---

## Author Response (AR3)

**Reply to Editor comments**

*In the following the comments of the editor are presented (black) alongside with our replies (in blue) and changes made to the manuscript (in red).*

General statement: I am pleased the authors have addressed the referees' comments in detail. With respect to the main comment by referee #2, the treatment of error propagation has been adequately corrected. Nevertheless, I am not fully satisfied with the error estimates.

Dear editor, we appreciate your positive feedback. Below we provide detailed responses to your comments.

**Comment 1:** The CLD instrument measures NO and NOc by chemiluminescence. The NO detection involves photon counting and the statistical noise of the signals should follow Poisson statistics. The absolute $1\sigma$ precision should scale with the square root of the NO signal for a given integration time, whereas the relative precision (in percent) is expected to decrease with the square root of the NO concentration. In line 156 of their revised manuscript, the authors specify a general constant value for the relative precision (5%). How can that be? Same arguments apply to the measurement of NOc.

It is true that the detection involves photon counting and that the statistical noise of each signal should follow Poisson statistics. However, the precision, which is used for calculating the total measurement uncertainty of both NO and $NO_c$, is defined as the reproducibility of all in-field calibrations. As the number of counts obtained for each 5 second calibration data point is large (> 50.000, yielding a relative Poisson uncertainty (noise) of less than 0.5 %), we find that the precision of both channels is mainly determined by drifts in the detector sensitivity (and practically not by Poisson noise of the signal). We have added a sentence after line 157, Page 7: Note that the precision is calculated from the reproducibility of all in-field calibrations, which is mainly determined by drifts in the detector sensitivity rather than by statistical Poisson noise of the measured signal.

**Comment 2:** In line 157, the authors calculate a total relative uncertainty of NO from the signal precision (5%) and the error of the calibration gas mixture (3%). The same is done for NOc. In the calculation of the total uncertainty for NO2, the total uncertainties for NO and NOc are treated as statistically independent. This is not correct, because the calibrations of NO and NOc rely on the same calibration standard. The correct procedure to calculate the uncertainty of NO2 would be to calculate first the precision of NO2 from the precisions of NO and NOc. In a second step, the precision of NO2 can be combined with the uncertainty from the calibration mixture.

We agree that NO and $NO_c$ cannot be considered statistically independent, already by the fact that both channels are sensitive to ambient NO (the $NO_c$-channel further measures $NO_2$). However, and this should be clear from our statement to "Comment 1", it is not possible to calculate the precision of $NO_2$ as described above, simply as the chemiluminescence technique does not involve a direct $NO_2$ measurement (which could be used to calculate the reproducibility of calibrations associated with drifts in the detector sensitivity). Instead, we can estimate the relative uncertainty of $NO_2$ by use of the estimation of the largest error possible of statistically dependent variables NO and $NO_c$.

$$\text{TMU}([NO_2]) = \frac{1}{[NO_2]} \cdot \left( \left| \frac{\Delta[NO_c]}{K_e} \right| + \left| \frac{\Delta[NO]}{K_e} \right| + \left| \frac{\Delta K_e \cdot ([NO_c] - [NO])}{K_e^2} \right| \right)$$

This yields a campaign average and median measurement uncertainty of 23 % and 13 %, respectively, instead of 16 % and 8 %, respectively. We have revised the error formula for $NO_2$ and the given $NO_2$ uncertainty to 23 % (instead of 16%) in the manuscript. On Page 7, Line 161 now it says: The TMU for $NO_2$ has been estimated as the largest error possible of the statistically dependent variables NO and $NO_c$.

$$\text{TMU}([NO_2]) = \frac{1}{[NO_2]} \cdot \left( \left| \frac{\Delta[NO_c]}{K_e} \right| + \left| \frac{\Delta[NO]}{K_e} \right| + \left| \frac{\Delta K_e \cdot ([NO_c] - [NO])}{K_e^2} \right| \right)$$

Note that propagating a 23 % uncertainty in $NO_2$ yields revised median relative uncertainties associated with the $(HO_2 + RO_2)$ and NOPR estimate of 74 % and 91 %, respectively. A 23 % $NO_2$ relative uncertainty yields average relative uncertainties of $(HO_2 + RO_2)$ and NOPR of 176 % and 21 %, respectively. We have corrected these values in the manuscript on Page 21, L 445: Over the course of the campaign, the median relative $(HO_2+RO_2)$ uncertainty is 74 %. The average is 176 % and heavily biased by single data outliers and therefore not representative. The relative error associated with the $(HO_2+RO_2)$ calculation is hence estimated at 74 %. And on Page 24, L499: Incorporating a relative error of 74 % associated with $(HO_2+RO_2)$, the median of the relative NOPR error of all data points obtained during AQABA is 91 %. The average relative uncertainty of NOPR is 21 % and strongly biased by single data outliers, which are in the case of NOPR significantly negative (due to fresh emissions and titration of $O_3$ by NO). Again the median is a more representative measure for the general uncertainty associated with the NOPR calculations. The relative error associated with the NOPR estimates based on measured data is hence estimated at 91 %.

**Comment 3:** In line 166, the authors assume that the error of the NO measurement is zero when no NO is present. However, when NO approaches zero, the signal noise will be dominated by the instrumental background (e.g., dark signal) which is responsible for the limit of detection. The role of background noise should be clarified.

This is true. Also since the equation on Page 8, Line 166: $\sqrt{(6\%^2+0\%^2+3\%^2)} \approx 6.7$ % used to calculate the $NO_2$ uncertainty during nighttime does not apply anymore, the error in $NO_2$ is instead calculated as the largest error possible from statistically dependent variables. We now compare the median and average to give a conservative upper limit for the $NO_2$ uncertainty. We have rewritten the following passage on Page 8, L164: "The median is lower than the average as NO is practically zero during nighttime. During nighttime, the relative error of the $NO_2$ data points is about 6.7 %. Note for convenience that is exactly the value calculated from errors in quadrature (if NO is practically zero during nighttime): $\sqrt{(6\%^2+0\%^2+3\%^2)} \approx 6.7$ %."

To: Over the course of the campaign the median and the average relative uncertainty of $NO_2$ are 13 % and 23 %, respectively. The relative uncertainty in $NO_2$ has been estimated as a conservative upper limit at 23 % as the average of the relative uncertainties of all data points obtained during AQABA.

**Comment 4:** In the paper, HO2 data are called "preliminary" because it has not been corrected for interferences from organic peroxy radicals. Since the discovery of possible RO2 interferences in the detection of HO2 by LIF, uncorrected HO2 is generally called HO2* (e.g., Lu et al., Atmos. Chem. Phys., 12, 1541–1569, 2012). I recommend to adopt this nomenclature.

We have adopted the suggested nomenclature in the manuscript (text, tables, legends and captions of figures).

Additionally on Page 9, L 200 now it says: As $HO_2$ data are preliminary in this study, they will be called $HO_2*$.

**Comment 5:** In line 199, the possible RO2 interference is treated as a statistical error when calculating the uncertainty of HO2. This is not appropriate. The interference is an additive positive bias. If the interference is 7%, then all HO2* values will be too high by this amount. Please clarify: where does the value of 7% come from? Why don't you correct the HO2* data by this amount?

We agree that the interference is generally an additive uncertainty rather than a statistical uncertainty, however note that 7 % (or 3 ppt, whichever value is larger) represents the **largest** uncertainty due to not yet corrected interference of organic peroxy radicals $RO_2$. The real quantitative interference remains unknown at this point.

**Comment 6**: In Table 1, I have the impression that TMU is used either for precision, or accuracy, or a combination of both. I suggest to specify instead the limit of detection and accuracy (usually related to calibration) for each substance in separate columns.

We agree that the TMUs given in Table 1 have been estimated differently, however, it has been non-trivial to compare and cluster the errors and total measurement uncertainties of each data set and observation to a general statement. The values given in Table 1 represent the relative uncertainties which were used in this study to determine the relative uncertainty associated with the ($HO_2$ + $RO_2$) and NOPR estimate. Only in the case of OH and $HO_2$, the uncertainty of the data sets were estimated as the 1 sigma accuracy of the respective measurement as these values give a good representation of the uncertainty associated with the OH and $HO_2$ data sets. Therefore we have renamed the column "TMU" as Associated relative uncertainty. Also we have rewritten the caption of Table 1 to: List of observations and gas phase measurements during AQABA. The relative uncertainty associated with each data set is given. Note that for OH and $HO_2$*, the relative uncertainty is estimated as the 1 sigma accuracy (and the 7 % interference of $RO_2$ in the case of $HO_2$* data). In addition, a reference of the measurement operability is given.

**Comment 7:** Equation (4) in line 271 looks weird. I find the re-definition of RO2 as the total sum of peroxy radicals (including HO2) confusing and inconsistent with other parts of the paper (e.g., line 96, line 239, reactions R4-R6). According to the general nomenclature, "RO2" should be used for the sum of organic peroxy radicals without HO2. I suggest to modify the equation accordingly:

P(O3) = kNO+RO2 [NO] [RO2] + kNO+HO2 [NO] [HO2]) = kNO+HO2 [NO] ([RO2] + [HO2])

We also state that $RO_2$ is used as the sum of organic peroxy radicals without $HO_2$. Note that this necessitates a revision of the manuscript as, for instance, the initial $RO_2$ estimate is now called ($HO_2$+$RO_2$) estimate. The manuscript has been revised carefully in the different places.

Page 11, Line 252 now says: …, we can combine the sum of all organic peroxy radicals $R_iO_2$ to the entitiy $RO_2$. The sum of $HO_2$ and $RO_2$ can be estimated using the steady state equation

$$[HO_2] + [RO_2] = \frac{j(NO_2) \cdot [NO_2] - k_{NO+O_3} \cdot [NO][O_3]}{k_{NO+HO_2} \cdot [NO]}. \tag{3}$$

We have modified P12, L269 and Eq. 4 accordingly: The production of ozone can be approximated by the rate of oxidation of NO with $HO_2$ and $RO_2$ to form $NO_2$ that will rapidly form $O_3$ (R1-R2) (Parrish et al., 1986, Thornton et al., 2002; Bozem et al., 2017).

$$P(O_3) = k_{NO+HO_2}[NO] \cdot ([HO_2] + [RO_2]) \tag{4}$$

Also we have modified Eq. 7 to:

$$NOPR = k_{NO+HO_2}[NO] \cdot ([HO_2] + [RO_2]) - [O_3] \cdot (\alpha \cdot j(O^1D) + k_{OH+O_3}[OH] + k_{HO_2+O_3}[HO_2]). \tag{7}$$

Futhermore, the term "$RO_2$ estimate" has been replaced by "($HO_2$+$RO_2$) estimate" at various places throughout the whole manuscript and supplements (text, legends, captions). The error formulas for ($HO_2$+$RO_2$) and NOPR given in the manuscript have also been revised.

[revised manuscript text omitted]

$$\frac{1}{[\text{HO}_2] + [\text{RO}_2]} \cdot \sqrt{ \begin{array}{c} \left(\dfrac{\Delta j(\text{NO}_2) \cdot [\text{NO}_2]}{k_{\text{NO+HO}_2} \cdot [\text{NO}]}\right)^2 + \left(\dfrac{\Delta [\text{NO}_2] \cdot j(\text{NO}_2)]}{k_{\text{NO+HO}_2} \cdot [\text{NO}]}\right)^2 + \left(\dfrac{\Delta [\text{O}_3] \cdot k_{\text{NO+O}_3} \cdot [\text{NO}]}{k_{\text{NO+HO}_2} \cdot [\text{NO}]}\right)^2 + \\ \left(\Delta[\text{NO}]\left(\dfrac{-k_{\text{NO+O}_3} \cdot [\text{O}_3] \cdot k_{\text{NO+HO}_2} \cdot [\text{NO}] - k_{\text{NO+HO}_2} \cdot (j(\text{NO}_2) \cdot [\text{NO}_2] - k_{\text{NO+O}_3} \cdot [\text{O}_3][\text{NO}])}{\
[revised manuscript text omitted]

Figure S1 shows a day to day calculation of the correction factor to scale the fractional noontime integral to a diurnal value. An average $\pm$ standard deviation of $(46.1 \pm 2.8)$ % of the diurnal integral within $\pm$ 2h around noontime was estimated.

[Figure]

**Figure S1: Ratio of the noontime actinic flux ($\pm$ 2h around noon) with regard to the total actinic flux $j$(NO₂) of that particular day. Dashed line represents the campaign average of $(46.1 \pm 2.8)$ %. The errors bars are represented by the relative uncertainty (6 %) of the campaign average.**

[Figure]

**Figure S2: Timeline of NO, NO₂ (both CLD), O₃, OH, HO₂ preliminary (HO₂\*) and *j*(NO₂) data during the first leg. See Table ST2 for additional information on the ship cruise. Note that HO₂ data are preliminary.**

1005

[Figure]

**Figure S3: Timeline of NO, NO₂ (both CLD), O₃, OH, HO₂ preliminary (HO₂\*) and *j*(NO₂) data during the second leg. See Table ST2 for additional information on the ship cruise. Note that HO₂ data are preliminary.**

[Figure]

1010   **Figure S4: Ship cruises with color-scaled OH mixing ratios a) during the first and b) the second leg and color-scaled HO₂* mixing ratios c) during the first and d) the second leg. Note that OH and HO₂* data have been filtered for own stack contamination.**

[Figure]

**Figure S5: Ship cruises with color-scaled absolute humidity a) during the first and b) the second leg.**

1015

[Figure]

**Figure S6: Scatterplot of simulated and measured regional NO$_x$ median in ppb$_v$. 1:1 line added for orientation. The error bars represent the 25-75-percentile variation.**

[Figure]

1020 Figure S7: Scatterplot of simulated and measured regional O₃ median in ppbᵥ. 1:1 line added for orientation. The error bars represent the 25-75-percentile variation.

[Figure]

**Figure S8: Scatterplot of simulated regional (HO$_2$+RO$_2$) median and regional (HO$_2$+RO$_2$) median estimated based on measured tracer data in ppt$_v$. 1:1 line added for orientation. The error bars represent the 25-75-percentile variation.**

[Figure]

1025

**Figure S9: Scatterplot of median NOPR in ppb$_v$ day$^{-1}$ estimated based on simulated and measured tracer data. 1:1 line added for orientation. The error bars represent the 25-75-percentile variation.**

[Figure]

1030

Figure S10: Comparison of the regional, absolute contribution of $k_{\mathrm{NO+HO_2}}[\mathrm{NO}]([\mathrm{HO_2}] + [\mathrm{RO_2}])$ to NOPR in the six different regions investigated during AQABA. The horizontal black bar indicates the median value, the box the 25- and 75-percentiles and the whiskers the 10- and 90-percentiles.

[Figure]

1035    **Figure S11: Comparison of the regional, absolute contribution of** $-j(O^1D) \cdot \alpha \cdot [O_3]$ **to NOPR in the six different regions investigated during AQABA. The horizontal black bar indicates the median value, the box the 25- and 75-percentiles and the whiskers the 10- and 90-percentiles.**

[Figure]

1040

**Figure S12: Comparison of the regional, absolute contribution of** $-k_{HO_2+O_3}[HO_2][O_3]$ **to NOPR in the six different regions investigated during AQABA. The horizontal black bar indicates the median value, the box the 25- and 75-percentiles and the whiskers the 10- and 90-percentiles.**

[Figure]

1045

**Figure S13: Comparison of the regional, absolute contribution of** $-k_{OH+O_3}[OH][O_3]$ **to NOPR in the six different regions investigated during AQABA. The horizontal black bar indicates the median value, the box the 25- and 75-percentiles and the whiskers the 10- and 90-percentiles.**

1050    **Table ST1: Range of latitudinal and longitudinal coordinates and dates during both legs of the different regions.**

| region (abbreviation) | latitudinal range | longitudinal range | Date (1st leg) | Date (2nd leg) |
| --- | --- | --- | --- | --- |
| Mediterranean (M) | 31.810° N- 39.923° N | 12.620° E- 31.850° E | --- | 25.08.2017 – 31.08.2017 |
| Northern Red Sea (NRS) | 23.343° N- 30.986° N | 32.305° E- 37.085° E | 03.07.2017 – 08.07.2017 | 21.08.2017 – 24.08.2017 |
| Southern Red Sea (SRS) | 12.672° N- 22.494° N | 37.411° E- 43.327° E | 09.07.2017 – 16.07.2017 | 17.08.2017 – 20.08.2017 |
| Arabian Sea (AS) | 11.797° N- 22.782° N | 44.035° E- 60.636° E | 18.07.2017 – 24.07.2017 | 07.08.2017 – 16.08.2017 |
| Oman Gulf (OG) | 23.050° N- 25.622° N | 56.492° E- 59.913° E | 24.07.2017 – 27.07.2017 | 05.08.2017 – 07.08.2017 |
| Arabian Gulf (AG) | 25.396° N- 29.425° N | 47.920° E- 56.772° E | 27.07.2017 – 31.07.2017 | 03.08.2017 – 05.08.2017 |

**Table ST2: Overview of the time spent in the particular regions during AQABA. Red color indicates periods with KI at anchor that are not included in the data analysis. Data measured during bunkering at Fujairah City (06 August, 07:00-15:00 UTC) were also not included in the analysis.**

| Date | Region | Date | Region |
|---|---|---|---|
| 03.07.2017 | Northern Red Sea | 02.08.2017 | Kuwait port |
| 04.07.2017 | Northern Red Sea | 03.08.2017 | Kuwait port/Arabian Gulf |
| 05.07.2017 | Northern Red Sea | 04.08.2017 | Arabian Gulf |
| 06.07.2017 | Northern Red Sea | 05.08.2017 | Arabian Gulf/Oman Gulf |
| 07.07.2017 | Northern Red Sea | 06.08.2017 | Oman Gulf |
| 08.07.2017 | Northern Red Sea | 07.08.2017 | Oman Gulf/Arabian Sea |
| 09.07.2017 | Southern Red Sea | 08.08.2017 | Arabian Sea |
| 10.07.2017 | Southern Red Sea | 09.08.2017 | Arabian Sea |
| 11.07.2017 | Southern Red Sea/Jeddah port | 10.08.2017 | Arabian Sea |
| 12.07.2017 | Jeddah port | 11.08.2017 | Arabian Sea |
| 13.07.2017 | Jeddah port/Southern Red Sea | 12.08.2017 | Arabian Sea |
| 14.07.2017 | Southern Red Sea | 13.08.2017 | Arabian Sea |
| 15.07.2017 | Southern Red Sea | 14.08.2017 | Arabian Sea |
| 16.07.2017 | Southern Red Sea | 15.08.2017 | Arabian Sea |
| 17.07.2017 | Djibouti port | 16.08.2017 | Arabian Sea |
| 18.07.2017 | Arabian Sea | 17.08.2017 | Southern Red Sea |
| 19.07.2017 | Arabian Sea | 18.08.2017 | Southern Red Sea |
| 20.07.2017 | Arabian Sea | 19.08.2017 | Southern Red Sea |
| 21.07.2017 | Arabian Sea | 20.08.2017 | Southern Red Sea |
| 22.07.2017 | Arabian Sea | 21.08.2017 | Northern Red Sea |
| 23.07.2017 | Arabian Sea | 22.08.2017 | Northern Red Sea |
| 24.07.2017 | Arabian Sea/Oman Gulf | 23.08.2017 | Northern Red Sea |
| 25.07.2017 | Oman Gulf | 24.08.2017 | Northern Red Sea |
| 26.07.2017 | Oman Gulf | 25.08.2017 | Mediterranean |
| 27.07.2017 | Oman Gulf/Arabian Gulf | 26.08.2017 | Mediterranean |
| 28.07.2017 | Arabian Gulf | 27.08.2017 | Mediterranean |
| 29.07.2017 | Arabian Gulf | 28.08.2017 | Mediterranean |
| 30.07.2017 | Arabian Gulf | 29.08.2017 | Mediterranean |
| 31.07.2017 | Arabian Gulf/Kuwait port | 30.08.2017 | Mediterranean |
| 01.08.2017 | Kuwait port | 31.08.2017 | Mediterranean |

**Table ST3: Overview of measured NO$_x$ (upper table) and measured O$_3$ (lower table) spatial volume mixing ratio average, standard deviation, 1$^{st}$ quantile, median, 3$^{rd}$ quantile (all in ppb$_v$) and number of considered data points.**

| NO$_x$ (upper), O$_3$ (lower) | Mediterranean | Northern Red Sea | Southern Red Sea | Arabian Sea | Oman Gulf | Arabian Gulf |
|---|---|---|---|---|---|---|
| data points | 1767 | 1694 | 1755 | 2656 | 1056 | 1539 |
| average | 1.24 | 4.69 | 1.62 | 0.95 | 4.16 | 3.65 |
| stdev | 3.34 | 7.9 | 3.7 | 3.15 | 4.33 | 9.24 |
| 1$^{st}$ quantile | 0.12 | 0.68 | 0.18 | 0.10 | 1.03 | 0.52 |
| median | **0.25** | **1.76** | **0.46** | **0.19** | **2.74** | **1.26** |
| 3$^{rd}$ quantile | 0.96 | 5.68 | 1.6 | 0.54 | 5.92 | 3.47 |
| data points | 2010 | 2717 | 2307 | 4130 | 1249 | 1809 |
| average | **61.56** | **63.39** | **50.35** | **21.53** | **34.04** | **73.99** |
| stdev | 8.25 | 18.45 | 12.96 | 6.8 | 11.27 | 35.68 |
| 1$^{st}$ quantile | 57.05 | 53.51 | 40.68 | 17.45 | 26.66 | 53.08 |
| median | **61.54** | **64.16** | **46.93** | **22.52** | **31.5** | **62.5** |
| 3$^{rd}$ quantile | 66.48 | 75.51 | 60.28 | 26.19 | 38.03 | 90.42 |

**Table ST4: Overview of simulated NO$_x$ (upper table) and simulated O$_3$ (lower table) spatial volume mixing ratio average, standard deviation, 1$^{st}$ quantile, median, 3$^{rd}$ quantile (all in ppb$_v$) and number of considered data points.**

| NO$_x$ (upper), O$_3$ (lower) | Mediterranean | Northern Red Sea | Southern Red Sea | Arabian Sea | Oman Gulf | Arabian Gulf |
|---|---|---|---|---|---|---|
| data points | 2012 | 2719 | 2310 | 4464 | 1253 | 1810 |
| average | 0.84 | 1.27 | 1.13 | 0.31 | 1.88 | 1.91 |
| stdev | 0.75 | 1.97 | 0.62 | 0.29 | 1.47 | 1.37 |
| 1$^{st}$ quantile | 0.33 | 0.43 | 0.67 | 0.14 | 0.82 | 1.17 |
| median | **0.43** | **0.76** | **0.97** | **0.18** | **1.59** | **1.61** |
| 3$^{rd}$ quantile | 1.27 | 1.08 | 1.51 | 0.39 | 2.05 | 2.16 |
| data points | 2012 | 2719 | 2310 | 4464 | 1253 | 1810 |
| average | **65.15** | **64.76** | **49.5** | **36.91** | **55** | **76.85** |
| stdev | 4.99 | 7.7 | 7.21 | 3.87 | 9.94 | 12.8 |
| 1$^{st}$ quantile | 61.35 | 60.34 | 43.47 | 34.12 | 48.37 | 65.88 |
| median | **65.53** | **64.6** | **49.33** | **36.54** | **55.01** | **76.39** |
| 3$^{rd}$ quantile | 69.33 | 68.25 | 54.58 | 38.85 | 61.42 | 86.22 |

**Table ST5:** Overview of noontime $(HO_2+RO_2)$ spatial volume mixing ratio average, standard deviation, 1$^{st}$ quantile, median, 3$^{rd}$ quantile (all in ppt$_v$) estimated based on measured tracer data. Number of considered data points added in the first line.

| HO$_2$+RO$_2$ | Mediterranean | Northern Red Sea | Southern Red Sea | Arabian Sea | Oman Gulf | Arabian Gulf |
|---|---|---|---|---|---|---|
| data points | 288 | 126 | 190 | 338 | 166 | 242 |
| average | 13 | 27 | 7 | 64 | 23 | 94 |
| stdev | 24 | 20 | 34 | 83 | 42 | 113 |
| 1$^{st}$ quantile | 1 | 15 | -13 | 23 | -8 | 11 |
| median | **16** | **28** | **15** | **33** | **22** | **73** |
| 3$^{rd}$ quantile | 26 | 44 | 27 | 54 | 49 | 176 |

**Table ST6:** Overview of simulated noontime $(HO_2+RO_2)$ spatial volume mixing ratio average, standard deviation, 1$^{st}$ quantile, median, 3$^{rd}$ quantile (all in ppt$_v$). Number of considered data points added in the first line.

| HO$_2$+RO$_2$ | Mediterranean | Northern Red Sea | Southern Red Sea | Arabian Sea | Oman Gulf | Arabian Gulf |
|---|---|---|---|---|---|---|
| data points | 336 | 192 | 192 | 720 | 203 | 293 |
| average | 41 | 46 | 36 | 40 | 48 | 49 |
| stdev | 5 | 4 | 11 | 5 | 6 | 7 |
| 1$^{st}$ quantile | 36 | 43 | 25 | 38 | 44 | 44 |
| median | **41** | **46** | **38** | **41** | **50** | **49** |
| 3$^{rd}$ quantile | 46 | 49 | 42 | 43 | 53 | 54 |

**Table ST7:** Overview of NOPR average, standard deviation, 1$^{st}$ quantile, median, 3$^{rd}$ quantile (all in ppb$_v$ day$^{-1}$) estimated based on measured tracer data. Number of considered data points added in the first line.

| NOPR | Mediterranean | Northern Red Sea | Southern Red Sea | Arabian Sea | Oman Gulf | Arabian Gulf |
|---|---|---|---|---|---|---|
| data points | 148 | 114 | 89 | 187 | 84 | 111 |
| average | -3 | 16 | -5 | -67 | -105 | -24 |
| stdev | 43 | 39 | 12 | 576 | 362 | 449 |
| 1$^{st}$ quantile | -5 | 6 | -9 | 1 | -18 | 25 |
| median | **-1** | **16** | **-4** | **5** | **16** | **32** |
| 3$^{rd}$ quantile | 8 | 40 | -1 | 11 | 23 | 65 |

**Table ST8: Overview of NOPR average, standard deviation, 1st quantile, median, 3rd quantile (all in ppb$_v$ day$^{-1}$) estimated based on simulated tracer data. Number of considered data points added in the first line.**

| NOPR | Mediterranean | Northern Red Sea | Southern Red Sea | Arabian Sea | Oman Gulf | Arabian Gulf |
|---|---|---|---|---|---|---|
| data points | 336 | 192 | 192 | 720 | 203 | 293 |
| average | 11 | 16 | 12 | 5 | 30 | 38 |
| stdev | 8 | 12 | 6 | 6 | 11 | 10 |
| 1st quantile | 5 | 8 | 8 | 1 | 22 | 30 |
| median | **8** | **12** | **11** | **2** | **28** | **35** |
| 3rd quantile | 17 | 21 | 15 | 10 | 37 | 46 |

1075

**Table ST9: Overview of measured HCHO/NO$_2$-ratio average, standard deviation, 1st quantile, median, 3rd quantile and number of considered data points.**

| Ratio | Mediterranean | Northern Red Sea | Southern Red Sea | Arabian Sea | Oman Gulf | Arabian Gulf |
|---|---|---|---|---|---|---|
| data points | 203 | 79 | 48 | 252 | 108 | 122 |
| average | 5.4 | 1.5 | 8.5 | 11.1 | 2.7 | 9 |
| stdev | 4.7 | 0.7 | 6.5 | 8.9 | 2.1 | 6.4 |
| 1st quantile | 1.3 | 0.8 | 4.4 | 2.3 | 1 | 2.5 |
| median | **5** | **1.4** | **7.7** | **9.4** | **2.2** | **9.3** |
| 3rd quantile | 7.4 | 2.1 | 9.8 | 16.1 | 3.6 | 12.7 |

1080 **Table ST10: List of included peroxy radicals (with less than four carbon atoms) for the reaction with NO as recommended by Sander et al. (2019).**

| Species |
| --- |
| $HO_2$ |
| $CH_3O_2$ |
| $C_2H_5O2$ |
| $C_2H_5CO_3$ |
| $CH_3CO_3$ |
| C3DIALO2 ($C_3H_3O_4$) |
| $CH_3CHOHO_2$ |
| $CH_3COCH_2O_2$ |
| $CH_3COCO_3$ |
| $CHOCOCH_2O_2$ |
| $CO_2H_3CO_3$ |
| $HCOCH_2CO_3$ |
| $HCOCH_2O_2$ |
| $HCOCO_3$ |
| $HCOCOHCO_3$ |
| $HOC_2H_4CO_3$ |
| $HOCH_2CH_2O_2$ |
| $HOCH_2CO_3$ |
| $HOCH_2COCH_2O_2$ |
| $HOCH_2O_2$ |
| $CH_3CHO_2CH_2OH$ |
| IC3H7O2 (isopropylperoxy radical) |
| NC3H7O2 (propylperoxy radical) |
| $NCCH_2O_2$ |
| $NO_3CH_2CO_3$ |
| $CH_3CHO_2CH_2ONO_2$ |

---

## Author Response (AR4)

**Reply to Editor comments**

*In the following the comments of the editor are presented (black) alongside with our replies (in blue) and changes made to the manuscript (in red).*

General statement: Dear Ivan Tadic, my comments have been well answered. I still have one more question. I noted that the reaction of NO2 with OH has not been discussed. Equation (4) and (7) assume that all NO2 formed by reactions R4+R5 is photolyzed and converted to ozone. However, NO2 can also react with OH forming HNO3, which reduces the net ozone formation rate. How does the daily production rate of HNO3 compare with the NOPR shown in Figs. 8 and 9? I think this point needs some discussion.

Dear editor, we appreciate your additional comment. You are right that the conversion of $NO_2$ to $HNO_3$ via reaction with OH will reduce the net ozone formation rate. Here we assumed that all $NO_2$ formed via reactions R4 and R5 is photolyzed and converted to ozone. This is justified by the finding that the noontime ratio $(k_{NO_2+OH}[NO_2][OH])/(j(NO_2)[NO_2])$ varied between 0 and 1 % (average 0.5 %, standard deviation 0.4 %) during AQABA, yielding that an insignificant amount of $NO_2$ was lost by conversion to $HNO_3$ and most $NO_2$ was converted to ozone during AQABA. We have added a statement on page 12, line 274 following: Here we assumed that all $NO_2$ formed via reactions R4 and R5 is photolyzed and converted to ozone. This is justified by the finding that the noontime ratio $(k_{NO_2+OH}[NO_2][OH])/(j(NO_2)[NO_2])$ varied between 0 and 1 % (average 0.5 %, standard deviation 0.4 %) during AQABA, yielding that an insignificant amount of $NO_2$ was lost by conversion to $HNO_3$ and most $NO_2$ was converted to ozone during AQABA.

We have further corrected the numeration of Reactions R12 and R13 to R10 and R11, respectively, and accordingly changed the respective reaction reference in the text on page 12, line 287 and 288.

[revised manuscript text omitted]

**Regions**

| | |
|---|---|
| AG | Arabian Gulf |
| AS | Arabian Sea |
| M | Mediterranean Sea |
| NRS | Northern Red Sea |
| 670 OG | Oman Gulf |
| SRS | Southern Red Sea |

**Scientific**

[revised manuscript text omitted]